# Cold and hot fibrosis define clinically distinct cardiac pathologies

## Graphical abstract

## Authors

Shoval Miyara, Miri Adler,
Kfir B. Umansky, ..., Avi Mayo, Uri Alon,
Eldad Tzahor

## Correspondence

uri.alon@weizmann.ac.il (U.A.),
eldad.tzahor@weizmann.ac.il (E.T.)

## In brief

Miyara et al. identify two types of fibrosis in cardiac pathologies: "hot fibrosis," involving macrophage-myofibroblast interactions in chronic injuries, and acute-injury-driven "cold fibrosis," controlled by self-maintaining myofibroblasts. This study shows cold fibrosis following acute-myocardial infarction and highlights TIMP1 as a therapeutic target to reduce fibrosis.

## Highlights

- Cold and hot fibrosis are distinct outcomes of acute and chronic cardiac injuries

- The myofibroblast autocrine growth factor loop is a target to reduce cold fibrosis

- TIMP1 is a myofibroblast autocrine growth factor

- Targeting TIMP1 with neutralizing antibodies reduces cold fibrosis after MI in mice

 Miyara et al., 2025, Cell Systems 16, 101198
March 19, 2025 © 2025 The Author(s). Published by Elsevier Inc.

CellPress

## Article

# Cold and hot fibrosis define clinically distinct cardiac pathologies

Shoval Miyara,[1,15] Miri Adler,[2,14,15] Kfir B. Umansky,[1] Daniel Häußler,[3] Elad Bassat,[4,13] Yalin Divinsky,[1] Jacob Elkahal,[1] David Kain,[1] Daria Lendengolts,[1] Ricardo O. Ramirez Flores,[5] Hanna Bueno-Levy,[1] Ofra Golani,[6] Tali Shalit,[7] Michael Gershovits,[7] Eviatar Weizman,[7] Alexander Genzelinakh,[1] Danielle M. Kimchi,[1] Avraham Shakked,[1] Lingling Zhang,[1] Jingkui Wang,[4,13] Andrea Baehr,[8,9] Zachary Petrover,[1] Rachel Sarig,[1] Tatjana Dorn,[10] Alessandra Moretti,[9,10] Julio Saez-Rodriguez,[5,11] Christian Kupatt,[8,9] Elly M. Tanaka,[4,13] Ruslan Medzhitov,[2,12] Achim Krüger,[3] Avi Mayo,[1] Uri Alon,[1,*] and Eldad Tzahor[1,16,*]

[1]Department of Molecular Cell Biology, Weizmann Institute of Science, Rehovot, Israel
[2]Tananbaum Center for Theoretical and Analytical Human Biology, Yale University School of Medicine, New Haven, CT, USA
[3]TUM School of Medicine and Health, Institute of Experimental Oncology and Therapy Research, Technical University of Munich, Munich, Germany
[4]Research Institute of Molecular Pathology (IMP), Vienna BioCenter (VBC), Vienna, Austria
[5]Heidelberg University, Faculty of Medicine, and Heidelberg University Hospital, Institute for Computational Biomedicine, Heidelberg, Germany
[6]Department of Life Sciences Core Facilities, Weizmann Institute of Science, Rehovot, Israel
[7]The Mantoux Bioinformatics institute of the Nancy and Stephen Grand Israel National Center for Personalized Medicine, Weizmann Institute of Science, Rehovot, Israel
[8]Klinik und Poliklinik für Innere Medizin I, University Clinic rechts der Isar, Technical University of Munich, Munich, Germany
[9]DZHK (German Center for Cardiovascular Research), partner site Munich Heart Alliance, Munich, Germany
[10]First Department of Medicine, Cardiology, Klinikum rechts der Isar, Technical University of Munich, School of Medicine and Health, Munich, Germany
[11]European Molecular Biology Laboratory, European Bioinformatics Institute (EMBL-EBI), Hinxton, Cambridgeshire, UK
[12]Howard Hughes Medical Institute, Department of Immunobiology, Yale University School of Medicine, Yale, New Haven, CT, USA
[13]Institute of Molecular Biotechnology of the Austrian Academy of Sciences (IMBA), Vienna BioCenter (VBC), Vienna, Austria
[14]Present address: Department of Genetics, Silberman Institute of Life Sciences; and Department of Immunology and Cancer Research, Faculty of Medicine, The Hebrew University of Jerusalem, Jerusalem, Israel
[15]These authors contributed equally
[16]Lead contact
*Correspondence: uri.alon@weizmann.ac.il (U.A.), eldad.tzahor@weizmann.ac.il (E.T.)

## SUMMARY

Fibrosis remains a major unmet medical need. Simplifying principles are needed to better understand fibrosis and to yield new therapeutic approaches. Fibrosis is driven by myofibroblasts that interact with macrophages. A mathematical cell-circuit model predicts two types of fibrosis: hot fibrosis driven by macrophages and myofibroblasts and cold fibrosis driven by myofibroblasts alone. Testing these concepts in cardiac fibrosis resulting from myocardial infarction (MI) and heart failure (HF), we revealed that acute MI leads to cold fibrosis whereas chronic injury (HF) leads to hot fibrosis. MI-driven cold fibrosis is conserved in pigs and humans. We computationally identified a vulnerability of cold fibrosis: the myofibroblast autocrine growth factor loop. Inhibiting this loop by targeting TIMP1 with neutralizing antibodies reduced myofibroblast proliferation and fibrosis post-MI in mice. Our study demonstrates the utility of the concepts of hot and cold fibrosis and the feasibility of a circuit-to-target approach to pinpoint a treatment strategy that reduces fibrosis. A record of this paper's transparent peer review process is included in the supplemental information.

## INTRODUCTION

Fibrosis is a pathology in which excessive scarring replaces healthy tissue.[1–3] Fibrosis leads to progressive organ failure in heart,[4] kidney,[5] liver,[6] muscle,[7] lung,[8,9] and other tissues. In the heart, fibrosis is included in many pathologies, such as myocardial infarction (MI) and in heart failure (HF), two major

causes of mortality.[10] Preventing or reducing fibrosis is a major unmet medical need.[1,2]

Fibrosis is a complex biological process due to the interaction of multiple cell types and signaling molecules.[11,12] The structural component of the pathology, the scar, is composed of extracellular matrix (ECM) proteins deposited mostly by activated fibroblasts, called myofibroblasts.[13] Myofibroblasts interact with

other cell types including monocyte-derived macrophages,[14,15] which are recruited in large numbers into the site of injury and support the fibrotic process.[12,14,16–18]

Recently, we developed a mathematical model of fibrosis, focusing on a circuit of growth factor exchange between myofibroblasts and macrophages.[3,19,20] The model suggests three possible outcomes—healing, hot fibrosis, or cold fibrosis. Hot fibrosis is characterized by high number of both myofibroblasts and macrophages which support each other's growth. Cold fibrosis, in contrast, consists of abundant myofibroblasts without activated macrophages. Healing is accompanied by the collapse of the two cell populations and return to baseline (i.e., tissue homeostasis). The model predicts that myofibroblast growth is driven by an autocrine growth factor signaling loop[19] and that suitable modulation of the growth factor milieu can provide transitions between healing and fibrotic states. Hot and cold fibrosis have been recently found in histological analysis of transplanted human kidneys.[21] However, the relevance of hot and cold fibrosis in the heart has not been tested.

We sought to test these concepts and the ability to therapeutically manipulate them in distinct *in vivo* cardiac diseases—MI-induced fibrosis and fibrosis caused by ventricular pressure overload (induced by transverse aortic constriction; TAC). MI occurs when a blood vessel that feeds the heart is blocked, causing ischemia and cardiomyocyte cell death.[22,23] Subsequently, immune cells rapidly invade the injured zone leading to the local expansion and activation of myofibroblasts. This results in changes in the microenvironment dictated by the release of various cytokines, growth factors, and matrix proteins, as described mathematically by Lindsey and colleagues.[18,24,25] The injured non-regenerative myocardium is replaced by a fibrotic scar over a period of several weeks, a process called cardiac remodeling, that gradually leads to reduced heart function, morbidity, and mortality.[26–28] In TAC, increased ventricular pressure occurs when the aortic arch is constricted, leading to increased workload and compensatory hypertrophy. Chronic pressure overload eventually transitions to cardiac dysfunction along with extensive fibrosis throughout the ventricle.[29]

Cardiac fibrosis is an inevitable outcome in adult mammals following diverse cardiac injuries but is absent in neonatal mice and pigs due to the regenerative potential of the young heart.[30–34] Recent advances in animal models of cardiac injuries suggest that fibrosis can be reduced by certain interventions[35–44]—changing a long-held notion in the field that fibrosis (or scarring) is irreversible. These interventions have yet to be translated to the clinic.

Here, we develop a double-reporter mouse and used spatial histological analyses, bulk, single-cell, and spatial transcriptomics in mice, pigs, and humans to show that fibrosis after MI is a dynamic process leading to cold fibrosis, dominated by myofibroblasts without activated macrophages. In contrast, chronic insults such as TAC lead to hot fibrosis with coexisting macrophages and myofibroblasts. We used the mathematical circuit model to identify a target that drives cold fibrosis—the myofibroblast autocrine signaling loop. We then identified the growth factors that contribute to this autocrine loop using transcriptomic analysis coupled with *in vitro* and *in vivo* experiments. We show that inhibiting one of these factors, the (emerging) cytokine TIMP1,[45] by means of neutralizing antibodies reduces myofibroblasts prolifer-

ation and fibrosis following MI in mice. This study provides physiological support for the concept of cold and hot fibrosis, indicating that acute injury leads to cold fibrosis and chronic injury to hot fibrosis, and demonstrates the feasibility of a circuit-to-target approach to reduce cold fibrosis after acute cardiac injury by inhibiting a myofibroblast autocrine growth factor-TIMP1.

## RESULTS

### Acute MI results in cold fibrosis

Our starting point for understanding cardiac fibrosis is a quantitative theory of inflammation-dependent fibrosis[19] (Figure 1A). This theory is based on a dynamical model for the growth and interaction of myofibroblasts and macrophages based on *in vitro* co-culture experiments.[20] The state of the tissue is described in a phase portrait whose axes are the abundance of myofibroblasts and macrophages (Figure 1A). Arrows mark the dynamics of the cells over time. The model shows three possible outcomes for injury. All three start by macrophage infiltration, accompanied by fibroblast activation and proliferation leading to an increase in both cell types.

In healing (i.e., the regenerative setting), both populations eventually shrink back to pre-injury levels, providing a trajectory that rises and then falls in the phase portrait (Figure 1A, blue line).

The other two outcomes are fibrotic states. In hot fibrosis, myofibroblasts and macrophages support each other's growth by a reciprocal exchange of growth factors, providing an upward trajectory with high numbers of both myofibroblasts and macrophages (Figure 1A). In cold fibrosis, both macrophages and myofibroblasts initially rise, but then macrophage numbers return to baseline while myofibroblasts persist. Cold fibrosis is thus a state of abundant myofibroblasts that support their own growth, with a low number of macrophages similar to the pre-injury state (Figure 1A).

Whether hot or cold fibrosis occurs depends on the tissue parameters in the model and the type of injury. For example, strong paracrine signaling support between macrophages and myofibroblasts leads to hot fibrosis, whereas a weaker paracrine support is predicted to result in cold fibrosis.[19] We hypothesize that the pathology and therapeutic options for fibrosis should differ in a setting of hot or cold fibrosis.

To explore hot and cold fibrosis in clinically relevant settings, we began with acute MI induced by permanent ligation of the left anterior descending artery (LAD) in mice, a classical experiment in which fibrosis gradually develops after injury.[28]

We first explored the dynamics of macrophage and myofibroblast populations in mice after MI. To do so, we analyzed the single-cell RNA sequencing (scRNA-seq) data by Forte et al.[46] and quantified the fold change (FC) ratio shift in abundance of macrophages and myofibroblasts at seven time points. Macrophage and myofibroblast abundance first increased, peaking at days 3–7. At day 14, macrophages decreased leaving persistent myofibroblasts at days 14 and 28 (Figures 1B, S1A, and S1B).[50] At 28 days post-MI, the scar is mature, hence, this trajectory is consistent with cold fibrosis. Similar dynamics were observed by Jin et al.[25]

To directly quantify myofibroblasts and macrophages and their spatial arrangement *in situ*, we developed a macrophage-myofibroblast double-reporter mouse line, "MAMY," in which cardiac

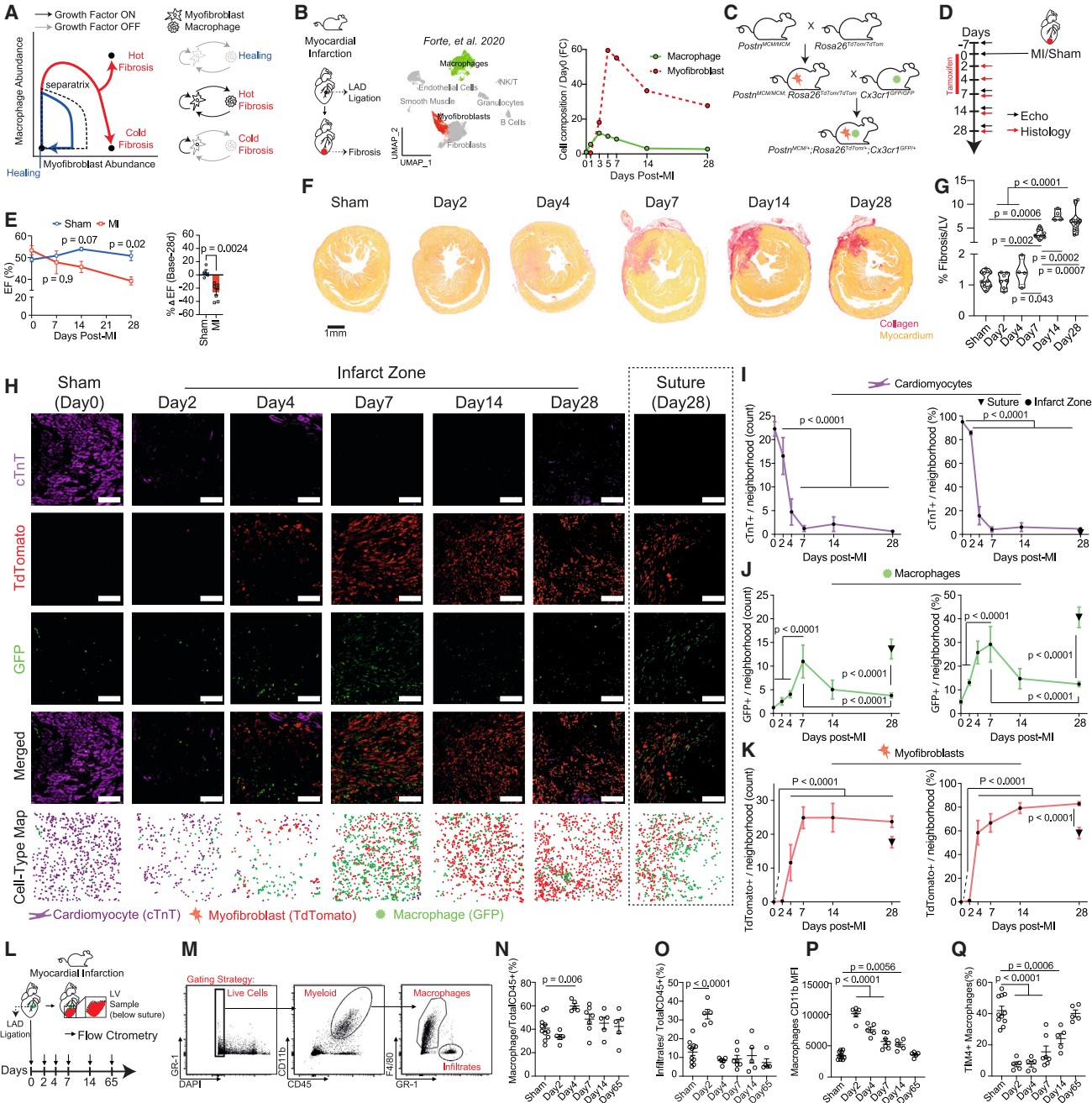

**Figure 1. Acute myocardial infarction results in cold fibrosis**

(A) Mathematical model of the myofibroblast-macrophage cell circuit predicts three possible outcomes: hot fibrosis, cold fibrosis, and healing based on the abundance of these two populations (axes).[19] The basin of attraction for the healing state is bounded by a separatrix. The scheme also denotes the growth factor signaling as either ON (black lines) or OFF (gray lines) at each specific fixed point.

(B) Single-cell RNA sequencing (scRNA-seq) dynamic analysis of total macrophage and myofibroblast populations (% of total interstitial cells) in an adult mouse heart following-MI data obtained from Forte et al.[46] Data are represented as cell composition fold change (FC) to day 0.

These data are further described in Figures S1A and S1B. UMAP of clusters highlighting macrophages (green, full line) and myofibroblasts (red, striated line). Cell-type clusters also include smooth muscle cells, fibroblasts, endothelial cells, granulocytes, NK/T cells, and B cells.

(C) Breeding strategy used to create the macrophage-myofibroblast double-reporter (MAMY) mice. First, we crossed Postn$^{MCM/MCM}$ mice with Rosa26$^{tdTomato/tdTomato}$ mice to obtain an established tamoxifen-inducible cardiac myofibroblasts lineage-tracer[47,48] (STAR Methods). We then crossed these mice with a global monocyte/macrophage reporter-Cx3cr1$^{GFP/GFP}$.[47] Triple-heterozygote mice were used as experimental animals (Postn$^{MCM/+}$ Rosa26$^{TdTomto/+}$Cx3cr1$^{GFP/+}$).

myofibroblasts are labeled with tdTomato and macrophages with GFP. The MAMY mouse is based on the extensively validated global monocyte-macrophage reporter: Cx3cr1$^{GFP/GFP}$ [47,51,52], and a tamoxifen-inducible Cre-Lox system, Postn$^{MCM/MCM}$ [51,52] crossed with Rosa26$^{TdTomto/TdTomto}$ [48,53] mice, a lineage reporter that robustly labels cardiac myofibroblasts [48,50] (Figure 1C, STAR Methods; Note S1). An important aspect of this mouse line is the assessment of macrophage numbers using the GFP reporter, reflecting hot (high) and cold (low) fibrosis states.

We performed MI in adult MAMY mice (Postn$^{MCM/+}$ Rosa26$^{TdTomto/+}$Cx3cr1$^{GFP/+}$) by permanently ligating the LAD. We next administered tamoxifen daily by intraperitoneal (IP) injections for 7 days and sampled sham and MI operated hearts at days 2, 4, 7, 14, and 28 post-MI (Figure 1D, STAR Methods). MI resulted in reduced ejection fraction (EF) (Figure 1E) and a gradual increase in the size of replacement fibrosis assayed by sirius red staining [54] (Figures 1F and 1G).

To quantify myofibroblasts, macrophages, and cardiomyocytes, we analyzed heart cross sections with fluorescent microscopy from the MI infarct zone—an area of massive myocardium loss and permanent scar, and from healthy myocardial tissues of sham-operated hearts (referred as time point 0).

We counted tdTomato and GFP cells, as well as cardiomyocytes marked by cardiac troponin T (cTnT) staining, within a 100-μm neighborhood around each cell (Figures 1H and S1C). In the infarct zone, cardiomyocyte number was markedly reduced following injury and remained low in the mature scar (Figures 1H and 1I). Macrophage counts increased, peaking at day 7, and then significantly declined by day 28 (Figures 1H and 1J). Myofibroblasts numbers increased after injury, peaking

at day 7, and remained elevated through day 28. This pattern is consistent with cold fibrosis.

The MAMY mouse allowed us to view the spatial organization of the cells. We noticed that in the permanent LAD ligation experiments, around the suture, there was a distinct spatial organization pattern of myofibroblasts and macrophages (Figures 1H–1K and S1C). This resembles a tissue foreign body response (FBR) characterized by fibrosis accumulating around a foreign body such as a surgical suture.[55,56] We, therefore, quantified cells in the suture region at day 28 and identified that abundant macrophages (GFP cells) were arrayed around the suture, along with abundant myofibroblasts (tdTomato cells). In contrast, macrophages were markedly reduced at distances beyond 100 μm from the suture (infarct zone). We conclude that the local response to a foreign body in the mature suture region (28 days) is consistent with hot fibrosis.[57,58]

We repeated the MI experiment in Hsd:ICR (CD1) mice, with an extended 65-day endpoint. Since these mice do not contain a double-reporter system, we quantified cells using flow cytometry. We harvested cells from the injured left ventricle, below the suture, at different time points, and analyzed them using cardiac macrophage markers specific to both recruited and resident cells (Figures 1L and 1M). The macrophage fraction relative to all immune cells (CD45$^+$) increased and then returned to baseline levels, as did the fraction of infiltrating monocytes and neutrophils (GR1$^+$) (Figures 1N and 1O) and the fraction of activated macrophages (Figure 1P). In contrast, TIM4$^+$ tissue resident macrophages, which are required for cardiac regeneration and repair [59–61] were depleted rapidly but were gradually restored in proportions to baseline levels 65 days post-MI (Figure 1Q). The

---

(D) Experimental scheme whereby MAMY mice (Postn$^{MCM/+}$Rosa26$^{TdTomto/+}$Cx3cr1$^{GFP/+}$) underwent sham or MI by permanent left anterior descending artery (LAD) ligation. To label myofibroblasts with tdTomato, mice were given daily tamoxifen injections following MI (red line). Cardiac function was tracked using 2D echocardiography at multiple time points following MI: 7 days prior to MI (baseline), and 7, 14, and 28 days post-MI (black arrows). Hearts were collected for histological fibrosis and immunofluorescence (IF) analysis on days: 2, 4, 7, 14, and 28 post-MI/sham (red arrows).

(E) Ejection fraction (EF; %) analysis was performed on MI/Sham operated MAMY mice (Sham, $n = 4$; MI, $n = 7$). To extract the direct % change in EF between days 28 post-MI and baseline, we calculated the ΔEF value per animal (STAR Methods). Red, MI; blue, sham. Results are represented as mean ± SEM. Statistical analysis performed using two-way ANOVA with Sidak's adjusted $p$ values (left) or two-tailed unpaired t test (right). Individual points represent individual biological replicates.

(F) Representative histology images of MAMY hearts stained with sirius red on days 2, 4, 7, 14, and 28 post-MI/sham. Scale bar, 1 mm. Collagen, red; healthy myocardium, yellow.

(G) MAMY mice fibrosis area/left ventricle (LV) in % was calculated per timepoint: Sham ($n = 12$), day 2 ($n = 7$), day 4 ($n = 4$), day 7 ($n = 9$), day 14 ($n = 4$), and 28 ($n = 13$) post-MI. Mix-max violin plot represent the median (middle full line) and quartiles (striated lines). Statistical analysis performed using one way ANOVA with Tukey's multiple comparisons test.

(H–K) Quantification of spatiotemporal distribution of cardiac-troponin-T cells (cardiomyocytes, cTnT+; purple), the lineage-traced (tdTomato+; red) myofibroblasts, and macrophages (GFP+, green) in Sham hearts and infarct zone of MI operated hearts. cTnT+, tdTomato+, and GFP+ cells were also measured at the suture area of 28 days post-MI hearts (STAR Methods, Figure S1C).

(H) Representative immunofluorescence (IF) images of Sham ($n = 5$; 1,487 ± 118.5 SEM cells per replicate) and MI operated hearts on days 2 ($n = 4$; 1,088 ± 341.8 SEM cells per replicate), 4 ($n = 4$; 1,027 ± 352.6 SEM cells per replicate), 7 ($n = 4$; 1,792 ± 153.3 SEM cells per replicate), 14 ($n = 3$; an average of 1,471 ± 275.2 SEM cells per replicate), and 28 ($n = 5$; 1,627 ± 248.7 and 1,049 ± 315.9 SEM cells per replicate for infarct zone and suture, respectively) post-MI. Images are represented as either single channel for GFP, tdTomato, and cTnT, as a merged IF image and as a cell-type map that illustrates the identified cells as representative points. White scale bar, 100 μm. cTnT+ cells (I), GFP+ cells (J), and tdTomato+ cells (K) were quantified as count per 100 μm neighborhood or as % of cells per 100 μm neighborhood (STAR Methods, Figure S1C). Results are represented as mean ± SEM. In (I)–(K), infarct zone is denoted as a full circle, day 28 suture is denoted as an upside-down triangle. Statistical analysis performed using the Mann-Whitney test with Bonferroni adjusted $p$ values (I–K).

(L) Experimental scheme whereby adult Hsd:ICR (CD1) mice hearts underwent MI by permanent LAD ligation and further processed by flow cytometry (STAR Methods). LV samples below the suture of MI or sham operated mice were collected at 6 different time points following injury at days: 0/Sham ($n = 11$), 2 ($n = 5$), 4 ($n = 5$), 7 ($n = 7$), 14 ($n = 5$), and 65 ($n = 5$).

(M–Q) (M) Representative flow cytometry plots of the gating scheme used to identify (N) total cardiac macrophages (out of total immune cells- CD45$^+$), (O) infiltrating cells (monocytes and neutrophils out of total immune cells-CD45$^+$), (P) cardiac activated macrophages, measured by CD11b mean fluorescence intensity (MFI),[49] and (Q) Resident macrophages (TIM4$^+$ macrophages out of total macrophages). Data presented as mean ± SEM. Statistical analysis performed using one way ANOVA with Dunnett's multiple comparisons procedure.

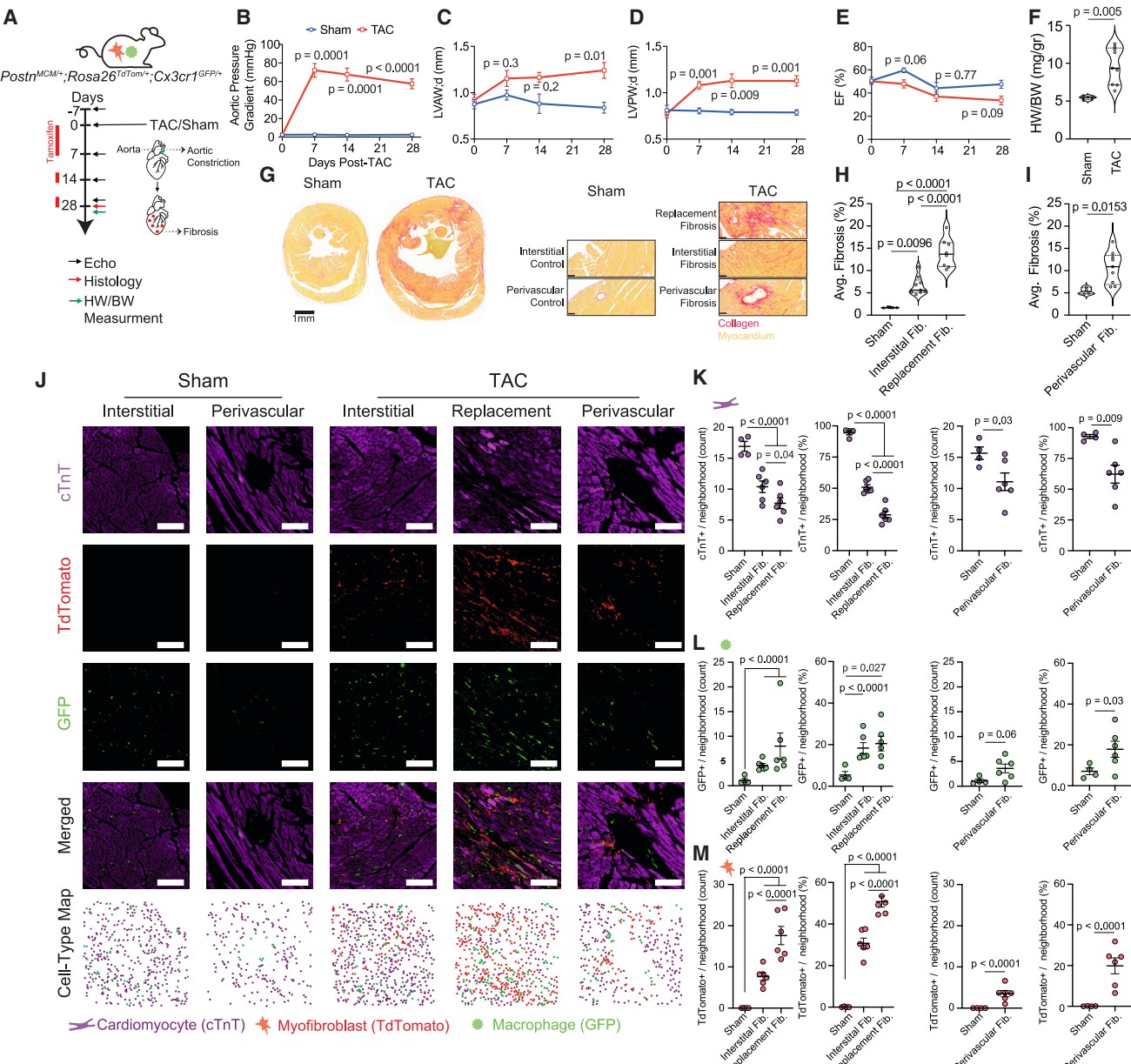

**Figure 2. Chronic ventricular pressure overload results in hot fibrosis**

(A) Experimental scheme whereby MAMY mice (Postn^MCM/+^Rosa26^TdTomto/+^Cx3cr1^GFP/+^) underwent sham or transverse aortic constriction (TAC) to induce chronic pressure overload. To label myofibroblasts with tdTomato, mice were given 7 daily tamoxifen injections immediately following TAC, and on days 13–14 and 27–28 post-TAC (red line). Cardiac function was tracked using 2D echocardiography at baseline (−7) and on multiple time points following TAC: 7, 14, and 28 days (black arrows). Hearts were collected for histological fibrosis and immunofluorescence (IF) analysis 28 post-MI/sham (red arrows). Heart weight (HW) to body weight (BW) ratios were further analyzed 28 days following TAC (green arrow).

(B–E) Temporal echocardiography measurements of MAMY mice following TAC/sham, including (B) aortic pressure gradient (mmHg), (C) left ventricle anterior wall diameter in diastole (LVAW;d, in mm), (D) left ventricle posterior wall diameter in diastole (LVPW;d, in mm), and (E) ejection fraction (EF; %). Results are represented as mean ± SEM. Statistical analysis performed using two-way ANOVA with Sidak's adjusted p values.

(F) HW/BW ratios (mg/gr) of MAMY mice sham (n = 4) and TAC (n = 9) hearts 28 days post-TAC. Mix-max violin plot represent the median (middle full line) and quartiles (striated lines). Statistical analysis performed using two-tailed unpaired t test.

(G) Left: representative images of MAMY hearts sirius red stained sections, 28 days post-TAC/sham. Scale bar, 1 mm. Right: fibrosis in TAC hearts was quantified separately based on its histological presentation: interstitial, replacement and perivascular fibrosis. Scale bar, 100 μm. Collagen, red; healthy myocardium, yellow.

(H) MAMY mice TAC interstitial (n = 9) and replacement (n = 8) fibrosis (fib) field of views (FOVs) were compared with healthy sham (n = 4) myocardium (data presented as average %/FOV). Mix-max violin plot represent the median (middle full line) and quartiles (striated lines). Statistical analysis performed using one way ANOVA with Tukey's multiple comparisons test.

cardiac macrophage population thus returns to baseline in the aftermath of MI, consistent with a cold fibrosis trajectory.

### Chronic ventricular pressure overload results in hot fibrosis

We next studied HF resulting from ventricular pressure overload. We performed permanent TAC in adult MAMY mice and tracked them for 28 days (Figure 2A). TAC generated a persistent increase in aortic pressure gradient that leads to an enlarged left ventricle with increased heart weight to body weight (HW/BW) ratios, and a marginal decrease in EF (Figures 2B–2F). TAC hearts showed extensive fibrosis indicated by sirius red staining (Figures 2G–2I). We identified three distinct patterns of sirius red staining, which correspond to replacement fibrosis—patched fibrosis with no cardiomyocytes, interstitial fibrosis—fibrosis between cardiomyocytes, and perivascular fibrosis surrounding blood vessels[54] (Figures 2G–2I).

Cardiomyocyte numbers were reduced in all types of fibrosis (Figures 2J and 2K), whereas myofibroblast numbers were elevated in all fibrotic states and were positively correlated with fibrosis severity- replacement fibrosis showed higher counts compared with interstitial fibrosis (Figures 2J–2M). Macrophages were elevated in both interstitial and replacement fibrosis regions to a similar extent (Figures 2J and 2L). Myofibroblast and macrophage numbers were also elevated in TAC perivascular fibrosis regions (macrophages showed only a trend). We conclude that interstitial and replacement fibrosis following TAC injury are consistent with hot fibrosis.

Figure S2A shows the myofibroblast to macrophage ratios in all fibrosis cases studied. MI (cold fibrosis) shows high myofibroblast content over macrophage (6:1 ratio), whereas in TAC and FBR (hot fibrosis) we calculated a cell ratio close to 1:1 myofibroblasts to macrophages. Taken together, we conclude that while acute MI in mice leads to cold fibrosis, chronic injuries such as TAC and a FBR show hot fibrosis.

### Cold fibrosis following MI is conserved in humans and pigs

We focused on cold fibrosis after MI and asked whether it is conserved from mice to humans. To do so, we analyzed human Visium spatial transcriptomic data by Kuppe et al.[62] These data contain samples of uninjured human hearts and from MI hearts at early (days 0–15 post-MI) and late (30+ days post-MI) time points following injury (Figure 3A). We performed deconvolution analysis on each spatial Visium spot to determine the abundance of fibroblasts, myofibroblasts, and myeloid cells (see STAR Methods). Myeloid cells increased in the first 2 weeks and returned to baseline 30 days after MI. In contrast, both fibroblasts

and myofibroblasts increased and remained high in the late time-frame post-MI (Figure 3B). This pattern is consistent with cold fibrosis.

We next investigated acute MI in a clinically relevant porcine model. Pigs underwent a 60 min ischemia-reperfusion injury by an inflation of a balloon within the LAD coronary artery.[63] Following reperfusion, pigs were treated, in an antegrade intracoronary trajectory, with saline solution (control) or with human recombinant Agrin protein (rhAgrin), a regenerative ECM protein that promotes cardiac repair in mice and pigs.[35,64] We obtained heart samples from the remote and infarct zones at days 3 and 28 following MI and performed bulk-mRNA sequencing and histological scar analysis (Figure 3C).

As expected, hearts treated with saline (control) showed extensive sirius red staining indicating severe fibrosis at day 28, whereas hearts treated with rhAgrin had less fibrosis, consistent with our previous findings[64] (Figures 3D, 3E, S3A, and S3B). Unlike saline-treated hearts, rhAgrin-treated infarct zone samples showed a similar gene expression profile to their paired healthy remote zones at day 28 (Figures 3F–3H and S3D–S3J; Tables S1, S2, S3, and S4).

Next, we quantified myofibroblast and macrophage abundance at both time points (Figures 3I and 3J) using deconvolution (STAR Methods, Figure S3C; Table S5). Saline-treated pig hearts had increased macrophage abundance at day 3 which declined by day 28 in the infarct region, whereas myofibroblasts abundance remained high at 28 days, indicative of cold fibrosis. In contrast, rhAgrin-treated hearts showed a healing trajectory with increased myofibroblast and macrophage abundance at day 3, which returned to baseline levels at day 28, similar to the healing loop predicted by the mathematical model (Figure 1A). We conclude that acute MI leads to cold fibrosis in humans and pigs and that cold fibrosis can be prevented under appropriate interventions (as demonstrated with rhAgrin treatment).

### Macrophages return to a homeostatic state, whereas fibroblasts acquire a pro-fibrotic phenotype in cold fibrosis

To further explore the cell states in cold fibrosis development, we used scRNA-seq data by Forte et al.[46] from cardiac interstitial cells following MI (Figures 1B, S1A, and S1B). We analyzed the gene expression of fibroblasts (including myofibroblasts) and macrophages at day 0 (i.e., pre-injury), day 3 and day 28 at the peak of fibrosis. Each cell population is a continuum of gene expression, rather than discrete clusters. We identified the main functions of each cell type using Pareto task inference (ParTI)[65] (STAR Methods). ParTI analyzes a continuum of gene expression according to specialist and generalist cells, where

---

(I) MAMY mice TAC perivascular fibrosis ($n = 9$) was compared with sham perivascular regions ($n = 4$). Average fibrosis area/FOV (%) was calculated. Mix-max violin plot represent the median (middle full line) and quartiles (straited lines). Statistical analysis performed using two-tailed unpaired t test.

(J–M) MAMY mice neighborhood quantification of cardiac-troponin-T cells (cardiomyocytes, cTnT+; purple), the lineage-traced (tdTomato+, red) myofibroblasts, and macrophages (GFP+, green) in TAC/Sham hearts (STAR Methods, similar analysis demonstrated for MI hearts in Figure S1C). (J) Representative IF images of interstitial and perivascular sham ($n = 4$; 1,833 ± 117.4 and 946.5 ± 71.68 SEM cells per replicate, respectively), and interstitial ($n = 6$; 1,422 ± 156.7 SEM cells per replicate), replacement ($n = 6$; 1,474 ± 286.5 SEM cells per replicate) and perivascular ($n = 6$; 963.5 ± 68.77 SEM cells per replicate) fibrosis (fib) in hearts 28 days post-TAC. Images are represented as either single channel for: GFP (green), tdTomato (red), and cTnT (purple), as a merged IF image and as a cell-type map illustrated the identified cells as representative points. White scale bar, 100 μm. cTnT+ cells (K), GFP+ cells (L), and tdTomato+ cells (M) were quantified as count per 100 μm neighborhood or as % of cells per 100 μm neighborhood (STAR Methods, Figure S1C). Results are represented as mean ± SEM. Each dot represents a single biological replicate. Statistical analysis performed using the Mann-Whitney test with Bonferroni-adjusted $p$ values when appropriate.

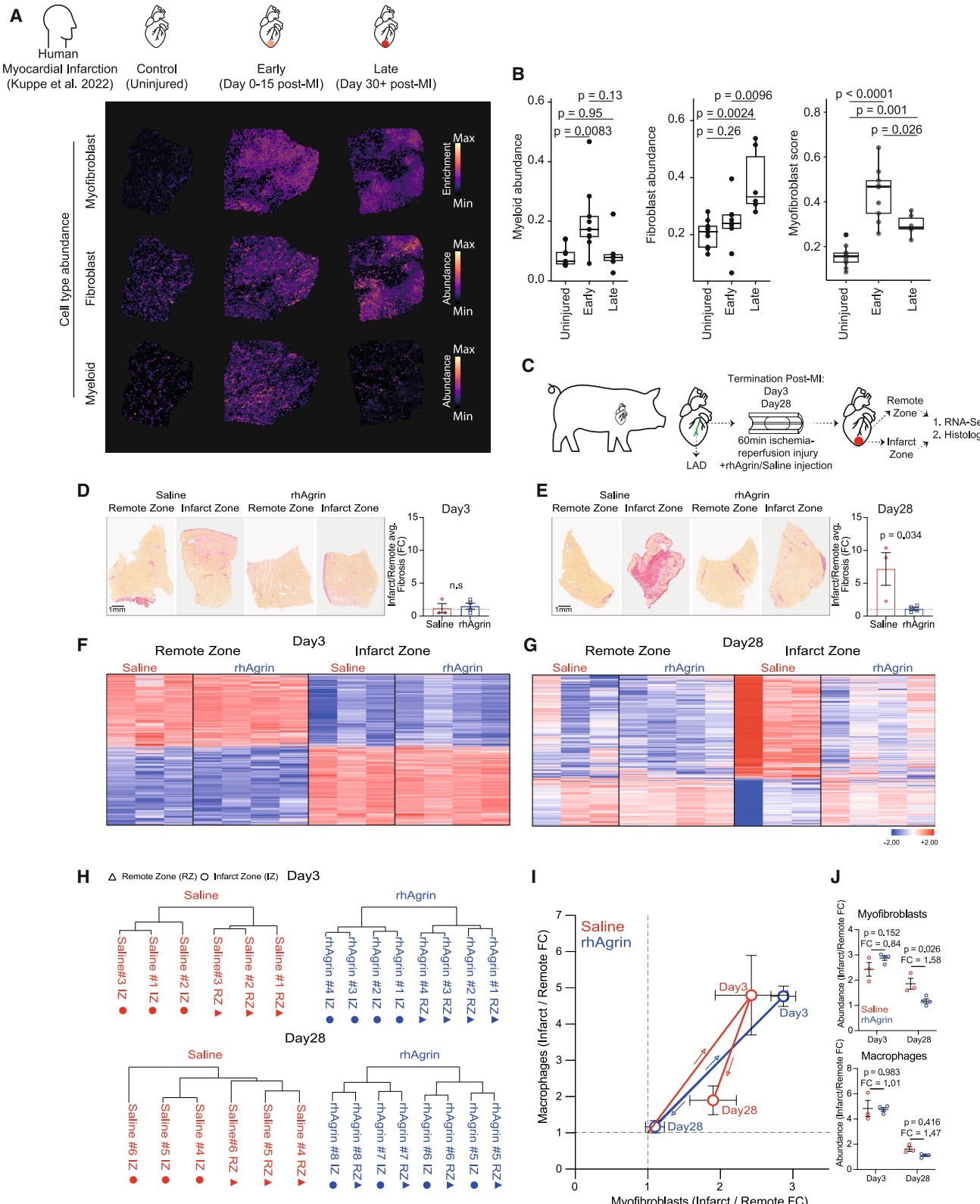

**Figure 3. Cold fibrosis after MI is conserved in humans and in a clinically relevant porcine model**

(A and B) (A) Representative human left ventricle spatial transcriptomics slides (Visium) of patients following acute-myocardial infarction (MI) and non-transplanted donor hearts. Samples were divided based on time following MI as either: uninjured (*n* = 10), early (days 0–15 post-MI; *n* = 6), and late (30 days+; *n* = 6). (B)

*(legend continued on next page)*

specialist cells are close to the vertices of a polytope (triangle, tetrahedron, etc.) that encloses the gene expression continuum. The main tasks of the population can be inferred from the gene expression of cells closest to each of the vertices.

We started by analyzing fibroblasts and macrophages 3 days post-MI, in which both cell populations rapidly grow.[50,66] Both macrophages and fibroblasts showed 5 major tasks (*p* value < 0.02 fibroblasts, *p* value = 0.001 macrophages) (Figures 4A and 4E; Video S1). Gene Ontology (GO) analysis of enriched genes near the vertices illustrated that fibroblast tasks relate to cytokine response, retinoid metabolism, angiogenesis, IL12 and IL1 signaling, and keratan sulfate metabolism (Figures 4A, 4B, S4A, and S4C; Table S6). Macrophage tasks included IL1 signaling, respiratory burst and migration, translation, phagocytosis/ECM, and a response to ischemia (Figures 4E, 4F, S4B, and S4D; Table S6). Interestingly, fibroblasts archetype #4, associated with immune-regulating functions was populated almost exclusively by day 3 cells (day 3/day 0 FC; FC > 60), suggesting the acquisition of a new fibroblast function soon after injury (Figure 4C).

Concomitant with fibroblasts, macrophages acquired new tasks within the first 3 days. Archetype #1 (day 3/day 0 FC > 90) showed recruited monocyte markers such as *Plac8*,[61] and archetype #2 (day 3/day 0 FC > 20) showed resident macrophage marker genes (i.e., *Folr2*[67]). Both archetypes were enriched with GO terms associated with angiogenesis and a phagocytic response to injury (Figures 4G and S4B).

We further used ParTI to analyze macrophages and fibroblasts in the mature scar (day 28). Macrophages and fibroblasts revealed 4 major tasks (*p* value < 0.0001 fibroblasts, *p* value = 0.02 macrophages) (Figures 4I and 4L; Video S2). The macrophage tasks relate to stress response, protein synthesis, phagocytosis, and inflammation, and the fibroblast tasks include wound response, matrix disassembly/migration, leukocyte regulation, and fibrosis/ECM deposition, as determined by GO analysis (Figures 4I, 4J, 4L–4N, and S5A–S5D; Table S6). Interestingly, fibroblasts at day 28 post-MI almost exclusively populated archetype #4: fibrosis/ECM deposition task (Figure 4I). This indicates that during MI-driven cold fibrosis, fibroblasts acquire and sustain distinct cell state changes. These changes are consistent with a previous study that highlighted the transition of cardiac myofibroblasts to matrifibrocytes within the mature scar.[50]

In contrast, macrophages were spread more equally between the 4 archetypes in a comparison between day-0 and day-28 cells (Figures 4L, S5B, and S5D). This is consistent with our MAMY mice histological data and flow cytometry analysis suggesting that macrophage identity, activity, and number all gradually return to baseline levels between days 14 and 65, when cold fibrosis is established (Figures 1H–1Q).

We find similar behavior in the pig data, where the gene signature of fibroblast archetype #4 (fibrosis/ECM deposition) was enriched in saline-treated infarct zone samples 28 days following MI (Figure S6A). We conclude that the initial transient MI period with both high macrophages and myofibroblasts is enriched in immune functions, whereas late cold fibrosis has homeostatic macrophage phenotypes and a distinct fibroblast ECM producing state, which is conserved in mice and pigs.

## The circuit model predicts that the myofibroblast autocrine signaling loop is a target for reducing cold fibrosis

We next asked which molecular interactions might be suitable targets for reducing cold fibrosis using *in silico* perturbations of our mathematical model. We simulated different interventions by changing each of the model parameters. For each changed parameter, we computed the basin of attraction to the healing state (Figure 5A). The parameter with the largest impact is the myofibroblast autocrine growth factor loop.[19] According to the model, inhibition of the myofibroblast autocrine loop significantly expands the healing margins. Notably, at a threshold level of about 40% inhibition of autocrine growth factor production, the cold fibrosis fixed-point vanishes, and the only remaining steady state possible is the healing state, where both myofibroblasts and activated macrophages return to baseline (Figure 5B). Thus, our model indicates that the myofibroblast autocrine growth factor loop is a target for reducing cold fibrosis. It further suggests that moderate (40%) inhibition of this loop can be beneficial.

---

Abundance of fibroblasts and myeloid cells was quantified based on deconvolution scores of cell types per spot (STAR Methods). Myofibroblast were calculated as enrichment of the mean myofibroblast state score within spots with a minimal 10% value of cell-type abundance (STAR Methods). Statistical analysis used Wilcoxon tests with Benjamini-Hochberg adjusted *p* values.

(C) Experimental design of pig MI experiment: adult (3 months old) pigs underwent MI by temporarily occluding their LAD using a balloon catheter (STAR Methods). Following reperfusion (balloon deflation), pigs were immediately treated with recombinant human Agrin (rhAgrin) or Saline control, in an antegrade trajectory. Injured pig hearts were collected at either day 3 (*n* = 4 for rhAgrin; *n* = 3 for Saline) or day 28 (*n* = 4 for rhAgrin; *n* = 3 for saline) and dissected to distinct tissue areas (infarct and remote zones). Remote and infarcted samples were subjected to bulk-mRNA sequencing and histology for fibrosis assessment.

(D and E) Representative sirius red staining images are shown from (D) day 3 and (E) day 28 post-MI. Fibrosis was quantified as the average % fold change (FC) between infarct/remote zone sections for each pig individually. Fibrosis (day 3 or day 28) for Saline and rhAgrin hearts was measured using two-tailed unpaired t test. Scale bars: 1 mm. striated line denotes FC = 1. n.s, non-significant difference.

(F and G) Heatmaps based on log₂ transformed normalized counts for all (upregulated and downregulated) differentially expressed genes (defined by |log₂fold change| ≥ 1, *p*-adjusted value < 0.05 and max raw counts > 30) between remote and infarct zones for Saline and rhAgrin-treated hearts at day 3 (5,961 genes) (F) and 28 (1,079 genes) (G) post-MI. Rows represent genes and columns represent each biological sample and its spatial distribution according to infarct or remote zone. Data are represented as mean ± SD.

(H) Hierarchical clustering per condition (day [3 or 28] +treatment [rhAgrin or Saline]), based on the 1,000 most variable genes. Triangles and circles represent remote and infarct zones, respectively.

(I) Deconvolution of bulk-mRNA sequencing of pig hearts following MI of either rhAgrin (blue) or Saline-treated (red) samples. Macrophage and myofibroblast abundances were assessed by gene signatures as FC, between infarct and remote zones (STAR Methods, Figure S3C, and Table S5).

(J) Macrophage and myofibroblast abundances, based on deconvolution of bulk-mRNA sequencing (as in I). Treated samples were compared per time point (day 3 or 28), separately by two-tailed unpaired Student t test. Results are represented as mean ± SEM.

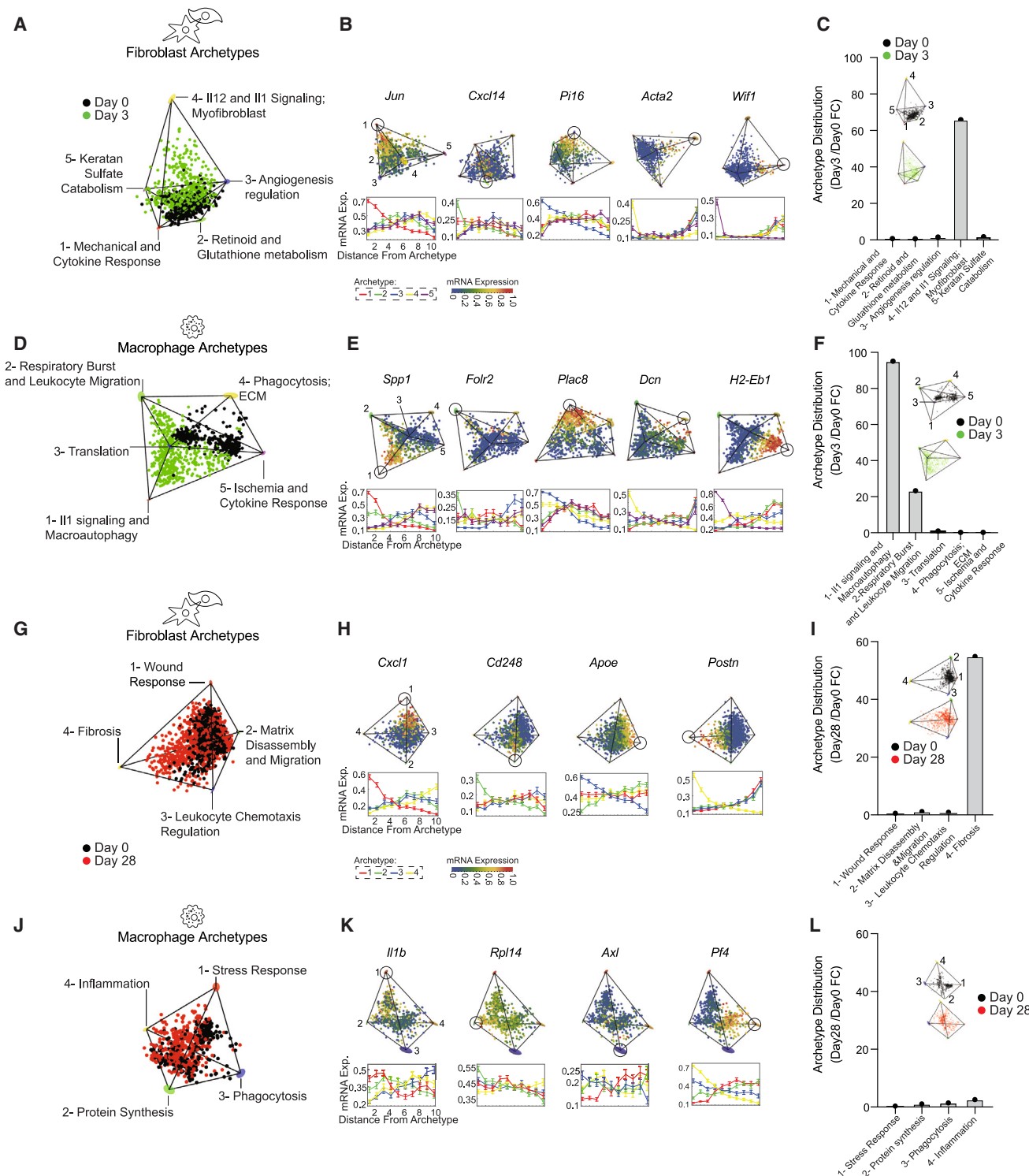

**Figure 4. In cold fibrosis, macrophages return to homeostatic functions whereas fibroblasts acquire a persistent fibrotic cell state**

(A) Pareto analysis was used to characterize the continuum of gene expression within each cell type as distributed between specialist and generalist cells. Pareto analysis was done using ParTI.[65] Fibroblasts (including myofibroblasts) from day 0 (black) and day 3 (green) following MI fill in a 5-vertex polyhedron in gene expression space (p value < 0.02). The cells are projected on the first three principal components (Video S1).

(B) Gene expression of the top enriched genes for the day 0 + 3 post-MI fibroblast archetypes. Upper: cells within the polyhedron are colored by the normalized mRNA expression level of each gene (on a scale of 0–1). Lower: mRNA expression level plotted as a function of the euclidean distance from each archetype (in gene expression space) where cells were binned into 10 bins and the mean expression across the bins was computed.

*(legend continued on next page)*

## TIMP1 promotes cardiac myofibroblast proliferation

We sought to characterize the growth factors involved in the myofibroblast autocrine signaling loop. To do so, we used NicheNet, an algorithm that infers ligand-receptor interactions within a transcriptomic dataset, using prior knowledge on ligand-receptor interactions and regulatory networks.[68] We applied NicheNet to the scRNA-seq dataset by Forte et al.[46] (Figures 5C–5F and S7E). To focus on growth factor interactions, we retained ligands according to a curated list of growth factors (see STAR Methods). This analysis detected paracrine and autocrine growth factor interactions of macrophages (M), fibroblasts (F), and myofibroblasts (mF) (Figures 5C and 5D). Among the growth factor interactions between the cell populations, the strongest is the autocrine loop of myofibroblasts (Figures 5D–5F). The myofibroblast autocrine interaction increased between days 3 and 28, whereas all other interactions involving macrophages decreased, as expected in a cold fibrosis outcome (Figure 5F).

To establish whether each interaction is specific to cold fibrosis development or maintenance, we investigated whether the interaction is differentially expressed between day 0 and day 3 or 28 post-MI (Figures 5E and 5F). We found known factors that play important roles in cardiac fibrosis, such as transforming growth factor beta (*Tgfb1*, *Tgfb2*, and *Tgfb3*)[69,70] and Oncostatin-M (*Osm*).[71] We also identified tissue inhibitor of metalloproteinase 1 (*Timp1*) and thrombospondin 4 (*Thbs4*), which were not previously characterized as cardiac myofibroblast growth factors.

We focused on *Timp1* since it was strongly expressed at both early (day 3) and late phases of cold fibrosis (day 28), consistent with its increased levels in serum samples following MI in mice.[24] TIMP1 has been studied in relation to fibrosis mostly in the context of its inhibition of matrix metalloproteinases (MMPs) activity as a pro-fibrotic and an anti-fibrotic factor[24,72,73] but was also reported as a growth factor in scleroderma fibroblasts.[74]

In the heart, *Timp1* is expressed almost exclusively by myofibroblasts (Figure S7A). Following MI, its expression peaks at day 3, coinciding with the peak of myofibroblast proliferation (Pearson-*r* = 0.94, *p* = 0.015) (Figures S7B and S7C). *Timp1* expression persists for 28 days after MI in both mice and pigs (Figures S7B and S7E; Table S4). The P1 (post-natal day 1) mouse heart is well known for its robust regenerative potential.[75] *Timp1* shows a difference in expression in the regenerative (P1) versus non-regenerative hearts (P8, 8-day old mice) in response to MI, it peaked and dropped quickly in P1 hearts following MI and remained high in post-MI P8 hearts (Figure S7D). *TIMP1* expression also dropped in rhAgrin-treated pig hearts 28 days post-MI compared with control samples (Figure S7J).

We next validated the relevance of TIMP1 in human cold fibrosis following MI using LIANA+,[76] an algorithm that decodes signaling events from transcriptomics datasets. We applied LIANA+ to the human MI Visium transcriptomic data by Kuppe et al.[62] and quantified the spatial colocalization of the *TIMP1-POSTN* gene pair (Figure S7F). Following MI, the co-expression of *TIMP1* and the myofibroblast marker gene *POSTN* increased and remained high as cold fibrosis was established (30+ days following MI). This result suggests that *TIMP1* is spatially associated with myofibroblasts in human hearts following MI. We conclude that TIMP1 is linked to cold fibrosis in mice, pigs, and humans and that its reduction at later stages is associated with healing.

We analyzed the expression of potential TIMP1 receptors: *Lrp1*, *Cd63*, *App*, *Cd82*, *Cd74*, and *Cd44*[45,73,77] (Figure S7G). All interstitial cell types in the heart were found to express at least one of the known TIMP1 receptors. Myofibroblasts predominantly express three receptors: *Lrp1*, *Cd63*, and *App*. NicheNet predicted that the receptor for TIMP1 is the low-density-lipoprotein-receptor-related protein 1 (LRP1) (Figure S7H). Indeed, TIMP1 can directly interact with LRP1.[78] In myofibroblasts, *Lrp1* mRNA peaks at day 3 and persists to day 28 (Figure S7I). Similar to *Timp1*, bulk-mRNA analysis of regenerative, P1 hearts after MI in mice showed a transient peak of *Lrp1*, whereas non-regenerative P8 hearts had sustained expression of *Lrp1* up to 7 days post-MI (Figure S7J).

We next co-stained TIMP1 and LRP1 in primary cardiac myofibroblasts obtained from adult mouse hearts (Figure 5G). The majority (60%) of cardiac myofibroblasts (αSMA+ cells) co-expressed TIMP1 and LRP1. A minority (20%) expressed LRP1 alone, with very few cells expressing TIMP1 alone or neither (Figure 5H).

---

(C) Fold change (FC) of the proportion of fibroblasts from day 3 (green) and day 0 (black) that are closest to each archetype.

(D) Macrophages from day 0 (black) and day 3 (green) following MI fill in a 5-vertex polyhedron in gene expression space (*p* value < 0.001). The cells are projected on the first three principal components (Video S1).

(E) Gene expression of the top enriched genes for the day 0 + 3 post-MI macrophage archetypes. Upper: cells within the polyhedron are colored by the normalized mRNA expression level of each gene (on a scale of 0–1). Lower: mRNA expression level plotted as a function of the euclidean distance from each archetype (as in B).

(F) FC of proportion of macrophages from day 3 (green) and day 0 (black) that are closest to each archetype.

(G) Fibroblasts (including myofibroblasts) from day 0 (black) and day 28 (red) following MI fill in a 4-vertex polyhedron in gene expression space (*p* value < 0.0001). The cells are projected on the first three principal components (Video S2).

(H) Gene expression of the top enriched genes for the day 0 + 28 post-MI fibroblast archetypes. Upper: cells within the polyhedron are colored by the normalized mRNA expression level of each gene (on a scale of 0–1). Lower: mRNA expression level plotted as a function of the euclidean distance from each archetype (as in B).

(I) FC of proportion of fibroblasts from day 28 (red) and day 0 (black) that are closest to each archetype.

(J) Macrophages from day 0 (black) and day 28 (red) following MI fill in a 4-vertex polyhedron in gene expression space (*p* value < 0.02). The cells are projected on the first three principal components (Video S2).

(K) Gene expression of the top enriched genes for the day 0 + 28 post-MI macrophage archetypes. Upper: cells within the polyhedron are colored by the normalized mRNA expression level of each gene (on a scale of 0–1). Lower: mRNA expression level plotted as a function of the euclidean distance from each archetype (as in B).

(L) FC of proportion of macrophages from day 28 (red) and day 0 (black) that are closest to each archetype.

Archetypes for macrophages and fibroblasts are denoted based on color (1, red; 2, green; 3, blue; 4, yellow; 5, purple).

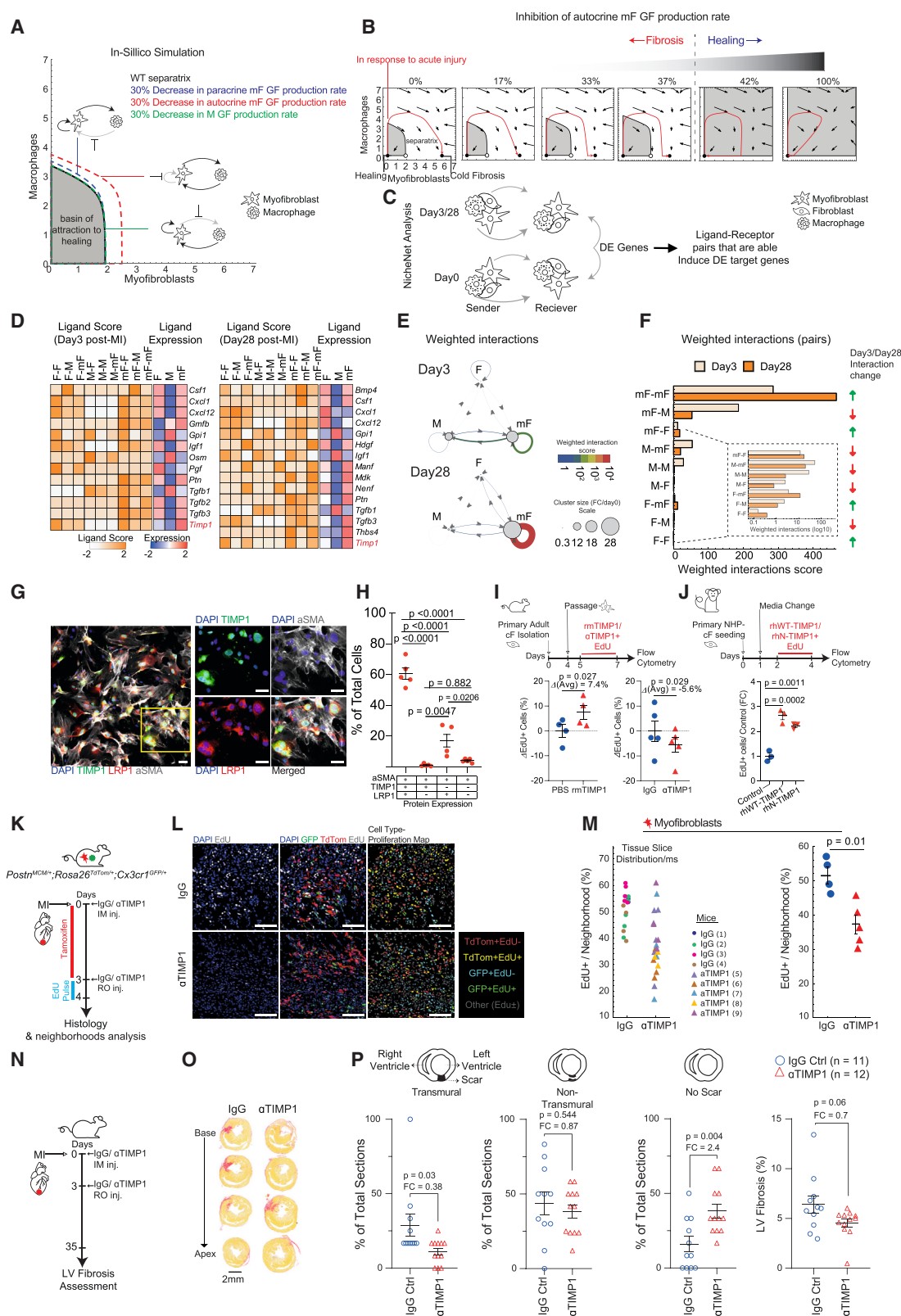

To explore the spatiotemporal expression dynamics of *Timp1* and *Lrp1* following MI, we performed Visium spatial transcriptomics on adult mouse tissue sections from days 2, 4, 7, and 14 post-MI (Figure S8A). We find three gene expression clusters by unbiased clustering (Figures S8B–S8E, STAR Methods). The spatial position of these clusters indicates that they correspond to the infarct zone (with scar components such as *Col1a1*, *Col3a1*, and *Fn1*), remote zone (with functional cardiomyocyte marker genes such as *Myh6*, and *Myl2*) and a defined border zone population between them (marked by genes such as *Nppa, Ankrd1*, and *Clu*[79–81]) (Figures S8D and S8E; Table S7). *Timp1-Lrp1* co-expressing spots were increasingly abundant in the infarct zone over time (Figure S8F).

To test the possible growth factor activity of TIMP1 *in vitro*, we used a culture assay of primary cardiac myofibroblasts which were treated with either recombinant mouse TIMP1 (rmTIMP1) or with a neutralizing antibody against TIMP1 (αTIMP1) (Figures 5I and S7K). We found that rmTIMP1 induced

an increase in 5-ethynyl-2′-deoxyuridine (EdU) incorporation in myofibroblasts, a marker of cell division, whereas inhibition of TIMP1 by the neutralizing antibodies reduced EdU incorporation (Figure 5I).

To test the mouse-to-human translational relevance of TIMP1, we repeated our culture assay using primary non-human-primate cardiac fibroblasts (NHP-cFs). NHP-cFs were incubated with recombinant human TIMP1 (rhWT-TIMP1) (Figure 5J). rhWT-TIMP1 induced a 2.5-fold increase in EdU+ NHP-cFs. To gain additional mechanistic insight, we used a fragment of TIMP1, which lacks the C-terminal domain of the protein, rhN-TIMP1.[82] We found that rhN-TIMP1 increased NHP-cFs proliferation to a similar extent as rhWT-TIMP1 excluding signaling via CD63 or APP.[77,82] Overall, this demonstrates that the growth factor activity of TIMP1 is conserved from mice to primates/humans.

To test whether TIMP1 promotes proliferation *in vivo*, we performed MI using permanent LAD ligation in adult MAMY mice

---

**Figure 5. Targeting the autocrine loop of myofibroblasts by inhibiting TIMP1 reduces fibrosis after acute MI in adult mice**

(A) Phase plot where the axes represent the number of cells. The separatrix (dashed lines) separates the basins of attraction of healing (gray area) and cold fibrosis (white). The separatrix is calculated with reference (WT) parameters (black), and with a 30% decrease in the production rates of the macrophage paracrine growth factor (green), myofibroblast paracrine growth factor (blue), and myofibroblast autocrine growth factor (red).

(B) Simulations of the cell-circuit response (red lines) to acute injury that leads to cold fibrosis with reference (WT) parameters (left). Healing is seen with a 42% or larger decrease in the myofibroblast autocrine growth factor production rate.

(C) Schematic representation of NicheNet analysis to identify ligand-receptor interactions between fibroblasts (Fs), macrophages (Ms), and myofibroblasts (mFs) at days 3 and 28 following MI. NicheNet performed on the scRNA-seq data of Figures 1B, S1A, and S1B.

(D) Heatmap of potential ligands based on F, M, and mF differentially expressed (DE) genes at days 3 (right) and 28 (left) post-MI. Ligand scores (Pearson correlation) and average ligand expression per cell type are presented as mean ± SD (by either white-orange scale or blue-red scale, respectively).

(E and F) Weighted interactions between F, M, and mF following MI (days 3 and 28) (STAR Methods). In (E), line color represents weighted interaction strength; circle size represents the fold change (FC) shift in cluster size compared with day 0. In (F), individual weighted interactions/pair of communicating cells is plotted as a bar graph for both days 3 and 28. Green arrow, increase in weighted interaction strength; red arrow, decrease in weighted interaction strength. Inset graph showing the individual weighted interactions in $\log_{10}$ scale.

(G) Representative immunofluorescence images of TIMP1 (green), LRP1 (red), alpha smooth muscle actin (αSMA, gray), and DAPI (blue) in primary adult cardiac myofibroblasts cultures. (*n* = 5 biological replicates, 2,510 ± 515 SEM cells per replicate). Scale bars, 50 μm, white frame represents the inset on the right.

(H) Quantification of images based on their TIMP1, LRP1, and αSMA protein expression (as % of total cells) presented as mean ± SEM. Statistical analysis performed using one way ANOVA with Tukey's multiple comparisons test.

(I) Top: schematic representation of cardiac myofibroblasts proliferation assay *in vitro*. Primary cardiac myofibroblasts were treated in cell culture for 48 h with either recombinant mouse TIMP1 (rmTIMP1) or anti-TIMP1 neutralizing antibodies (αTIMP1) and the appropriate control (either PBS or IgG, respectively). Cells were further incubated with 5-ethynyl-2′-deoxyuridine (EdU) to directly measure cell cycle using flow cytometry. Bottom: cardiac myofibroblast proliferation was assessed in cultures by means of delta EdU+ (%) cells of control (either PBS or IgG) and 48-h-treated cells (either rmTIMP1 or αTIMP1, respectively). In rmTIMP1 and αTIMP1 experiments, *n* = 4 and 5 biological replicates, respectively. Data are represented as mean ± SEM. Avg, average. Statistical analysis performed using two-tailed paired t test.

(J) Top: schematic representation of non-human-primate (NHP) cardiac fibroblasts (cFs) proliferation assay *in vitro*. Primary cardiac fibroblasts were treated in cell culture for 48 h with either full recombinant human TIMP1 (rhWT-TIMP1; *n* = 3 replicates) or with the human N terminus fragment of TIMP1 (rhN-TIMP1; *n* = 3 replicates) anti-TIMP1 and control (untreated cells; *n* = 3 replicates). Cells were further incubated with 5-ethynyl-2′-deoxyuridine (EdU) to directly measure cell cycle using flow cytometry. Bottom: NHP-cFs proliferation was assessed in cultures by FC of EdU+ (%) cells relative to control, respectively. Data are represented as mean ± SEM. Statistical analysis performed using one way ANOVA with Tukey's multiple comparisons test.

(K) Schematic experimental plan by which adult MAMY mice underwent MI and treated with either TIMP1 neutralizing antibodies (αTIMP1, *n* = 5) or IgG control (*n* = 4). To activate the MerCreMer system and obtain tdTomato expression, tamoxifen was injected daily following MI (STAR Methods). To label proliferating cells, mice were induced with a 24 h EdU pulse on day 3, and hearts were collected on day 4 post-MI for histological and immunofluorescence analysis.

(L) Representative immunofluorescence images of MAMY hearts treated with either αTIMP1 or IgG control 4 days post-MI and 24 h EdU pulse. Scale bar, 100 μm. GFP-green, tdTomato-red, EdU-white, and DAPI-blue. Following segmentation and annotation cells were further analyzed based on their proliferative state, EdU±: myofibroblasts (tdTomato+), macrophage (GFP+), other (tdTomato−GFP−). Images are represented either by their immunofluorescence channels or as a cell-type proliferation map illustrates the identified cells and their proliferative state as representative points.

(M) Quantification of proliferative myofibroblasts (tdTomato+EdU+) abundance (% of cells) per cellular neighborhood (100 μm in diameter) in the infarct zone of αTIMP1 or IgG-control-treated MAMY hearts (STAR Methods, Figure S9A). Left: distribution of data per tissue slice per biological replicate. Each color represents a biological replicate. Right: average tdTomato+EdU+/neighborhood per biological replicate. Results are represented as mean ± SEM. Blue circles: IgG control; red triangle: αTIMP1. Statistical analysis performed using the Mann-Whitney test.

(N) Schematic experimental plan by which adult mice underwent MI and treated with either αTIMP1 (*n* = 12 biological replicates) or IgG control (*n* = 11 biological replicates) immediately after injury onset and 3 days after injury. At 35 days left ventricular (LV) fibrosis was assessed by sirius red staining.

(O) Representative sequential sirius red staining images along the base to apex axis of αTIMP1/IgG-control-treated hearts. Scale bars: 2 mm.

(P) Fibrosis parameters presented as % of total sections that are either transmural, non-transmural, or sections with no scar (STAR Methods), and scar area out of the LV quantification. Statistical analysis performed using two-tailed unpaired t test.

and treated them with αTIMP1 or immunoglobulin G (IgG) control antibodies immediately after the injury by intramuscular (IM) injection and 3 days later by retroorbital (RO) injection (Figure 5K; STAR Methods). To quantify proliferating cells, a pulse of EdU was given to mice at day 3 post-MI, 24 h prior to heart extraction at day 4.

We enumerated tdTomato and GFP cells in each 100 μm neighborhood based on their proliferative capacity (EdU±) and generated a cell-type proliferation map (Figure S9A, STAR Methods). We find that αTIMP1-treated mice showed reduced amounts of proliferative myofibroblasts (tdTomato+EdU+) compared with IgG controls (Figures 5L and 5M). αTIMP1 antibody treatment did not affect macrophage abundance (GFP+ cells), proliferation (GFP+EdU+), or the fraction of pro-fibrotic macrophages (marked by Galectin-3[83,84]) (Figures S9B–S9D and S9F). Moreover, while αTIMP1 treatment did not significantly attenuate the fraction of αSMA+ myofibroblasts, a slight trend was observed (Figure S9E).

Based on these *in vitro* and *in vivo* assays, we conclude that TIMP1 acts as a growth factor for cardiac myofibroblasts. It is expressed by cardiac myofibroblasts after injury together with its predicted receptors forming an autocrine signaling loop driving myofibroblast proliferation.

## Inhibiting TIMP1 reduces fibrosis after acute cardiac injury

The mathematical model predicts that inhibition of the autocrine growth factor loop should reduce fibrosis (Figure 5A). To test this *in vivo*, we performed MI using permanent LAD ligation on adult Hsd:ICR (CD1) mice. We injected IgG control or αTIMP1 neutralizing antibodies directly to the injured myocardium following injury, and again 3 days after injury by RO injection (Figure 5N). We analyzed heart sections 35 days after injury (Figures 5O and 5P). Mice treated with αTIMP1 antibody showed reduced fibrosis—the number of tissue sections with detectable scarring was 2.4-fold lower in the αTIMP1 treated mice compared with mice treated with IgG control antibody. αTIMP1 treatment reduced the severity of the scar, causing a 2.5-fold reduction in the number of sections that showed a transmural scar, defined as a scar that is continuous through the entire left ventricular wall. We conclude that inhibiting TIMP1 can attenuate fibrosis following MI.

We also tested heart function at days 3, 7, and 35 post-MI. We did not find a significant difference in EF nor fractional shortening (FS) values between treated and control groups. This may be due to the low correlation between fibrosis and systolic function parameters measured by echocardiography (i.e., EF and FS) following MI in mice (Figures S9G–S9I).

## DISCUSSION

We combined mathematical modeling together with *in vitro* and *in vivo* experimental approaches to investigate the concepts of hot and cold fibrosis in cardiac pathology and to identify targets to reduce fibrosis following cardiac injury. Previous mathematical modeling have demonstrated the important role of myofibroblast-macrophage interactions in driving and sustaining fibrosis.[18,19,24] Accordingly, we defined two types of fibrosis, hot fibrosis with coexisting myofibroblasts and macrophages

and cold fibrosis driven by myofibroblasts alone.[19] We developed a macrophage-myofibroblast double-reporter (MAMY) mouse that provides a tool to differentiate between cold and hot fibrosis *in vivo*. We find that acute MI results in cold fibrosis in mice, which is conserved in pigs and humans. After a transient immune-interacting state, myofibroblasts acquire a pro-fibrotic phenotype in cold fibrosis following MI and support their own proliferation by means of a growth factor autocrine loop. In contrast, chronic injury in mice resulted from either pressure overload injury (TAC) or the existence of a FBR (surgical suture) leads to hot fibrosis, with coexisting myofibroblasts and macrophages. Using a circuit-to-target approach, we predicted that inhibiting the myofibroblast autocrine loop should abrogate cold fibrosis. We identified TIMP1 as an autocrine growth factor for cardiac myofibroblasts. Antibody-mediated inhibition of TIMP1 reduced fibrosis in adult mice following MI.

Our findings suggest that fibrosis is different in distinct modes of cardiac injury. Acute injury in MI leads to cold fibrosis, whereas chronic injury—FBR and TAC—leads to hot fibrosis. Moreover, our results indicate that the pathology of acute MI consists of two phases, which could be viewed as two distinct disease states. The early phase is characterized by the activation of both myofibroblasts and macrophages with immune-interaction phenotypes. The late phase shows persistent myofibroblasts and a reduction in macrophage number and activity to basal levels.

Fibrosis in the heart has been mostly studied based on histological presentation. Histological definitions of fibrosis include replacement, interstitial, or perivascular fibrosis.[54] We propose that similar histological features can result in different fibrotic compositions (hot or cold fibrosis). Specifically, we identified replacement fibrosis as the outcome in both MI and TAC injuries. However, in MI, we observed cold fibrosis, whereas replacement fibrosis in TAC presents hot fibrosis. Our data thus suggest that fibrosis classification can be extended beyond the current histological categories.[21]

This study highlights the utility of a minimal mathematical model for identifying a target to reduce fibrosis. Targeting the autocrine loop makes sense because in cold fibrosis, the absence of macrophages leaves myofibroblasts as the primary source of their own growth factors. The model predicts that in order to abolish the cold fibrosis fixed point, it is not necessary to inhibit the autocrine loop completely. Instead, the loop needs only to be inhibited below a threshold (∼40%) leaving a single regenerative healing fixed point.

We identified TIMP1 as a cardiac myofibroblast autocrine growth factor both *in vivo* and *in vitro* and as a target for reducing cold fibrosis after MI. The canonical function of TIMP1 is inhibition of metalloproteinases, including MMPs.[85] Originally, TIMP1 was identified as a growth factor and is now emerging as a potent pro-inflammatory cytokine.[45] As a growth factor, TIMP1 facilitates cell-cycle activity in scleroderma fibroblasts[74] and hepatic stellate cells.[86] TIMP1-induced cellular proliferation was shown to be independent of its MMP inhibition activity in keratinocytes and multiple cell lines.[87,88] We show that TIMP1 acts as a growth factor for cardiac myofibroblasts and suggest that this activity is due to interaction with LRP1, but not with CD63 or APP. Future studies employing direct blocking or knockdown of LRP1 are necessary to fully clarify its role in

TIMP1-induced proliferation. Indirect activation of proliferation via MMP inhibition was not excluded. Our data suggest that TIMP1's function as a cardiac myofibroblast growth factor is conserved in mice, primates, and humans. In humans, increased levels of TIMP1 in the bloodstream have been linked to major cardiovascular risk factors.[89] Accordingly, TIMP1 knockout mice showed reduced fibrosis in a model of hypertension-induced chronic cardiac injury.[90] In contrast, these TIMP1 knockout mice had an exacerbated acute liver fibrosis.[72] Anti-TIMP1 antibody treatment reduced virus-induced myocarditis in mice.[91] It would be intriguing to see whether inhibiting TIMP1 alleviates fibrosis in other contexts in which it acts as a growth factor for pro-fibrotic cells. Agrin treatment, shown to reduce fibrosis after MI, appears to work by a distinct mechanism, inducing regenerative senescence in cardiac fibroblasts.[92]

Although TIMP1 is expressed exclusively by cardiac myofibroblasts post-MI, we find that TIMP1 receptors are present in all cardiac interstitial cells. This suggests, in the context of MI, a widespread influence of TIMP1 not limited to cardiac myofibroblasts. Our *in vivo* data suggest that TIMP1 inhibition did not affect monocyte (GFP+) infiltration, macrophage proliferation, or fate; however, we cannot rule out potential roles of TIMP1 in other arms of inflammation and subsequent fibrosis.[45]

Inhibiting the myofibroblast autocrine loop is an example of a more general strategy in which modulation of growth factor interactions promotes the collapse of an entire cell population. This strategy differs in a qualitative way from other approaches that aim at affecting ECM deposition from pro-fibrotic cells or modulating their phenotypes. The collapse of a cell type removes all of its secreted factors in a single step and thus may be more decisive as a treatment for a complex pathology such as fibrosis. An example is the recent use of chimeric antigen receptor-T (CAR-T) technology to specifically target cardiac myofibroblasts.[37]

In line with our study, a recent publication used a similar autocrine loop inhibition strategy in the context of late stage liver cirrhosis.[93] This study did not address the question whether liver cirrhosis (severe liver scarring) is hot or cold. Interestingly, elevated blood levels of TIMP1 induce an autocrine signaling loop of TIMP1 expression in hepatic stellate cells.[94] Thus, the inhibition of the autocrine signaling loop of a key pro-fibrotic cell population might be a general strategy to target fibrosis across organs.

Whereas cold fibrosis depends on the myofibroblast autocrine signaling loop, one may ask what drives hot fibrosis. Our model indicates that hot fibrosis requires paracrine signaling between macrophages and myofibroblasts. Key paracrine signals supporting hot fibrosis in chronic cardiac injuries were reported in recent studies showing that activated macrophages secrete IL1β that promotes fibroblast activation and subsequent fibrosis in cardiac-pressure-overloaded mice and humans.[95,96] It would be fascinating to explore hot and cold fibrosis in different organs and contexts.

The macrophage fibroblast circuit may apply beyond fibrosis: recent work identified a similar circuit with an autocrine loop in the breast tumor microenvironment.[97] Fibroblasts had a strong autocrine loop and the strongest outgoing signaling, whereas macrophages had the strongest incoming signals. Viewing the cancer microenvironment as a type of hot fibrosis may help to use the present circuit-to-target approach to design interventions to collapse the tumor microenvironment for therapeutic purposes.[19,98]

In summary, we tested *in vivo* the concepts of hot and cold fibrosis from a mathematical cell-circuit model and found that acute MI evolves to cold fibrosis, whereas chronic cardiac injuries evolve to hot fibrosis. We propose that the therapeutic strategies should differ between cold and hot fibrosis and use the cell circuit to identify the autocrine loop as a target in cold fibrosis. We further demonstrate a treatment strategy to reduce fibrotic scarring in mouse heart by inhibiting an autocrine growth factor of myofibroblasts. Our study highlights the utility of mechanistic yet simple mathematical models to cut through the complexity of pathological processes and to provide useful concepts and molecular strategies for treatment.[3]

### RESOURCE AVAILABILITY

**Lead contact**

Further information and requests for resources should be directed to and will be fulfilled by the lead contact, Eldad Tzahor (eldad.tzahor@weizmann.ac.il).

**Materials availability**

This study did not generate new materials.

**Data and code availability**

- Data: all data are available in the main text or the supplementary materials. All count matrices and metadata for each transcriptomic dataset: pig bulk-RNA sequencing and Visium spatial transcriptomics, will be publicly available in the Gene Expression Omnibus (http://www.ncbi.nlm.nih.gov/geo/) under data accession no. GEO: GSE247856, and GEO: GSE249517, as of the date of publication.
- Code: all custom generated code are available in the AlonLabWIS git. This repository has been archived at Zenodo with (https://doi.org/10.5281/zenodo.14555560).
- Any additional information required to reanalyze the data reported in this paper is available from the lead contact upon request.

### ACKNOWLEDGMENTS

We thank Marina Cohen from the Histology Unit, Department of Veterinary Resources, Weizmann Institute of Science for histological processing and staining. We also thank Dr. Hadas Keren-Shaul and Danna Robbins (The Nancy & Stephen Grand Israel National Center for Personalized Medicine, Weizmann Institute of Science) for pig bulk-mRNA sequencing library preparation and sequencing. We would like to acknowledge histology and next-generation sequencing facilities at the Vienna Biocenter Core Facilities (VBCF) for all aspects of Visium handling, sectioning, and sequencing. We thank Prof. Steffen Jung, Dr. Jung-Seok Kim, and Dr. Sebastien Trzebanski from the Department of Immunology and Regenerative Biology, Weizmann Institute of Science, Rehovot, Israel for insightful comments and technical assistance in flow cytometry experiments. This study has been supported by grants to E.T. from the European Research Council (ERC AdG grant no. 788194, CardHeal), REANIMA European Union's Framework Programme for Research and Innovation "Horizon2020." E.T. and R.S. were supported by the Israel Science Foundation (ISF, no. 2214/22). This study was further supported by a Cancer Research UK grant (C19767/A27145) and by the European Research Council under the European Union's Horizon 2020 research and innovation program (grant agreement No. 856487) to U.A. M.A. was supported by the EMBO Long-Term Fellowship (ALTF 304-2019), the Zuckerman STEM Leadership program, and the Israel National Postdoctoral Award Program for Advancing Women in Science. C.K. was funded by the ERC Advanced Grant Cor-Edit-P (grant no. 101021043). E.M.T. was funded by REANIMA European Union's Framework

## Article

Programme for Research and Innovation "Horizon2020" and FWF P36045 "Regenerative strategies for cardiac repair." E.B was funded by the EMBO ALTF447-2020 and the Marie Curie Post-doctoral individual fellowship "Axomatrx." A.K. and D.H. were funded by the Deutsche Forschungsgemeinschaft, Bonn, Germany (KR2047/8-2, KR2047/14-1, and KR2047/15-1).

## AUTHOR CONTRIBUTIONS

Conceptualization: S.M., M.A., A.M., U.A., and E.T.; methodology: S.M., M.A., and A.M.; investigation: S.M., M.A., A.M., E.B., D.H., Y.D., K.B.U., J.E., A.G., D.K., D.L., T.S., M.G., A.S., L.Z., J.W., A.B., C.K., H.B.-L., R.O.R.F., O.G., E.W., T.D., and Z.P.; visualization: S.M., M.A., A.M., and D.M.K.; funding acquisition: E.T. and U.A.; project administration: E.T. and U.A.; supervision: E.T., U.A., R.M., E.M.T., C.K., J.S.-R., A.K., and A.L.M.; writing—original draft: S.M., M.A., A.M., U.A., and E.T.; writing—review & editing: all authors.

## DECLARATION OF INTERESTS

E.T. is a Founder of a biomedical startup related to Agrin therapy for heart disease and has related patents. J.S.-R. reports funding from GSK, Pfizer, and Sanofi and fees/honoraria from Travere Therapeutics, Stadapharm, Astex, Pfizer, Grunenthal, Moderna, Tempus and Owkin.

## STAR★METHODS

Detailed methods are provided in the online version of this paper and include the following:

- KEY RESOURCES TABLE
- EXPERIMENTAL MODEL AND STUDY PARTICIPANT DETAILS
  - Mice
- METHOD DETAILS
  - Mouse myocardial infarction
  - Mouse transverse aortic constriction
  - Pigs heart samples
  - Flow Cytometry
  - Primary mouse cardiac myofibroblast cultures
  - Non-human-primate cardiac fibroblasts cultures
  - Immunofluorescence
  - Histology and picro-sirius red staining
  - Spatial transcriptome processing (Mouse)
  - Pig mRNA library preparation and sequencing
- QUANTIFICATION AND STATISTICAL ANALYSIS
  - Cell segmentation and quantification
  - Double-reporter (MAMY) mice neighborhood analysis
  - Fibrosis analysis
  - Spatial transcriptomics analysis (Mouse)
  - Human spatial transcriptomics Analysis
  - Pig Bulk-mRNA sequencing analysis
  - Deconvolution of bulk-mRNA sequencing
  - Analysis of mouse Bulk-mRNA sequencing
  - Analysis of mouse single-cell mRNA sequencing
  - Pareto analysis
  - NicheNet analysis
  - myofibroblast proliferation score
  - *In-silico* perturbations of cold fibrosis
  - Statistical analysis

## SUPPLEMENTAL INFORMATION

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

**Cell Systems**

## STAR★METHODS

### KEY RESOURCES TABLE

| REAGENT or RESOURCE | SOURCE | IDENTIFIER |
|---|---|---|
| **Antibodies** | | |
| anti-TIMP1 | R&D Systems | Cat# AF980 |
| IgG control | R&D Systems | Cat# AB-108-C |
| Anti-CD16/32 | BioLegend | clone: 93 |
| Anti-CD45 | BioLegend | clone: 30-F11 |
| Anti-CD11b | BioLegend | clone: M1/70 |
| Anti-GR-1 | BioLegend | clone: RB6-8C5 |
| Anti-F4/80 | Bio-Rad | Cl:A3-1 |
| Anti-TIM4 | BioLegend | Clone: RMT4-54 |
| Anti-LRP1 | Abcam | Cat# ab92544 |
| Anti- αSMA | Sigma Aldrich | Cat# A2547 |
| Anti-GFP | Abcam | Cat# ab6673 |
| Anti-RFP | Rockland | Cat# 600-401-379 |
| Anti-Galectin-3 | BioLegend | clone: M3/38 |
| Anti-cardiac troponin T | DSHB | clone: CT3 |
| Cy™3 AffiniPure Fab Fragment Goat Anti-Mouse IgG2a | Jackson ImmunoResearch | Cat# 115-167-186 |
| Cy™3 AffiniPure Donkey Anti-Rabbit IgG (H+L) | Jackson ImmunoResearch | Cat# 711-165-152 |
| Alexa Fluor® 647 AffiniPure Donkey Anti-Mouse IgG (H+L) | Jackson ImmunoResearch | Cat# 715-605-150 |
| Alexa Fluor® 488 AffiniPure Donkey Anti-Mouse IgG (H+L) | Jackson ImmunoResearch | Cat# 715-545-150 |
| Alexa Fluor® 647 AffiniPure Donkey Anti-Mouse IgG (H+L) | Jackson ImmunoResearch | Cat#705-605-147 |
| Donkey Anti-Goat IgG H&L (Alexa Fluor® 488) | Abcam | Cat# ab150129 |
| **Chemicals, peptides, and recombinant proteins** | | |
| Isoflurane | Abbott Laboratories | Cat# AWN-34014202 |
| Buprenorphine | Richter Pharma | N/A |
| Biological glue- Histoacryl® | B. Braun Melsungen AG | Cat# LUXBBS105005/2 |
| Tamoxifen | Sigma Aldrich | Cat# T5648 |
| Corn oil | Sigma Aldrich | Cat# C8267 |
| 5-ethynyl-2'-deoxyuridine (EdU) | Thermo-Fisher | Cat# C10424 |
| rhAgrin | R&D Systems | Cat# 6624-AG |
| Collagenase-I | Sigma Aldrich | Cat# C0130 |
| Collagenase-XI | Sigma Aldrich | Cat# C7657 |
| Hyaluronidase | Sigma Aldrich | Cat# H3506 |
| Collagenase II | Worthington | Cat# LS004177 |
| Recombinant-mouse TIMP1 | Abcam | Cat# ab206786 |
| Recombinant human WT TIMP-1 | In-house synthesized. Schoeps et al.[82] | N/A |
| Recombinant human N-TIMP-1 | In-house synthesized. Schoeps et al.[82] | N/A |
| **Deposited data** | | |
| Pig bulk-RNA sequencing | This paper | GSE247856 |
| Visium spatial transcriptomics (Mice) | This paper | GSE249517 |
| **Experimental models: Organisms/strains** | | |
| Hsd: ICR (CD1) mice | Envigo | N/A |

*Continued*

| REAGENT or RESOURCE | SOURCE | IDENTIFIER |
|---|---|---|
| Postn<sup>MerCreMer</sup> mice | Jackson Laboratory | #029645 |
| Rosa26<sup>TdTomato</sup> mice | Madisen et al.[53] | N/A |
| Cx3cr1<sup>GFP</sup> mice | Jackson Laboratory | #005582 |
| Postn<sup>MCM/+</sup>Rosa26<sup>TdTomto/+</sup>Cx3cr1<sup>GFP/+</sup> mice | This paper | N/A |
| **Oligonucleotides** | | |
| Genotyping primers for Postn<sup>MerCreMer</sup>: WT-Forward- GGT GCT TCT GTA AGG CCA TC; Common- CCT TGCAAT AAG TAAAACAGCTC, Mutant Forward-GGT GGGACATTTGAGTTGCT | Jackson Laboratory | #029645 |
| Genotyping primers for Rosa26<sup>TdTomato</sup>: Mutant forward- GGCATTAAAGCAGCGT ATCC, mutant reverse- CTGTTCCTGTAC GGCATGG, internal control forward- CCGAAAATCTGTGGGAAGTC, internal control reverse-AAGGGAGCTGCAGT GGAGTA. | Madisen et al.[53] Duva et al.[36] | N/A |
| Genotyping primers for Cx3cr1<sup>GFP</sup>: 1-TTCACGTTCGGTCTGGTGGG, 2-GGTTCCTAGTGGAGCTAGGG, 3-GATCACTCTCGGCATGGACG | Prof. Steffen's Jung's lab (Weizmann Institute of Science) | N/A |
| **Software and algorithms** | | |
| Original code | This paper | https://doi.org/10.5281/zenodo.14555560 |
| Vevo Lab Software Version 5.8.1 | VisualSonics | https://www.visualsonics.com/product/software/vevo-lab |
| Diva Software | BD | https://www.bdbiosciences.com/en-eu/products/software/instrument-software/bd-facsdiva-software |
| FlowJo Software Version 10.6.2 | BD | https://www.flowjo.com/ |
| QuPath software Versions 0.5.1 | Bankhead et al.[99] | https://qupath.github.io/ |
| Qupath StarDist Extension | Schmidt et al.[100] | https://github.com/qupath/qupath-extension-stardist |
| ImageJ/Fiji software | Schneider et al.[101] Schindelin et al.[102] | https://imagej.net/downloads |
| CaseViewer Software Version 2.4 | 3DHISTECH | https://www.3dhistech.com/ |
| SpaceRanger Version 1.3.0 | 10X Genomics | https://www.10xgenomics.com/support/software/space-ranger/latest |
| Seurat algorithm Version 4.1.1 | Satija et al.[103] Butler, et al.[104] | https://satijalab.org/seurat/ |
| Loupe Browser Software Version 6.1.0 | 10X Genomics | https://www.10xgenomics.com/support/software/loupe-browser/latest |
| Wolfram Mathematica Software Versions 14.0 | Wolfram Alpha | https://www.wolfram.com/mathematica/quick-revision-history/ |
| Scanpy algorithm Version 1.9.1 | Wolf et al.[105] | https://github.com/scverse/scanpy |
| Liana-py algorithm Version 1.1.0 | Dimitrov et al.[76] | https://github.com/saezlab/liana-py |
| Htseq-count version 0.11.2 | Anders et al.[106] | https://htseq.readthedocs.io/en/master/index.html |
| STAR algorithm Version 2.7.3a | Doblin et al.[107] | https://github.com/alexdobin/STAR |
| DESeq2 algorithm Version 1.26.0 | Love et al.[108] | https://www.bioconductor.org/packages/release/bioc/vignettes/DESeq2/inst/doc/DESeq2.html |
| Rstudio Software Version 3.6.1 | Posit PBC | https://posit.co/download/rstudio-desktop/ |
| Morpheus | N/A | https://software.broadinstitute.org/morpheus |

*(Continued on next page)*

*Continued*

| REAGENT or RESOURCE | SOURCE | IDENTIFIER |
|---|---|---|
| ParTi algorithm | Hart, etl al.[65] | https://github.com/AlonLabWIS/ParTI |
| Sanity algorithm | Breda et al.[109] | https://github.com/jmbreda/Sanity |
| NicheNet algorithm | Browaeys et al.[68] | https://github.com/saeyslab/nichenetr |
| GraphPad Prism software Version 8.0.1 | GraphPad | https://www.graphpad.com/ |
| Other | | |
| Trypsin EDTA solution B | Sartorius | Cat# 03-052-1B |
| Click-IT EdU flow cytometry kits | Invitrogen | Cat# C10424, #Cat C10420 |
| Neonatal heart dissociation kit | Miltenyi Biotec | Cat# 130-098-373 |
| GentleMACS C-Tubes | Miltenyi Biotec | Cat# 130-093-237 |
| DMEM/F12 | Sigma Aldrich | Cat# D6421 |
| L-glutamine | Biological Industries | Cat# 03-020-1B |
| Pyruvate | Biological Industries | Cat# 03-042-1B |
| Non-essential amino acids | Biological Industries | Cat# 01-340-1B |
| penicillin, streptomycin, amphotericin B | Biological Industries | Cat# 03-033-1B |
| Horse serum | Biological Industries | Cat# 04-004-1A |
| FBS | Biological Industries | Cat# 04-007-1 |
| Gelatin | Sigma Aldrich | Cat# G1393 |
| DMEM/F12 | Thermo Fisher Scientific | Cat# 21331020 |
| FBS | Sigma Aldrich | Cat# F7524-500ML |
| L-glutamine | Thermo Fisher Scientific | Cat# 25030081 |
| Penicillin–streptomycin | Thermo Fisher Scientific | Cat# 15140122 |
| Fibronectin-coated plates | Sigma Aldrich | Cat# F1141-5MG |
| Paraformaldehyde | Electron Microscopy Sciences | Cat# 15710 |
| Citrate buffer | Sigma Aldrich | Cat# C999 |
| Heat-inactivated horse serum | Biological Industries | Cat# 04-124-1A |
| Immu-Mount | Thermo Fisher | Cat# 9990402 |
| Paraffin | Leica | Cat# 39601006 |
| Picro-Sirius red staining | Abcam | Cat# ab246832 |
| Optimal Cutting Temperature (OCT) | Sakura Finetek | Cat# 4583 |
| SuperFrost plus slides | Thermo Scientific | Cat# 630-0951 |
| Hematoxylin | Fisher Scientific | Cat# 6765002 |
| Eosin | Fisher Scientific | Cat# 10483750 |
| Visium Spatial Tissue Optimization Slide | 10X Genomics | Cat# 1000193 |
| Visium spatial gene expression slide | 10X Genomics | Cat# 1000187 |
| miRNeasy Mini Kit | Qiagen | Cat# 217004 |

## EXPERIMENTAL MODEL AND STUDY PARTICIPANT DETAILS

### Mice

All mice were housed in a pathogen-free environment at the Weizmann Institute of Science animal facility. Mice were provided *ad libitum* access to food and water and were monitored daily for health and activity. The animal housing area maintained an ambient temperature of approximately 22.5°C and 50% humidity. Mice from different experimental groups were housed under 12-hour light–dark cycles and received identical treatment conditions. All experimental procedures were approved by the Animal Care Committee of the Weizmann Institute of Science (approval numbers: 00660121-2, 00510122-3, 00060124-3, 03770624-2), and all studies were compliant with all the relevant ethical regulations regarding animal research.

Mice used in this study were either purchased from Envigo or bred in house. Mice purchased from Envigo include Hsd: ICR (CD1), Female, 31-34 grams, 10-12 weeks. Mice arrived at least 4 days before any procedure. In house bred mice include the Postn$^{MCM/+}$Rosa26$^{TdTomto/+}$Cx3cr1$^{GFP/+}$ Macrophage-myofibroblast double reporter mice (MAMY). These were achieved in 2 consecutive breeding steps. First, to obtain a cardiac myofibroblast lineage tracer,[48] we crossed the tamoxifen inducible-Cre mice (Postn-MerCreMer, Jackson Laboratory, #029645) with Rosa26-TdTomato mice. Rosa26-tdTomato indicator mice carry a conditional allele for a red fluorescent protein variant, which is only expressed following CRE-mediated recombination.[53] We then crossed Postn$^{MCM/MCM}$Rosa26$^{TdTomto/TdTomato}$ mice with Cx3cr1$^{GFP/GFP}$[48] (Prof. Steffen Jung's lab, Weizmann Institute of Science.

Also available at Jackson Laboratory, #005582) to obtain the MAMY animals (Note S1). Genotyping primers used to validate transgenic animals is as follows: for Postn-MCM: WT-Forward- GGT GCT TCT GTA AGG CCA TC; Common- CCT TGCAAT AAGTAAAAC AGCTC, Mutant Forward-GGTGGGACATTTGAGTTGCT. For Cxrcr1-GFP we used 3 primers: 1-TTCACGTTCGGTCTGGTGGG, 2-GGTTCCTAGTGGAGCTAGGG, 3-GATCACTCTCGGCATGGACG. For Rosa26- TdTomato we used 4 primers: Mutant forward- G GCATTAAAGCAGCGTATCC, mutant reverse- CTGTTCCTGTACGGCATGG, internal control forward- CCGAAAATCTGTGGGAA GTC, and internal control reverse- AAGGGAGCTGCAGTGGAGTA.

## METHOD DETAILS

### Mouse myocardial infarction

Myocardial infarction (MI) was induced in adult male and female MAMY mice: Postn$^{MCM/+}$Rosa26$^{TdTomto/+}$Cx3cr1$^{GFP/+}$ (12-20 weeks of age), and in female 12 weeks Hsd: ICR (CD1) mice by a permanent ligation of the left anterior descending (LAD) coronary artery. Mice were anesthetized using isoflurane (Abbott Laboratories, Cat# AWN-34014202) and, following tracheal intubation, were artificially ventilated. Toe pinch was performed to test the depth of anesthesia. Prior to all surgical procedures mice were administered with buprenorphine in a subcutaneous manner, as an analgesic agent (0.066 mg/kg). Lateral thoracotomy was performed at the third intercostal space by blunt dissection of the intercostal muscles. Following the exposure of the left ventricle the LAD an 8-0 silk suture was used to tie the LAD coronary artery. The suture was validated by holding in the tie for 5 seconds, and the observance of an ischemic (white) area below the suture. Following this validation, the procedure ended by closing the chest cavity and skin by either suture or biological glue- Histoacryl® (B. Braun Melsungen AG, Cat# LUXBBS105005/2), respectively. Following awakening, mice were warmed for several minutes in a recovery cage. Sham operated animals underwent similar surgical procedures without the LAD constriction.

To activate the Tamoxifen inducible Postn driven MerCreMer system in MAMY (Postn$^{MCM/+}$Rosa26$^{TdTomto/+}$Cx3cr1$^{GFP/+}$) mice experiments, Tamoxifen (Sigma, Cat# T5648) diluted in corn oil (Sigma, Cat#C8267) according to mice weight (50ug/gr) was injected intraperitoneally (IP) immediately after MI and for 6 more days (daily).

For anti-TIMP1 neutralizing antibody experiments - immediately after LAD ligation, anti-TIMP1 (AF980, R&D) antibodies or IgG control (AB-108-C, R&D) were injected using an insulin syringe directly to the myocardium of the infarct area at a concentration of 0.1ug/ gr. Mice were also given a booster injection with a single retro-orbital sinus injection of anti-TIMP1 antibodies or IgG control (0.166ug/gr).

For the assessment of myofibroblasts proliferation following anti-TIMP1 antibodies or IgG control we performed a 24 hours EdU pulse in MAMY mice following MI. Mice were injected subcutaneously with 12.5mg/kg EdU (Thermo-Fisher, Cat#C10424) 3 days following-MI. Heart were collected on day 4, 24 hours followed EdU pulse initiation.

For the evaluation cardiac function, we used transthoracic echocardiography performed on mice sedated with 2-3% isoflurane (Abbot Laboratories) using Vevo3100 (VisualSonics). Analysis was performed in a blinded manner using Vevo Lab 3.2.6 software (VisualSonics). In all MI experiments mice with baseline ejection fraction (EF) values lower than 45% were excluded from the experiment. Mice were then randomized to treatment groups with equal EF values. In anti-TIMP1/IgG control fibrosis and electrocardiography experiment (Figures 5M–5O, S9G, and S9H), animals that were too severely injured 3 days following injury (EF< 30%) were excluded from the experiment.

### Mouse transverse aortic constriction

Chronic ventricular pressure overload was induced by transverse aortic constriction (TAC). Postn$^{MCM/+}$Rosa26$^{TdTomto/+}$Cx3cr1$^{GFP/+}$ MAMY mice were anesthetized using isoflurane and, following tracheal intubation, were artificially ventilated. Toe pinch was performed to test the depth of anesthesia. Prior to all surgical procedures mice were administered with buprenorphine in a subcutaneous manner, as an analgesic agent (0.066 mg/kg). Skin incision and anterior thoracotomy in 3rd intercostal space was performed. Aortic constriction was carried out by encircling the ascending aorta using two loose knots, against a 27G blunt needle placed in a parallel position to the aorta. The first and second knots were quickly tied against the needle followed by a prompt removal of the needle, leaving a permanent knot on the aorta. Chest wound was closed with a 5-0 suture and biological glue- Histoacryl®. Mice awakening and recovery was done under warm conditions in a separate recovery cage. Sham operated animals underwent similar surgical procedures without the aortic constriction.

Tamoxifen diluted in corn oil (according to mice weight; 50ug/gr) was injected intraperitoneally (IP) immediately after TAC and for 6 more days. As TAC is a form of chronic injury tamoxifen was further injected on days 13-14 and 26-27 post-TAC initiation to collect the complete TdTomato signal.

For the evaluation of baseline and post-injury cardiac function and aortic pressure gradient *(mmHg)*, we used transthoracic echocardiography performed on mice sedated with 2-3% isoflurane (Abbot Laboratories) using Vevo3100 (VisualSonics). To evaluate the pressure gradient of the aortic arch we used color doppler, identified the construction site and placed the pulsed doppler cursor in parallel to the blood flow distant to the surgical node and measured the blood flow velocity *(V)*. To determine the pressure gradient *(ΔP)*, we used the modified Bernoulli equation *($\Delta P=4*V^2$)* in each mouse. Analysis was performed in a blinded manner using Vevo Lab 3.2.6 software.

For the evaluation of muscle hypertrophy, we measured heart weight (HW) to body weight (BW) ratio of mice following TAC. both BW and HW were measured using a balanced analytical scale (PTX-FA210S).

### Pigs heart samples

All pig samples used in this study were obtained from a previous study.[64] In the original study, Landrace pigs, 3 months old (male and female) ~22-35 kg, were purchased from a local farm. Animal care and all experimental procedures were performed in accordance with the German and National Institutes of Health animal legislation guidelines. All procedures were approved by the local animal care and use committees and reviewed and approved by the responsible local regulatory authority (Regierung von Oberbayern, Sachgebiet 54). All pig experiments were conducted in the Institute for Surgical Research at the Technical University of Munich (TUM) as described previously.[63,64]

To induce ischemia-reperfusion injury, a percutaneous transluminal coronary angioplasty balloon was placed in the LAD distal to the bifurcation of the first diagonal branch and inflated with 6 atm. Correct localization of the coronary occlusion and patency of the first diagonal branch was ensured by injection of contrast agent. The percutaneous transluminal coronary angioplasty balloon was deflated after 60 minutes of ischemia; the onset of reperfusion was documented angiographically. At the onset of reperfusion, animals were treated with 33 μg/Kg (in 5mL of saline) of rhAgrin (#6624-AG, R&D Biosystems) or saline alone as a control.

Pig heart samples were collected on days 3 and 28 following the MI. Hearts were harvested and stained with Tetrazolium chloride (TTC) to visualize scar/infarct. Left ventricles were dissected transversely into 5, 1cm thick slices, and each slice was further sectioned into 4/8 sections. Sections were annotated as infarct zones (positive for TTC) and remote zones (TTC negative). Sections were frozen on dry ice and stored for subsequent histological (staining) and RNA (sequencing) analysis.

In this study, we exclusively utilized archived heart samples from *Behar et al.*[64] These samples were accessed with permission, and no new animal experiments were performed for this work.

### Flow Cytometry

Adult mice were euthanized by neck dislocation and hearts were perfused with 10 ml of cold phosphate buffered saline (PBS). Left ventricles were dissected, below the surgical suture, chopped finely and digested in RPMI containing Collagenase-I (Sigma, 450U/ml), Collagenase-XI (Sigma, 120U/ml), Hyaluronidase (Sigma, 60U/ml) and DNase-I (Sigma, 10mg/ml), 37°C, for 1 hour, while agitated (200rpm). Digested materials were passed through a 100um metal mesh into 12 ml PBS and pelted by cooled centrifugation (4°C, 1400rpm, 5min). Supernatant was removed and the pellet was resuspended in 3ml of MACS buffer (2%FBS,1mM EDTA in PBS). Cells were collected by cooled centrifugation (4°C, 1400rpm, 5min), supernatant was removed, and samples were blocked in 50ul of anti-CD16/32 antibodies (BioLegend, clone: 93, 1:200) in MACS buffer (20min, 4°C). Samples were resuspended in a 3ml MACS buffer, centrifuged (4°C, 1400rpm, 5min), supernatant was removed, and cell surfaces were stained in 50ul of antibody mix in MACS buffer (20 min, 4°C in the dark). Samples were washed in a 3ml MACS buffer containing DAPI (1:10,000) and pelted by cooled centrifugation (4°C, 1400rpm, 5min). Single cell suspension was resuspended in 0.5ml of MACS buffer and filtered through 40 um mesh prior to flow cytometry analysis by Fortessa (BD Biosciences, BD Diva Software).

For the primary mouse cardiac myofibroblast proliferation assay, cells were incubation with 10uM EdU (5-ethynyl-2' -deoxyuridine) (Invitrogen, C10424) for 48 hours together with either recombinant-mouse TIMP1 protein (1μg/ml, ab206786, Abcam), PBS, anti-TIMP1 (1ug/ml, AF980, R&D) antibodies or IgG control (1μg/ml, AB-108-C, R&D). Cells were washed once with warm (37°C) PBS and then lifted from wells using Trypsin EDTA solution B (Sartorius, Cat# 03-052-1B) (5mins, 37°C). Trypsin solution B was neutralized by adding full-DMEM media and cells were pelted by centrifugation (Room temperature, 2200rpm, 5min). Cells were then washed once with 1% bovine serum albumin (BSA) in PBS and pelted by cooled centrifugation (4°C, 1400rpm, 5min). Cells were fixed and permed using Click-IT fixative and perm buffers and further processed based on the manufacturer's instructions (Invitrogen, Cat# C10424). Cells were then filtered through a 40μm mesh prior to flow cytometry analysis. Following data acquisition single cell analysis was performed using FlowJo (v10.6.2) software.

Gating for flow cytometry analysis: Myeloid cells were defined as single cells, DAPI⁻CD45⁺CD11b⁺. Infiltrating cells (Neutrophils and Monocytes) were defined as single-cells, DAPI⁻CD45⁺CD11b⁺GR-1⁺ cells. Macrophages were defined as single-cells, DAPI⁻CD45⁺CD11b⁺GR-1-F4/80⁺ cells, and further parsed based on TIM4 in order to confirm tissue residency.[61] Proliferating primary cardiac myofibroblasts were defined as Edu⁺ cells/ total single cells identified.

Mouse Antibodies: CD45 (BioLegend, clone: 30-F11, 1:100), CD11b (BioLegend clone: M1/70, 1:100), GR-1 (BioLegend, clone: RB6-8C5, 1:100), F4/80 (Bio-Rad, clone: Cl:A3-1, 1:80), TIM4 (BioLegend, clone: RMT4-54, 1:100).

### Primary mouse cardiac myofibroblast cultures

For primary cardiac myofibroblast enriched cultures, cells were first isolated from adult (10-12 weeks) female ICR mice hearts using a neonatal dissociation kit (Miltenyi Biotec,130-098-373) and gentleMACS homogenizer (Miltenyi Biotec). Following euthanasia hearts were extracted, rinsed in PBS and chopped finely on a sterile surface using fine scissors. Chopped pieces were then transferred to GentleMACS C-Tubes (Miltenyi Biotec, Cat# 130-093-237) together with 2.5ml of the neonatal dissociation kit (Miltenyi Biotec,130-098-373) digestion buffer (prepared based on manufacturer's instructions). GentleMACS C-Tubes were then incubated for 25 mins at 37°C, and then spun using the GentleMACS homogenizer (Miltenyi Biotec) for 30secs (spin program: 'm_neoheart_01_01'). GentleMACS C-Tubes were then incubated at 37 °C for 25 mins (this step was repeated twice to achieve a total of 4 incubations and spins). Following the 4th incubation in 37 °C for 25 mins, cells were spun twice. Cell suspension was then neutralized with pre-heated (37 °C) full-DMEM media- DMEM/F12 (Sigma, Cat# D6421) supplemented with L-glutamine (1%; Biological Industries, Cat# 03-020-1B), pyruvate (1%; Biological Industries, Cat# 03-042-1B), non-essential amino acids (1%; Biological Industries, Cat# 01-340-1B), penicillin, streptomycin, amphotericin B (1%; Biological Industries, Cat# 03-033-1B), horse serum (5%; Biological

Industries, Cat# 04-004-1A) and FBS (10%; Biological Industries, Cat# 04-007-1). Cells were then pelleted by centrifugation (5mins, room temperature, 2200rpm). Resuspended in 1ml full-DMEM media, cells were filtered through a 100μm m mesh and cultured in gelatin-coated (0.1%; Sigma, Cat# G1393) 6-well plates. The next day cells were washed three times and media was replaced with fresh media. On the 3rd day following isolation cells were washed twice and media was replaced with fresh media. On day four, cells were split and seeded in 12-well plates ($10^5$ cells/well) for flow cytometry analysis or in 96-well plates (either $5*10^3$ or $2*10^4$ cells/well) for immunofluorescence staining. Cells were then cultured for 48 hours and either fixed, by 4% paraformaldehyde (Electron Microscopy Sciences, Cat# 15710), and stained for immunofluorescence analysis or further processed for flow cytometry analysis. Throughout the experimental period cells were placed in an incubator (37°C and 5% $CO_2$).

### Non-human-primate cardiac fibroblasts cultures

Non-human-primate (NHP) cardiac fibroblasts were isolated as previously described.[110] Briefly, explanted NHP left ventricle tissue obtained from German primate centre, Göttingen (reference 33.19-42502-04-16/2264), Karolinska Institutet, Sweden (reference N 277/14) or Walter Brendel Institute, Germany (reference ROB-55.2–2532.Vet_02-14-184) were minced and incubated with 550 units/ml collagenase II (Worthington,Cat# LS004177) for 15 min at 37°C for a series of six rounds of dissociations. After each digestion, supernatant containing dissociated cells was collected. Isolated cardiac fibroblasts were washed twice with DMEM–F12 (Thermo Fisher Scientific, Cat# 21331020) and cultured in cFibro medium containing: DMEM–F12, 10% FBS (Sigma-Aldrich, Cat# F7524-500ML), 2 mM L-glutamine (Thermo Fisher Scientific, Cat# 25030081) and 1% penicillin–streptomycin (Thermo Fisher Scientific, Cat# 15140122) on fibronectin-coated plates (Sigma-Aldrich, Cat# F1141-5MG).

1.25x$10^4$ NHP cardiac fibroblasts were plated per well on fibronectin-precoated 96-well plates for 24h, washed twice with warm 1x PBS and cultured in serum-free media (DMEM + 0.1% BSA + P/S). After another 24h, cells were treated with 17.54nM of in-house synthesized recombinant human TIMP-1 (rhWT-TIMP1) or N-TIMP-1 (rhN-TIMP1),[82] or PBS in the presence of 10nM EdU for 48h. Proliferation of cells was determined with the Click-it EdU Alexa fluor 488 flow cytometry kit (Thermo-Fisher Scientific, #Cat C10420) based on manufacturer's instructions and detected on a SA3800 (Sony) flow cytometer. Analysis was performed with FlowJo version 10 (FlowJo Inc.). Single cells were gated from all detected events and EdU-positive cells were detected based on unstained control.

### Immunofluorescence

Immunofluorescence was performed for both tissue culture plates and tissue sections from formalin fixed paraffin embedded (FFPE) blocks. Cells in 96-well plates were fixed by 4% paraformaldehyde for 10 minutes, followed by 3 washes of PBS (Room temperature) on a shaker. Following fixation cells were permeabilized by 0.1% Triton X-100 for 7 minutes and blocking with 2% BSA in PBS for 1 hour in room temperature, on a shaker. Primary antibodies were prepared in blocking solution and antibodies were incubated overnight (4°C, gentle agitation). Cells were then washed 3 times for 10 min with PBS and stained with the suitable secondary antibody (1:200, Jackson ImmunoResearch) and DAPI (4,6-diamidino-2-phenylindole dihydrochloride) for 1h at room temperature. Wells were later washed 3 times with PBS in room temperature, on a shaker. Images were obtained with a Nikon Eclipse Ti2 fluorescent microscope with Nikon NIS-Elements software.

Immunofluorescence of FFPE MAMY mice tissue sections (5um-thick) following MI/Sham or TAC/sham was performed by deparaffinization and heat-mediated antigen retrieval in citrate buffer (Sigma, Cat# C999, pH6). Slides in citrate buffer were left to cool off to room temperature on an orbital shaker. Slides were then permeabilized (0.5%Triton X-100 in PBS) for 12 minutes and blocked in a humidity chamber at room temperature conditions for 1 hour in blocking solution: 5% heat inactivated horse serum (Biological Industries Cat# 04- 124-1A) in 0.1% Triton X-100 in PBS). primary antibodies were then added in blocking solution for overnight incubation in a humidity chamber (at 4°C). Slides were then washed 3 times for 10mins in PBS, followed by the application of secondary antibodies (1:200) and DAPI (1:1000) diluted in PBS for 1hr in a humidity chamber at room temperature conditions. Slides were then washed 3 times for 10mins in PBS and mounted with Immu-Mount (Thermo Fisher, Cat# 9990402). Slides were then imaged using PANNORAMIC SCAN II slide scanner (3DHISTECH).

The following primary antibodies were used for staining. anti-LRP1 (cell culture, 1:100, ab92544, Abcam), anti-TIMP1 (cell culture, 1:50, AF980, R&D systems), anti-aSMA (cell culture & tissue sections, 1:400, A2547, Sigma), Anti-GFP (tissue sections, 1:100, ab6673, Abcam), anti-RFP (tissue sections, 1:200, 600-401-379, Rockland), Anti-Gal3 (tissue sections, 1:100, M3/38, Biolegend), anti-cardiac troponin T (cTnT) (tissue sections, 1:20, CT3, DSHB).

The following secondary antibodies were used for staining: Cy™3 AffiniPure Fab Fragment Goat Anti-Mouse IgG2a (Jackson ImmunoResearch, Cat#115-167-186), Cy™3 AffiniPure Donkey Anti-Rabbit IgG (H+L) (Jackson ImmunoResearch, Cat#711-165-152), Alexa Fluor® 647/488 AffiniPure Donkey Anti-Mouse IgG (H+L) (Jackson ImmunoResearch, Cat#715-605-150/ 715-545-150), Alexa Fluor® 647 AffiniPure Donkey Anti-Mouse IgG (H+L) (Jackson ImmunoResearch, Cat#705-605-147), Donkey Anti-Goat IgG H&L (Alexa Fluor® 488) (Abcam, ab150129).

### Histology and picro-sirius red staining

Mouse hearts were briefly perfused with 10ml cold PBS and then fixed with 4% paraformaldehyde (Electron Microscopy Sciences, Cat# 15710) (overnight, shaking, 4°C). Paraformaldehyde fixed hearts samples were cut in the middle in a transverse manner and both half-heart pieces representing Base and Apex were embedded together in paraffin (Leica, Cat# 39601006). Embedded hearts were cut into 5-6 rows (holding 12 sections, 5μm thick), separated by 300μm intervals. Following block cutting, sections were baked

overnight (37°C). Picro-sirius red staining was done based on manufacturer's protocol (Abcam, Cat# ab246832). Sirius red sections were imaged in bright-field mode using the PANNORAMIC SCAN II slide scanner (3DHISTECH). Images were processed by CaseViewer v2.4 (3DHISTECH).

### Spatial transcriptome processing (Mouse)

Following MI, adult mice were euthanized, and hearts were harvested on days 1, 4, 7 and 14 post-Injury. Hearts were briefly washed in Optimal Cutting Temperature (OCT) (Sakura Finetek, Cat# 4583) solution and placed in a new mold containing OCT. The molds were placed in pre-cooled Isopentane bath in liquid $N_2$. Molds were placed on dry ice until storage in -80°C. For tissue sectioning, blocks were placed in cryostat (Epredia CryoStar NX70 cryostat, Thermo Fisher) until equilibration and were sectioned at 10μm thickness. Blocks were cut and sections were placed on SuperFrost plus slides (Thermo Scientific, Cat# 630-0951) and were rapidly stained with H&E (quick wash with $H_2O$, 15sec in Hematoxylin (Cat# 6765002, Fisher Scientific), quick wash with $H_2O$, counterstain for 15sec with Eosin (Cat# 10483750, Fisher Scientific) and wash with $H_2O$ before mounting the slide) and evaluated for localization and extent of damage. This process was repeated until the optimal region was reached.

To assess optimal permeabilization conditions for Visium experiment, 8 adjacent sections were placed on the Visium Spatial Tissue Optimization Slide (Cat# 1000193, 10X Genomics) and were subjected to different permeabilization times. Optimal permeabilization time was 30mins and was used in the Visium experiment.

Once all sections were selected for Visium, blocks were placed in cryostat until equilibration along with the Visium spatial gene expression slide (10X Genomics, Cat# 1000187). Blocks were then aligned and a single 10μm section was placed on the Visium slide. After all sections were placed on the Visium slide, the downstream pipeline was continued in accordance with the manufacturer's guidelines (10x genomics, Visium Spatial Gene Expression Reagent Kits User Guide). The slides were imaged with Axio Imager.Z2 (upright) with Zeiss Axiocam 506 Color camera. Visium data was processed with SpaceRanger (v1.3.0) with the images manually aligned, with reference genome mm10. The filtered feature barcode matrix of each slide was imported to Seurat (v4.1.1) and merged.

### Pig mRNA library preparation and sequencing

Total RNA was extracted from 28 frozen pig samples in a paired manner (Remote and Infarct zones from the same animal were used to obtain optimal internal control [rhAgrin day 3 (n = 4), Saline day 3 (n = 3), rhAgrin day 28 (n = 4), Saline day 28 (n = 3)]. Following acquisition, frozen samples were crushed into fine powder using a mortar and pestle and further processed using miRNeasy Mini Kit (Qiagen, Cat# 217004) according to manufacturer's guidelines. Replicates of high RNA integrity (RIN), determined by TapeStation (Agilent) (RIN=8.2±0.1) were processed for RNA-seq library preparation.

RNA-seq libraries were prepared at the Crown Genomics institute of the Nancy and Stephen Grand Israel National Center for Personalized Medicine, Weizmann Institute of Science. Libraries were prepared using the INCPM-mRNA-seq protocol. Briefly, the polyA fraction (mRNA) was purified from 500ng of total input RNA followed by fragmentation and the generation of double-stranded cDNA. After Agencourt Ampure XP beads cleanup (Beckman Coulter), end repair, A base addition, adapter ligation, and PCR amplification steps were performed. Libraries were quantified by Qubit (Thermo fisher scientific) and TapeStation (Agilent). Sequencing libraries were constructed with barcodes to allow multiplexing. Sequencing was done on a NovaSeq600 instrument (Illumina) using an S1 100 cycles kit, allocating approximately ±35.7million single-end 100-base-pair reads per sample (single read sequencing). Fastq files for each sample were generated by the usage bcl2fastq v2.20.0.422.

## QUANTIFICATION AND STATISTICAL ANALYSIS

### Cell segmentation and quantification

In culture, following staining for LRP1, TIMP1, αSMA and DAPI, images were obtained with a Nikon Eclipse Ti2 fluorescent microscope with Nikon's NIS-Elements software and analyzed using QuPath v0.3.0 software.[99] Nuclei detection and cell segmentation was performed based on DAPI staining using a custom StarDist model followed by expansion by up to 5 um to approximate the full cell area.[100] A Random-trees object classifier was trained on multiple representative images to identify positive signals for each of the markers that were used for staining.

In MAMY mice tissue sections- following staining for GFP, TdTomato, cTnT or Gal-3, aSMA and DAPI, immunofluorescence images were obtained with a PANNORAMIC SCAN II slide scanner (3DHISTECH). and analyzed using QuPath v0.5.1 software. Cell segmentation was performed based on nuclei DAPI staining, using a custom StarDist model followed by expansion by up to 5 um to approximate the full cell area.[100]

For the multiple staining of GFP, TdTomato and cTnT, A Random-trees object classifier was trained on representative images to identify positive signals for each of the relevant cell-types (macrophage, myofibroblast and cardiomyocytes, respectively).

For the multiple staining of GFP and Gal-3, TdTomato and aSMA or GFP, TdTomato and EdU, an individual random-trees object classifier was trained using a multitude of representative images for each of the relevant cell-types/ state/ proliferation marker. These were merged into a single composite classifier that was used for all FOVs analyzed.

Heart sections from MAMY mice subjected to MI, Sham (MI), TAC, and Sham (TAC) were stained and imaged across three separate imaging sessions. For each session, an object classifier was individually calibrated using autofluorescence correction and no-primary-antibody controls.

## Double-reporter (MAMY) mice neighborhood analysis

MAMY mice heart sections were collected from either sham, MI or TAC operated animals. In MI, sections were collected on days: 2, 4, 7, 14, and 28 days following injury or sham operation (control). At day 28 following MI, a sample was also taken from the area near the suture to analyze the response to a foreign body (FBR). Additionally, samples were taken from mice 28 days post-TAC/ sham surgery. Sham operated animals were used as controls. In TAC, fibrotic area was divided based on histological presentation to: areas with interstitial fibrosis, replacement fibrosis, and the perivascular (PV) fibrosis area that was compared to sham PV regions as controls. The analysis focused on three cell types: cardiomyocytes, macrophages, and myofibroblasts identified based on cellular markers: cTnT, GFP and TdTomato, respectively. For each cell-type identified, the cell composition within a 50μm radius (100μm in diameter from the center of each cell; demonstrated in Figure S1C) was calculated using a custom-made code in Mathematica version 14. Cell composition data were averaged per field of view (FOV) and then further averaged across multiple FOVs for each animal. Both absolute counts and percentages of the three cell-types were analyzed. Statistical comparisons for MI samples were made between the time points and sham, as well as between day 7 and the other time points, with day 7 being the time point at which fibrosis enters the stationary phase and macrophage abundance reaches its maximum before declining back to approximate background levels. For TAC samples, either interstitial sham or perivascular sham were used as controls for either interstitial and replacement fibrosis, or PV fibrosis, respectively. Statistical tests used included the Mann-Whitney test with Bonferroni correction when appropriate.

For MAMY mice MI/ sham experiments, the number of biological replicates and number of cells used is as follows: Sham (n = 5; an average of 1487±118.5 SEM cells per replicate), day 2 (n = 4; an average of 1088±341.8 SEM cells per replicate), day 4 (n = 4; an average of 1027±352.6 SEM cells per replicate) day 7 (n = 4; an average of 1792±153.3 SEM cells per replicate), day 14 (n = 3; an average of 1471±275.2 SEM cells per replicate), day 28 infarct zone (n = 5; an average of 1627±248.7 SEM cells per replicate), day 28 suture (n = 5; an average of 1049±315.9 SEM cells per replicate).

For MAMY mice 28days post-TAC/ sham the number of biological replicates and number of cells used is as follows: TAC (n = 6), sham (n = 4). MAMY mice images were further analyzed based on the fibrotic histological presentation: Sham interstitial (an average of 1833±117.4 SEM cells per replicate), interstitial fibrosis (an average of 1422±156.7 SEM cells per replicate), replacement fibrosis (an average of 1474±286.5 SEM cells per replicate), PV sham (an average of 946.5±71.68 SEM cells per replicate), PV fibrosis (an average of 963.5±68.77 SEM cells per replicate).

For MAMY mice 4days post-MI, EdU pulse and aTIMP1/IgG control treatment experiment, the number of biological replicates and number of cells used is as follows: aTIMP1 treatment group includes 5 biological replicates; 3-6 tissue slices sampled per animal (Figure S9A); An average of 57,485±5974 SEM cells per biological replicate. IgG control group includes 4 biological replicates; 2-5 tissue slices sampled per animal (Figure S9A); An average of 86,453±24,456 SEM cells per biological replicate.

## Fibrosis analysis

Scarred tissues were quantified in a blinded manner, based on picro-sirius red staining of serial cardiac sections spanning through the entire heart (base to apex). In MI animals, we used colour deconvolution in ImageJ[101] to assess the average percentage of left ventricle fibrotic area. While in TAC animals we similarly applied colour deconvolution (ImageJ) to specific regions of different histological types of fibrosis: replacement, interstitial or perivascular. Mice transverse heart sections were also subjected to scar severity classification as previously described.[111] Each section was classified to one of three classes: transmural scar (scar that crossed the entire left ventricular wall), non-transmural scar (scars on a single or both sides of the ventricular wall that does not cross) or no scar. The number of sections for each category was represented as a percentage of total sections on the slide.

## Spatial transcriptomics analysis (Mouse)

To filter out poor quality spots, as well as spots localized in blood vessels and not overlapping cells, fractions of hemoglobin genes out of the total UMI reads was calculated for each spot. Spots with a hemoglobin fraction above 1% were removed. Next, using Loupe Browser (6.1.0), spots localized at the outer layers of the tissue (inside and out), were manually annotated as spots distanced up to 2 spots from tissue edges, and removed from analysis. Weakly expressed genes, with a total of less than 1000 UMI over all slides were ignored. UMI counts of the remaining spots were normalized by dividing each count to the spot's UMI total, resulting in fractions of UMIs. This dataset of ∼7000 spots over all slices and ∼4500 genes was used for differential gene expression. Counts were further normalized using log normalization, and non-linear dimension reduction was carried out using the UMAP algorithm using default parameters. Unbiased clustering was performed using the Mathematica function *'FindClusters'* with algorithm 'MeanShift' and sub option *'NeighborhoodRadius'* set to 0.45. To further visualize the clusters on the first 2 components of the projection, smooth kernel distribution was calculated ('SmoothDensityHistogram') and density contours were plotted. The clustering resulted in 3 clusters which were projected back on the slices from the 4 time points, and identified as the three distinct regions of remote, border and infarct zones. Gene signatures for the border and infarct zones were selected by calculating fold change (FC) expression of all genes and selecting genes with high FC relative to the expression in the remote zone (internal control per slice/time point). Mann-Whitney test with Bonferroni correction was used to calculate statistical significance. For the sake of visualization each slice was rotated so that the infarct zone median (defined by infarct zone spots) is positioned at the top, and log expression was rescaled (between 0 and 1) over all slices so comparison between time points is valid. Double positive spots (e.g., *Timp1*⁺/*Lrp1*⁺; Figure S8F) were identified as spots with expression higher than the median expression over all slices.

### Human spatial transcriptomics Analysis

Spatial transcriptomics slides (Visium 10X) of left ventricle tissues of patients with acute myocardial infarction and non-transplanted donor hearts were obtained from Kuppe et al.[62] Labels provided by the original authors describing the distinct pathological zones and time-points after human myocardial infarction were used in the analyses. Tissue samples labeled as border zone were excluded from any comparison, and remote zones tissues from myocardial infarction patients were grouped with donor tissues as controls. This led to the analysis of 25 distinct spatial transcriptomics slides.

Gene expression data from spatial transcriptomics slides was analyzed for filtered spots as provided by the original publication. In addition, we filtered out spots with less than 400 genes. Genes were filtered out if they were not expressed in at least 5 spots. Log-normalized data was generated with normalize_total and log1p functions from scanpy v1.9.1.[105] Deconvolution scores of cell-types per spot, previously computed with cell2location,[112] were used as provided by the original authors. In addition, slide annotations of molecular niches (i.e. group of spots with similar gene expression and cellular composition) and myofibroblasts state scores, computed with decoupleR,[113] were recovered from the original datasets.

Proportion of myeloid cells in each slide was calculated as a ratio between the sum of the abundance estimations of myeloid cells across all spots and the sum of the abundance of all cells across the slides. Abundance of myofibroblasts in a slide was quantified as the mean level of myofibroblast state scores within spots where at least we could quantify 10% of abundance of fibroblasts. For visualization purposes, myofibroblast scores were min-max normalized.

We quantified the spatial colocalization of the gene pair TIMP1-POSTN, using spatially-weighted cosine similarity as implemented in spatial_neighbors function of liana-py's v1.1.0.[76] Bandwidth and expression cut-offs were set to 150 and 0.1, respectively.

### Pig Bulk-mRNA sequencing analysis

Poly-A/T stretches and Illumina adapters were trimmed from the reads using cutadapt[114]; resulting reads shorter than 30bp were discarded. Reads for each sample were aligned independently to the *Sus-Scrofa* reference genome Sscrofa11 using STAR (2.7.3a),[107] supplied with gene annotations downloaded from Ensembl (release 103). The EndToEnd option was used and outFilter-MismatchNoverLmax was set to 0.04. The percentage of the reads that were aligned uniquely to the genome was 90%. Deduplication of reads with the same UMI was performed by the PICARD (2.22.4) MarkDuplicate tool (using the BARCODE_TAG parameter). Expression levels for each gene were quantified using htseq-count (version 0.11.2),[106] using the gtf above. Only uniquely mapped reads were used to determine the number of reads falling into each gene (intersection-strict mode). Differential analysis was performed using DESeq2 package (1.26.0)[108] with the betaPrior, cooksCutoff and independentFiltering parameters set to False. Raw p-values were adjusted for multiple testing using the procedure of Benjamini and Hochberg. Pipeline was run using snakemake.[115] Differentially expressed genes were determined by a p-adj of < 0.05 and absolute fold changes > 2 and max raw counts > 30.

Hierarchical clustering (distance: Pearson's dissimilarity, method: Ward.d) was performed, based on the 1000 most variable genes per condition [day (3 or 28) + treatment (agrin or saline)]. Clustering analysis was performed with Rstudio v3.6.1.

Unsupervised analysis was executed to explore the pattern of gene expression by clustering the genes based on 5961 genes (day 3) and 1079 genes (day 28). These genes were determined as differential expressed (DE) genes (upregulated or downregulated) in comparisons of infarct zones and remote zones in day 3 and 28 post-MI, in each of the treatment conditions (agrin and saline). Standardized, $\log_2$ normalized counts were used for the clustering analysis. Heat maps were constructed using Morpheus (https://software.broadinstitute.org/morpheus).

### Deconvolution of bulk-mRNA sequencing

To deconvolve our pig bulk-mRNA data, raw counts were normalized per repeat to calculate the fraction of each gene from the total counts per sample. Repeats were averaged per gene per condition (rhAgrin or Saline), per time point (Day 3 and 28), and region (remote and infarct zones). Fractions of myofibroblasts/macrophages per time point, condition and zone were calculated by taking the average of the expression of marker gene lists for each cell type (macrophages, myofibroblasts). Fold change of cell fractions in the infarct zone per condition (rhAgrin, Saline) compared to the internal control (remote zone) were then calculated per time point (day 3 and 28). Signature gene list for myofibroblasts (88 genes), and macrophages (102 genes) were modified from the re-analyzed scRNAseq data of Forte et al.[46] (Table S5). Finally, deconvolution of Pig bulk-mRNA data based on the mice pareto analysis archetypes (fibroblasts/macrophages, day 3 and 28 post-MI). This was achieved similarly to the macrophage/myofibroblast cell-type deconvolution signature but using the top enriched gene signature lists near the vertices (Table S6). The fractional expression of each gene in each archetype signature list was then summed to represent the fractional contribution of the archetype to the overall gene expression, and by proxy, its abundance within the bulk sample. The deconvolution algorithm estimated the proportions of each archetype within a bulk RNAseq sample. All calculations were done in Wolfram Mathematica 14.0.

### Analysis of mouse Bulk-mRNA sequencing

Raw counts matrix of GSE123863, generated by Wang et al.[116] was downloaded from the GEO database. Genes with maximum counts of 5 or lower were filtered out and then differential expression analysis was done DESeq2[108] with the '*betaPrior*', '*cooksCutoff*' and '*independentFiltering*' parameters set to False. Raw p-values were adjusted for multiple testing using the procedure of Benjamini and Hochberg. Genes with average(logFC)>=1, p.adj<0.05 and above 30 counts in at least one of the samples, were considered as differentially expressed.

### Analysis of mouse single-cell mRNA sequencing

Single-cell mRNA sequencing (scRNAseq) of cardiac interstitial cells from 7 different timepoints following MI[46] was downloaded from Array Express repository (E-MTAB-7895), and re-analyzed using Seurat V3.241. Cells were filtered according to 4 Median absolute deviation (MAD) below and above the median for each parameter ('nFeature_RNA', 'nCount_RNA', 'percent.mt') and sample. For % of mitochondrial reads, if the max % using the MAD is above 25% the last was used for filtering. We have also filtered out genes that appeared in less than 10 cells. Counts were normalized using log normalization. Next, non-linear dimension reduction was carried out using uniform manifold approximation and projection (UMAP) algorithm with the first 19 PCs of the data as input, using default parameters. Clusters were defined using 'FindNeighbors' and 'FindClusters' functions with 19 principal components (PCs) and resolution set to 0.5. Cluster annotation was done manually by visualizing the expression of known cell type specific marker genes and the top differentially expressed genes per cluster ('FindAllMarkers' with the parameters: logfc. threshold = 0.5, min.diff.pct = 0.5). Calculations of macrophage and myofibroblast gene-signature enrichment was performed by 'AddModuleScore'. UMAP figures were produced by 'FeaturePlot', and dot plots of enriched genes were produced by Seurat's 'DotPlot' function.

### Pareto analysis

We analyzed fibroblasts and macrophages scRNAseq data from Forte et al.[46] and performed Pareto Optimality analysis to infer the trade-offs and tasks that the cells specialize in. The raw data includes 17,202 genes, 13,100 macrophages and 20,746 fibroblasts from 7 different time points following MI. We next consider cells from day 0, day 3, and day 28 only. Before preprocessing the data, we randomly sampled 500 fibroblasts from each time point such that we end up analyzing 1500 fibroblasts from days 0, 3, and 28. For macrophages, we considered all 323 cells from day 0, 500 cells from day 3, and all 401 cells from day 28 amounting to 1224 cells. We then separately analyzed cells from day 0+day 3 and cells from day0+day28. We used 'Sanity' – a recently developed method to normalize scRNAseq data and to infer the transcription activity of the genes. Sanity is a bayesian procedure for normalizing scRNAseq data from first principles.[109] Following the Sanity normalization, we removed genes with low mean expression and low variation and considered genes with $\log_{10}$ mean expression larger than -14 and standard deviation larger than 0.03. This left 6187 genes for fibroblasts from days 0 and 3, 6007 genes for fibroblasts from days 0 and 28, 6692 genes for macrophages from days 0 and 3, and 7028 genes for macrophages from days 0 and 28. We next used the ParTi package in Matlab[65] to fit the data of each cell type population from each condition to a polytope. We find that fibroblasts from days 0 and 3 are distributed within a 5-vertex simplex (*p-value = 0.02*) and macrophages from days 0 and 3 are also best fit with a tradeoff among 5 tasks (*p-value = 0.001*). We find that fibroblasts from days 0 and 28 fill a tetrahedron with 4 archetypes (*p-value < 0.0001*) and macrophages from days 0 and 28 are also best described by a tetrahedron (*p-value = 0.02*). We next used ParTi to find enriched genes for all the archetypes to infer the tasks the cells specialize in. To infer specific pathway enrichment of either macrophage or fibroblasts tasks we used the enriched genes in each of the vertices. We specifically used pathway gene sets from biological processes of Gene Ontology (GO). All GO analysis was done using the Mathematica Package MathIOmica.[117]

To evaluate the distribution of the cells from day 0 versus day 3 or day 28 within the polytope, we sought to represent every cell as a convex combination of the archetype:

$G_c = \sum_{i=1}^{N} \theta_i G_i^*$ where $G_c$ represents the gene expression of cell c, $N$ is the number of archetypes, $G_i^*$ is the position of archetype i in gene expression space, and we are looking for $\theta_i$ such that $\sum_{i=1}^{N} \theta_i = 1$. We then categorize the cells based on the archetype they are closest to (or its largest $\theta_i$). To compare the distribution of the cells from day 0 and day 3 or day 28, we compared the proportion of the cells closest to each archetype from the total population per time point.

### NicheNet analysis

The re-analyzed scRNAseq by Forte et al.[46] was used for Ligand-Receptor analysis by the NicheNet algorithm.[68] The pairwise interaction between cell types (macrophages, fibroblasts, myofibroblasts) was calculated independently for each day (day 3 and 28 post-MI). For each cell type and for each day (day 3 and 28), genes of interest (GOIs) were defined genes that were upregulated between that day (3 or 28) and day zero and expressed in at least 25% of the cells in that cluster. Interaction score was calculated as the Pearson coefficient calculated by 'predict_ligand_activities' function, using the GOIs of that day and the genes that were expressed in the 'receiver' clusters as targets. Results were then refined by filtering against a list of growth factors (GO:0008083).

To analyze the interaction strength (weighted interaction) between macrophages, fibroblasts, and myofibroblasts at different time points (3 days or 28 days post-MI), we first estimated the fold change in cell type percentage relative to day 0 for each cell type. These fold changes were used to determine the sizes of the vertices in the graph. Next, the edge weights were calculated by multiplying three terms: (1) fold change of the sender cell ($C_S(t) = \%cells(t)/\%cells(0)$ - the number of "broadcasting antennas"), (2) the regulatory potential of the sent ligands as estimated from NicheNet (summed over all ligand-receptors pairs: $\sum_{l \in \{LR\,pairs\}} \rho_l^{S \to R}(t)$), and (3) fold change of the receiver cell (the number of "listening antennas": $C_R(t) = \%cells(t)/\%cells(0)$):

$$W_{S \to R}(t) = C_S(t)C_R(t) \sum_{l \in \{LR\,pairs\}} \rho_l^{S \to R}(t)$$

**Cell Systems**

$$C_i(t) = \frac{\%cells(t)}{\%cells(0)}, t \in \{day3, day28\}$$

All calculations and visualizations were performed using custom scripts in Mathematica 14.0.

### myofibroblast proliferation score

To estimate proliferation in myofibroblasts, we considered the expression of 98 mouse cell cycle genes.[118] We considered the fraction of RNA counts of these cell cycle genes out of the total RNA counts in the myofibroblast cells considering cells that have total count between $10^3$ and $10^4$ molecules. To determine the difference between the distributions of proliferation scores from the different time points, we used a Mann-Whitney test to test whether the data is sampled from different distributions.

To compare the myofibroblast proliferation score with Timp1 expression, we considered the average expression of Timp1 in myofibroblasts for each time point in the data. We then calculated the Pearson correlation between the average proliferation score and average Timp1 expression across the different time points. The Pearson correlation and p-value were calculated using the *PearsonCorrelationTest* function in Mathematica 14.0.

### *In-silico* perturbations of cold fibrosis

We computed the effect of various potential targets on the circuit dynamics by comparing the outcome of weakening three growth factor interactions: the two paracrine interactions between myofibroblasts and macrophages and the autocrine growth factor interaction within the myofibroblast population. We consider the following mathematical model based on the model presented in Adler et al.[19]:

$$\frac{dc_{12}}{dt} = \beta_{12}F - \alpha_{12}\frac{c_{12}}{k_{12}+c_{12}}M - \gamma c_{12} \qquad \text{(Equation 1)}$$

$$\frac{dc_{21}}{dt} = \beta_{11}F + \beta_{21}M - \alpha_{21}\frac{c_{21}}{k_{21}+c_{21}}F - \gamma c_{21} \qquad \text{(Equation 2)}$$

$$\frac{dF}{dt} = F\left(\lambda_1\frac{c_{21}}{k_{21}+c_{21}}\left(1 - \frac{F}{K}\right) - \mu_1\right) \qquad \text{(Equation 3)}$$

$$\frac{dM}{dt} = M\left(\lambda_2\frac{c_{12}}{k_{12}+c_{12}} - \mu_2\right) \qquad \text{(Equation 4)}$$

Where F, M are the concentration of fibroblasts and macrophages, and $c_{12}$, $c_{21}$ are the concentration of the two growth factors produced by the cells. We consider the following parameter values for the wild-type circuit:

Growth factor production rates: $\beta_{12} = 6,768\frac{molecules}{cell\ day}, \beta_{11} = 345,600\frac{molecules}{cell\ day}, \beta_{21} = 100,800\frac{molecules}{cell\ day}$

Growth factor endocytosis rates:

$$\alpha_{12} = 1,353,600\frac{molecules}{cell\ day}, \alpha_{21} = 734,400\frac{molecules}{cell\ day}$$

Growth factor degradation rate:

$$\gamma = 2\frac{1}{day}$$

Growth factor binding affinities:

$$k_{12} = 3.6x10^9 molecules, k_{21} = 1.3x10^7 molecules$$

Cell maximal proliferation rates:

$$\lambda_1 = 1.08\frac{1}{day}, \lambda_2 = 0.8\frac{1}{day}$$

Cell death rates:

$$\mu_1 = \mu_2 = 0.3\frac{1}{day}$$

Carrying capacity of fibroblasts: $K = 10^6 cells$.

Most of these parameter values were determined based on Zhou et al.,[20] with few exceptions as follows: We consider the production rate of the macrophage growth factor, $\beta_{12}$, to be 100-fold smaller in order to simulate a situation where the cold fibrosis state is a stable state (and there is no hot fibrosis). The macrophage growth factor binding affinity, $k_{12}$, is considered to be 200-fold larger in order to correct for spatial effects.

To consider the effect of perturbing the growth factor production rates on the circuit dynamics, we compare the circuit with the wild-type parameters to three scenarios. In the three scenarios, we keep all model parameters at the same value and only decrease one of the production rates $\beta_{12}, \beta_{21}$, or $\beta_{11}$ by 30%. To plot the separatrix, we numerically solve the final steady state levels of the cells (Equations 1–4) where we sample different initial conditions.

To exemplify the effect of weakening the fibroblast autocrine loop (Figure 4B), we considered the circuit (Equations 1–4) with the wild-type parameters where we consider the decrease in the maximal proliferation rate of fibroblasts, $\lambda_1$.

### Statistical analysis

All experiments were carried out with n $\geq$ 3 biological replicates [except for mice Visium data in Figure S8, produced from n = 1 per time point (day 1, 4, 7 and 14)]. Experimental groups were balanced for animal age, sex and weight. Different experimental groups were caged together and treated in the same way. Statistical analyses were carried out using the GraphPad Prism software (version 8.0.1) or by Mathematica (version 14.0). When comparing between two conditions, data were analyzed using a two-tailed Student's t-test or the Mann-Whitney test, as appropriate. When comparing more than two conditions, we employed ANOVA analysis with multiple comparisons or the Mann-Whitney test, as appropriate with Bonferroni multiple comparison correction. Statistical analysis of the multiple correlation matrix between echocardiography parameters and cardiac fibrosis was carried out using the Pearson correlation test. Pearson correlation was further calculated for the association between average *Timp1* expression in cardiac myofibroblast to average proliferation score in these cells. Specific details of statistical tests used for each figure panel can be found in the appropriate figure legend. Individual data points represent data derived from different biological repeats (unless mentioned differently in figure legends) to demonstrate the distribution of data across different experiments. Statistical analysis of bulk-mRNA sequencing data is described under the appropriate methods subheadings. Statistical analysis is derived from the biological repeats of an experiment. Measurements are reported as the mean and the error bars denote the S.E.M. throughout the study (unless mentioned differently in figure legends). P-values are presented on figures in the appropriate comparison location. Statistical analysis of Pareto archetype inference and enrichment of genes was done using the ParTi package[65] in Matlab (R2022a). Statistics of myofibroblast proliferation score distributions were computed using the function '*MannWhitneyTest*' in Mathematica 13.1.

