## [Document S2. Transparent peer review records for Miyara et al · Cell Systems]

Cold and hot fibrosis define clinically distinct cardiac pathologies

Shoval Miyara, Miri Adler, Kfir B. Umansky, Daniel Häußler, Elad Bassat, Yalin Divinsky, Jacob Elkahal, David Kain, Daria Lendengolts, Ricardo O. Ramirez Flores, Hanna Bueno-Levy, Ofra Golani, Tali Shalit, Michael Gershovits, Eviatar Weizman, Alexander Genzelinakh, Danielle M. Kimchi, Avraham Shakked, Lingling Zhang, Jingkui Wang, Andrea Baehr, Zachary Petrover, Rachel Sarig, Tatjana Dorn, Alessandra Moretti, Julio Saez-Rodriguez, Christian Kupatt, Elly M. Tanaka, Ruslan Medzhitov, Achim Krüger, Avi Mayo, Uri Alon, and Eldad Tzahor

Summary

Initial Submission: Received Oct 31, 2023
Preprint: doi.org/10.1101/2023.01.01.522422
Scientific editor: Bernadett Gaal, DPhil

First round of review: Number of reviewers: 3
3 confidential, 0 signed
Revision invited Feb 15, 2024
Major changes anticipated
Revision received Sept 28, 2024

Second round of review: Number of reviewers: 3
3 original, 0 new
3 confidential, 0 signed
Accepted Jan 27, 2025

This Transparent Peer Review Record is not systematically proofread, type-set, or edited. Special characters, formatting, and equations may fail to render properly. Standard procedural text within the editor's letters has been deleted for the sake of brevity, but all official correspondence specific to the manuscript has been preserved.

Editorial decision letter with reviewers' comments, first round of review

Dear Eldad,

I'm enclosing the comments that reviewers made on your paper, which I hope you will find useful and constructive. Please accept once again our apologies for the delay in the review process. As you'll see, the Reviewers express interest in the study, but they also have a number of criticisms and suggestions. Based on these comments, it seems premature to proceed with the paper in its current form; however, if it's possible to address the concerns raised with additional experiments and analysis, we'd be interested in considering a revised version of the manuscript.

As a matter of principle, I usually only invite a revision when I'm reasonably certain that the authors' work will align with the reviewers' concerns and produce a publishable manuscript. In the case of this manuscript it seems to me that a revision will require significant additional data and analysis and that it will be a 'heavy lift'. That said, Reviewer 3's report seem particularly constructive and I believe that the Reviewer appreciates and is supportive the goals of your work. You will see that they raise significant concerns that the data and analyses as they currently stand fall short of supporting the major conclusions and goals of the paper. I would advise that you focus your efforts on addressing these key concerns by Reviewer 3 (some of which overlap with specific concerns raised by Reviewer 2). While we ask that you also address Reviewer 1's concerns about previous works and better contextualise your model in the literature, I believe that if you are able to fully support your conclusions with the additional data and analyses then the novelty concerns will no longer be an issue.

As you address these concerns, it's important that you and I stay on the same page. I'm always happy to talk, either over email or on Zoom, if you'd like feedback about whether your efforts are moving the manuscript in a productive direction. Do note that we generally consider papers through only one major round of revision, so the revised manuscript would be either accepted or rejected based on the next round of comments we receive from the reviewers. If you have any questions or concerns, please let me know. More technical information and advice about resubmission can be found below my signature. Please read it carefully, as it can save substantial time and effort later.

STAR PROTOCOLS

Complement your primary research article by publishing a step-by-step procedure with STAR Protocols, an open-access peer-reviewed journal from Cell Press. STAR Protocols aims to make the daily work of the scientific researcher easier by providing complete, authoritative, and consistent instructions on how to conduct experiments. The primary criteria for publication in STAR Protocols is usability and reproducibility. You can check out their most recent protocols here. If you have any questions, please email starprotocols@cell.com.

I look forward to seeing your revised manuscript.

All the best,

Bernadett

Bernadett Gaal, DPhil
Editor-in-Chief, Cell Systems

Reviewers' comments:

Reviewer #1: In this manuscript, Miyara et al. combine mathematical modeling of a macrophage-fibroblast cell circuit with in vivo experiments of myocardial infarction and single-cell RNA sequencing to find that post-MI inflammation is not sustained and that fibrosis may be targeted with TIMP1 inhibitors.

The main strength of this manuscript is the range of experimental and computational methods that are brought together that are motivated from a simple mathematical model. Figure 4 is impressive on this point, as it goes from macrophage-fibroblast simulations that prioritize autocrine fibroblast signals to ligand-receptor analysis in scRNA-seq to in vitro and in vivo experiments perturbing TIMP1.

However, there are significant weaknesses that diminish enthusiasm, such as substantial overlap and lack of novelty compared to non-cited previous studies that have previously modeled these systems, established "cold fibrosis" post-MI, and previously characterized the fibrotic role of TIMP1.

Major Comments

Figure 1 performs timecourses of scRNA-seq and also spatial RNA sequencing. While this is nice, it is not clear that, compared to previous studies, these results lead new a new conclusion about whether macrophage numbers remain elevated post-MI ("reveals cold fibrosis" in the title).

It is unfortunate that the manuscript neglects the contributions of previous computational models of macrophage-fibroblast interactions after myocardial infarction. Including those that have used stability analysis or focused on TIMP-1/MMP activity, the emphasis of this study. See for example, PMIDs: 20522255 21545710 23134700 35862254.

Do the authors believe that their findings of cold fibrosis are consistent or inconsistent with those previous computational models? For example, PMID 21545710 published 12 years ago in BMC Systems Biology included gene and protein measurements of TIMP-1, and Figure 4 shows transient post-MI kinetics of macrophages and MMP9 expression but sustained proliferation of fibroblasts (cold fibrosis).

The manuscript lacks any discussion of previous computational model of intracellular signaling networks of these cells in the heart, including those that identified autocrine factors including TIMP1 that contribute to fibroblast activation and proliferation.

The main target emerging from single-cell RNA sequencing studies is TIMP1, which is validated in fibroblast culture. A number of previous studies have examined TIMPs including TIMP1 in cardiac fibrosis, for example: 28373589 or for a review,

<https://www.frontiersin.org/articles/10.3389/fmolb.2021.750626/full>

The authors suspect a non-MMP mechanism that would be more unique but this is not demonstrated.

A main stated goal of this paper is to test the concept of therapeutic modulation of hot and cold fibrosis (with and without sustained inflammation) after myocardial infarction. While they find that hot fibrosis does not occur post-MI, TIMP1 neutralizing antibody is tested post-MI in Figure 4m. The fibrosis phenotype is not well described quantitatively and no gross cardiac phenotyping or in vivo fibroblast proliferation was measured. This should probably be compared to previous studies finding a greater phenotype. It may be interesting to look at fibroblast proliferation genes in the scRNA-seq.

Reviewer #2: This manuscript describes the contribution of 2 circuits, hot and cold fibrosis, during the remodeling process post ischemic injury. By performing extensive bioinformatical and molecular analyses on both a murine and porcine model of ischemic injury the authors are able to show that the fibrotic response of the heart after damage develops into cold fibrosis, a process in which myofibroblasts activate myofibroblasts. Follow up experiments indicate TIMP1 to play a role in this process.

This is a very complete and clear study performed by key experts in the field. By using both a small and large animal model of post-ischemic injury the authors can confirm the presence and relevance of the 2 types of fibrosis-regulating circuits and are able to identify TIMP1 as regulator of the cold fibrosis process.

Some remaining questions are:

- Related to Figure 1, why was the Visium analysis not performed beyond day 14 (day 28 or 65), when all the pro-fibrotic macrophages have disappeared?
- Can the authors explain in a bit more detail how the different archetypes were determined in pigs?
- In Figure 3 the authors focus on day 0 and day 28 post-MI. Why not add day 3 also, where the hot fibrosis circuit is still more active?
- Based on Figure 4e one would think there is still a lot of communication between macrophages and myofibroblasts occurring 28 days after MI. Can the authors explain how this would match with the cold fibrosis circuit being dominant at this point?
- In Figure 4k the authors look at a change in Edu positive cells in response to TIMP1 regulation, but would it not be better to look at a change in myofibroblast identify or alphaSMA positive cells?
- The authors show data on a porcine IR study in which the pigs are treated with Agrin. Based on the data it seems like Agrin is blocking only the cold fibrosis process. Would this fit with the proposed function of Agrin?
- Why do we need the inhibition of TIMP1 if Agrin has a comparable effect on fibrosis (in addition to additional protective effects)?
- The in vivo study with a TIMP1 antibody needs more explanation and examination. Can the authors confirm a lowering in TIMP1 activity? How does the treatment effect cardiac function? What is the effect on the macrophage population in the heart after TIMP1 antibody treatment?
- Can the authors explain what eventually halts or slows down cold fibrosis in their view?

Reviewer #3: OVERVIEW

In this article, the authors build on their previous concept of hot and cold fibrosis by studying preclinical

models of cardiac fibrosis following myocardial infarction (MI) in mice and pigs, demonstrating that at later timepoints following MI these animals develop "cold" fibrosis which is characterised by myofibroblast (and not macrophage) accumulation. The proceed to suggest that this cold fibrosis is propagated by autocrine signalling by myofibroblasts and highlight TIMP-1 as a potential therapeutic target to disrupt this signalling and alleviate fibrosis. This is a very conceptually interesting manuscript and the idea of "hot" and "cold" fibrosis as pioneered by this group is potentially paradigm shifting for the fibrosis field and has important implications for chronic diseases in multiple different organs including heart, lung, liver, kidney and skin. One of the challenges is that the hot vs cold fibrosis concept is currently largely theoretical, with a lack of definitive data about the existence of these disease states in vivo, how they might differ and how they might be targeted therapeutically. The authors attempt to address some of these challenges in this manuscript. This is clearly important for the field, but in its current form there remain a number of aspects where the data presented do not adequately address these gaps in the knowledge. Specific comments below.

DATA & CODE AVAILABILITY

The primary data (spatial transcriptomics and bulk RNA-seq) as well as code will be available from public repositories and/or the authors Git page. Much of the data used (mouse scRNAseq) was previously published and is already available online.

REQUIRED MAJOR REVISIONS

1. Model of cold fibrosis: To me it remains unclear that the models presented truly reflect "cold" fibrosis rather than just a later timepoint in the post-MI "healing" response. There appears to be a spontaneous decline in the macrophages and myofibroblast niches in the mouse model (Fig 1M) whilst in the pig model both populations in the saline group (Fig 2G) are returning towards baseline. So is this really cold fibrosis or just late healing and how can you tell the difference? The inclusion of later timepoints would assist with the interpretation of this, as would alternative injury models.
2. Cellular heterogeneity: It is well recognised now that macrophages and myofibroblasts are heterogeneous in fibrosis with different cell types and states. When describing the kinetics of hot and cold fibrosis no account is taken of which type of macrophage or myofibroblast is present in the infarct and or border zone. This is likely to be important, as we would anticipate that scar-associated macrophages would populate the infarct/border zone and more resident macrophages would populate the remote zones. A more nuanced analysis of this (rather than simply using broad markers CD68 and Postn) is needed to fully understand the kinetics of these cell types. I would suggest that rather than simply using 2 markers to study spatial kinetics of these populations post-MI, that it would be more informative to use a deconvolution approach on the spatial transcriptomics data, applying properly annotated scRNAseq data to fully evaluate which macrophages and fibroblasts are in the infarct zone and how they change in composition over time.
3. No mention of selection biases in scRNAseq has been included. The authors rely on previously published scRNAseq data from a mouse MI model to enumerate changes in macrophages and myofibroblasts following MI. However, scRNAseq is notoriously unreliable for cell enumeration due to selection biases in cell isolation and sequencing for different populations. This is particularly relevant for

fibrotic diseases where cells embedded in dense ECM are likely to be most impacted. How can the authors be certain that all macrophage and myofibroblast populations have been captured and enumerated in the scRNAseq? It would be helpful to enumerate the myofibroblasts and macrophages at the different timepoints and cardiac regions using immunohistochemistry/immunofluorescence to confirm that the changes observed in scRNAseq are robust.

4. In Fig 1D-F, the authors confirm changes in macrophage number using proportional analysis using flow cytometry. This would be much more informative if absolute numbers of the different macrophage types per gram of tissue were quantitated rather than proportions. For example the changes in proportions in macrophage subpopulations (Fig 1D-F) might reflect changes in numbers of other CD45+ cells (e.g. T cells, neutrophils) rather than a genuine change in macrophage numbers.

5. Fig 4e-f: I'm slightly puzzled by this. The assertion from the manuscript is that D28 reflects a cold fibrosis state and the model of cold fibrosis suggests that this is dominated by autocrine myofibroblast signalling, with fewer macrophages in this niche. Would we not therefore expect there to be reduced macrophage-myofibroblast signalling in this cold fibrosis state when compared to a more hot/active fibrosis state? This is not reflected in these figures, where there appears to be no reduction in macrophage ligands in D28 when compared to D3. This calls into question whether this model truly does recapitulate cold fibrosis.

6. Spatial Transcriptomics numbers: Am I correct in saying that only 1 biological replicate per timepoint has been included in the spatial transcriptomics analysis? In my opinion this is insufficient to draw robust conclusions. Either further replicates need to be added or any findings need to be validated using other approaches (e.g. immunohistochemistry) in a more robust number of samples.

7. Pareto analysis (Fig 3): If I understand correctly, only D0 and D28 scRNAseq data is included here. However, based on the authors assertions this has entirely missed the peak in macrophage infiltration and myofibroblast activation as D28 represents cold fibrosis. Why not also include data from other timepoints here to properly dissect the kinetics of how the cellular phenotypes change over time? There may well then be more than 4 archetypes. The authors also state that macrophages have returned to a "quiescent homeostatic" state at Day 28 - however no analysis has been included here for the more activated macrophage state so how can this conclusion be drawn based on the data presented?

8. The TIMP-1 data is interesting but the role of TIMP-1 in fibrosis is not novel. The novelty in this context would be the role of autocrine TIMP-1 signalling in propagating cold fibrosis. However, the data presented does not confirm this. Firstly, in the in vivo TIMP-1 blockade experiments, the TIMP-1 blocking antibody is administered early (at D0 and D3) long before cold fibrosis is established. How can the authors be sure the antifibrotic effect of this is due to modulation of autocrine signalling/cold fibrosis? Administration of the blocking antibody at much later timepoints would have been helpful. Also, it is unclear how the TIMP-1 blockade mediates the antifibrotic effect. Is this truly via reduction of autocrine myofibroblast signalling or could it reflect increased matrix degradation by increased MMP activity? I would expect more detailed characterisation of this, ideally with molecular analyses (e.g. RNA-seq, scRNAseq or spatial transcriptomics) to prove that TIMP-1 inhibition disrupts the autocrine signalling in the infarct area.

9. Mechanisms on TIMP-1 action - for the in vitro work the authors discuss the potential role of LRP1 in mediating actions of TIMP-1 on myofibroblasts. However, no proof is provided for this. It would have been helpful to block or knockdown LRP1 in these cells to determine whether TIMP-1 acts via this receptor.

MINOR REVISIONS, SUGGESTIONS & COMMENTS

- a) I find fig 3d, e difficult to follow. Has the orientation of the archetype plots changed from Fig 3b,c? It would be better to keep this consistent so that the reader can understand where specific genes are expressed
- b) Figure 4D: I don't see much difference in TIMP-1 interactions between D3 and D28. Why focus in this as an autocrine signalling pathway for cold fibrosis when it is equally relevant in D3 data? Have the authors considered a differential nichenet (or other differential ligand-receptor analysis algorithm) to actually determine which ligand-receptor interactions are more enriched in the "cold" fibrosis situation compared to the earlier/inflammatory state.
- c) Typo: "pelted" should be "pelleted" in the methods
-

Authors' response to the reviewers' first round comments

Attached.

Editorial decision letter with reviewers' comments, second round of review

Dear Eldad,

I hope this email finds you well. The reviews are back on your revised manuscript and I've appended them below. You'll see that while the reviewers appreciate the additional work and your revisions, Reviewer 3 raises a few points that will need to be addressed.

Please do not hesitate to let me know if you have any questions!

I look forward to seeing your revised manuscript.

All the best,

Bernadett

Bernadett Gaál, DPhil
Editor-in-Chief, *Cell Systems*

Reviewer comments:

Reviewer #1: In this revision, the authors have added substantial new experiments to further characterize macrophage-myofibroblast dynamics post-MI in mice, pigs, and human samples. They also include new

in vitro and in vivo experiments showing the role of TIMP1 on fibroblast proliferation. As the authors note, the paper now contains 117 figure panels.

Comment 1

While the new experiments are very comprehensive, the response does not clarify why the authors disagree with the concern about conceptual novelty, which is that MI results in cold fibrosis.

Comments 2-4

The revised text now appropriately cites previous studies such as "Similar dynamics were observed by Jin et al. 25." and the authors acknowledged that those studies were consistent with cold fibrosis. This seems to contradict the authors' response to Comment 1.

Comment 5-6

The new anti-TIMP1 antibody experiment significantly bolsters the strength of the focus on TIMP1/fibroblast proliferation.

Reviewer #2: The authors were able to answer most questions raised by the reviewers and by doing so they generated a much improved version of the manuscript. There are no further questions and congratulations on such a insightful study.

Reviewer #3: In this revised manuscript the authors have significantly improved their study by conducting additional experiments and improving their analyses. This has largely addressed my previous queries, although I still have a few outstanding points for clarification:

1. The MAMY mouse model is a nice addition to the manuscript. However basic characterisation of the model is lacking and needed for proper interpretation. What is the specificity of the labelling in myofibroblasts or macrophages? Is there any off target labelling? What is the efficiency of labelling - if only a subpopulation of either cell type is labelled then the quantitation will be misleading. This should be quantitated by flow or immunofluorescence.
2. How does the MAMY model add anything over a simpler approach such as immunostaining of myofibroblasts and macrophages in the models? Could immunostaining be added for the ICR mice to confirm the spatial location of the macrophages and the fact that day 65 represents cold fibrosis?
3. The regenerative (P1 mice) and non-regenerative model (P8 mice) is not well described (Fig s7D). Please explain this better.
4. LRP1 signalling: I still think it would have been helpful to either block or knockdown LRP1 in fibroblasts to show this abrogates TIMP1 induced proliferation. As a minimum, the fact that the mechanism regulating this process is not defined should be included in the discussion.
5. Typo Fig S3J - should be "none" in the table
6. Typo - Toe "pinche" in the methods should be pinch

Authors' response to the reviewers' second round comments

Attached.

Editorial decision letter with reviewers' comments, third round of review

Dear Eldad,

I'm very pleased to let you know that the review of your revised manuscript is back, and only a few minor, editorially-guided changes are needed to move forward towards publication.

In addition to the final comments from the reviewer, which we ask you to address, I've made some requests within the "Editorial Notes" section below. Please review these notes along with the detailed formatting requirements listed in the Final Files Checklist. We've also put together this FAQ (click the Final Formatting Checks tab) for your convenience. Please ask any questions you may have, make any necessary changes to your manuscript files, and then upload your final files into Editorial Manager. Once we receive your formatted files, we will go through our formatting checks and let you know if further changes are needed.

Introducing new referencing style

To standardize the referencing style across Cell Press journals, starting from October 2022, we ask that all in-text citations be formatted as superscripted numbers (e.g. "Multiple reports support this observation.^{1,2}"). Moving away from the Harvard referencing style (e.g. Smith *et al.*, 2020) will improve author and reader experiences. All manuscripts accepted from now on must use **the superscript numbered Cell Press referencing style**. Make sure to use this numbered referencing style for all new and revised submissions as well. Switching is easy. Just use the updated CSL and EndNote referencing styles for Cell Press articles.

Below my signature, you'll find specific information about what to expect next regarding formatting checks and working with our Production Department after acceptance.

All the best,

Bernadett

Bernadett Gaál, DPhil
Editor-in-Chief, Cell Systems

Editorial Notes

Transparent Peer Review: Thank you for electing to make your manuscript's peer review process transparent. As part of our approach to Transparent Peer Review, we ask that you add the following sentence to the end of your abstract: "A record of this paper's Transparent Peer Review process is included in the Supplemental Information." Note that this **doesn't** count towards your 150 word total!

Also, if you've deposited your work on a preprint server, that's great! Please drop me a quick email with your preprint's DOI and I'll make sure it's properly credited within your Transparent Peer Review record.

Manuscript Text:

- House style disallows editorializing within the text (e.g. strikingly, surprisingly, importantly, etc.), especially the Results section. These terms are a distraction and they aren't needed—your excellent observations are certainly impactful enough to stand on their own. Please remove these words and others like them. “Notably” is suitably neutral to use once or twice if absolutely necessary.
- We don't allow “priority claims” (e.g. new, novel, etc.). For a discussion of why, read: <http://crosstalk.cell.com/blog/getting-priorities-right-with-novelty-claims>, <http://crosstalk.cell.com/blog/novel-insights-into-priority-claims>.

Figures and Legends:

Please look over your figures keeping the following in mind:

- When color scales are used, please define them, noting units or indicating "arbitrary units," and specify whether the scale is linear or log.
- Bar graphs are not acceptable because they obscure important information about the distributions of the underlying data. Please display individual points within your graphs unless their large number obscures the graph's interpretation. In that case, box-and-whisker plots are a good alternative.
- Please ensure that every time you have used a graph, you have defined "n's" specifically and listed statistical tests within your figure legend.
- When figures include micrographs, please ensure that scale bars are included and defined within the legend, montages are made obvious, and any digital adjustments (e.g. brightness) have been applied equally across the entire image in a manner that does not obscure characteristics of the original image (e.g. no "blown out" contrast). **Note that all accepted papers are screened for image irregularities, and if this advice is not followed, your paper will be flagged.**
- Please ensure that if you include representative images within your figures, a "representative of XXX individual cells"-type statement is made in the legend.
- Please ensure that all figures included in your point-by-point response to the reviewers' comments are present within the final version of the paper, either within the main text or within the Supplemental Information.

Resource Availability: Please note that Cell Press has recently changed the way it approaches "availability" statements for the sake of ease and clarity. Please revise your resource availability section as follows, noting that the examples used might not pertain to your study. Please note that **the Resource Availability section should immediately follow the Discussion section in the main manuscript.**

RESOURCE AVAILABILITY

Lead Contact: Further information and requests for resources and reagents should be directed to and will be fulfilled by the Lead Contact, Jane Doe (janedoe@qwerty.com).

Materials Availability: This study did not generate new materials. *-OR-* Plasmids generated in this study have been deposited at [Addgene, name and catalog number]. *-OR-* etc.

Data and Code Availability:

- **Source data statement** (described below)
- **Code statement** (described below)
- Any additional information required to reanalyze the data reported in this paper is available from the lead contact upon request.

Data and Code Availability statements **have three parts and each part must be present. Each part should be listed as a bullet point, as indicated above.**

Instructions for section 1: Data. The statements below may be used in any number or combination, but at least one must be present. They can be edited to suit your circumstance. ***Please ensure that all datatypes reported in your paper are represented in section 1. Please note that the statement "data are available in the main text or the supplementary materials" is acceptable only if you have provided the underlying data, i.e. data displayed in figures is not sufficient.*** For more information, please consult this list of standardized datatypes and repositories recommended by Cell Press.

- [Standardized datatype] data have been deposited at [datatype-specific repository] and are publicly available as of the date of publication. Accession numbers are listed in the key resources table.
- [Adjective] data have been deposited at [general-purpose repository] and are publicly available as of the date of publication. DOIs are listed in the key resources table.
- This paper analyzes existing, publicly available data. These accession numbers for the datasets are listed in the key resources table.
- [Adjective or all] data reported in this paper will be shared by the lead contact upon request.

Instructions for section 2: Code. The statements below may be used in any number or combination, but at least one must be present. They can be edited to suit your circumstance. ***If you are using GitHub, please follow the instructions here to archive a "version of record" of your GitHub repo at Zenodo, then report the resulting DOI. Additionally, please note that the Cell Systems strongly recommends that you also include an explicit reference to any scripts you may have used throughout your analysis or to generate your figures within section 2.***

- All original code has been deposited at [repository] and is publicly available as of the date of publication. DOIs are listed in the key resources table.

Instructions for section 3. Section 3 consists of the following statement: Any additional information required to reanalyze the data reported in this paper is available from the lead contact upon request.

STAR Methods:

STAR Methods follows a standardized structure. Please reorganize your experimental procedures to include these specific headings in the following order: EXPERIMENTAL MODEL AND SUBJECT DETAILS (when appropriate); METHOD DETAILS (required); QUANTIFICATION AND STATISTICAL ANALYSIS (when appropriate); ADDITIONAL RESOURCES (when appropriate). We're happy to be flexible about how each section is organized and encourage useful subheadings, but the required sections need to be there, with their headings. They should also be in the order listed. Please see the STAR Methods guide for more information or contact me for help.

Studies that use live vertebrates must perform their work in accordance with relevant institutional and national guidelines and regulations, and it is required that authors identify the committee approving the experiments and confirming that all experiments conform to the relevant regulatory standards. ***Please identify the committee(s) that approved the pig experiments.***

Please ensure that original code has been archived in a general purpose repository recommended by Cell Press and that its DOI is provided in the Software and Algorithms section of the Key Resources Table. If you've chosen to use GitHub, please follow the instructions here to archive a "version of record" of your GitHub repo at Zenodo, complete with a DOI. Thank you!

References included with the Key Resources Table should be incorporated into the main reference list.

Thank you!

Reviewer comments:

Reviewer #3: The authors have addressed all of my previous comments with this revised manuscript. I congratulate them on this really important body of work. I have a couple of remaining minor points:

- 1) Please consider the color schemes when comparing macrophages and myfibroblasts (e.g Fig 1B, C, H or Fig 2J). These are currently red and green and it would be challenge for readers with red green color blindness to interpret these figures.
- 2) I'd suggest referring to note S1 in the main text/results
- 3) Please review the interpretation of Fig S9E - they describe that TIMP1 blockade did not reduce a-SMA myofibroblasts, but there appears to be a tend to reduction in these data. Would you not expect the a-SMA population to be reduced with TIMP1 inhibition?

Authors' response to the reviewers' third round comments

Attached.

Executive Summary of the Revision (for all 3 reviewers)

Dear reviewers,

We sincerely appreciate the reviewers' time and efforts to evaluate our work and offer invaluable feedback. Based on their suggestions we have implemented significant changes over the past 8 months. These revisions include **1)** Introduction of a novel macrophage-myofibroblast double-reporter mouse line, **2)** Evaluation of an extended 28 days post-myocardial-infarction (MI) timepoint that establishes cold fibrosis, **3)** Demonstration of cold fibrosis as a conserved outcome following MI in humans, **4)** Expansion of our cold/hot fibrosis analyses to include two forms of chronic cardiac injuries resulting in hot fibrosis, **5)** Analysis of the growth factor activity of TIMP1 using recombinant human proteins and primary non-human-primate cardiac fibroblasts, and **6)** Additional *in-vivo* studies that reinforces that TIMP1 acts as an inducer of cardiac myofibroblasts proliferation. We have also clarified the text, added references, and discussed previous work. The revised manuscript now contains 117 figure panels, of which 56 (48%) were added during these revisions. These numerous additions make the revised manuscript much more powerful, rigorous, and clear.

Below is a concise summary of the major additional experiments incorporated in the revised version of our manuscript, explained in detail where we have addressed the reviewers' comments point-by-point.

- 1. A novel macrophage-myofibroblast double reporter mouse line:** To directly quantify macrophages and myofibroblasts *in situ*, we developed a double reporter mouse line (MAMY): Postn^{MCM/+}Rosa26^{TdTomto/+}Cx3cr1^{GFP/+}. We performed myocardial infarction

(MI) and analyzed tissue sections from multiple time points to demonstrate the macrophage-myofibroblast cellular dynamics following injury. Following MI, both macrophages and myofibroblasts peak in the infarct zone, but only myofibroblasts persist. This result is consistent with cold fibrosis- a state of abundant myofibroblasts supporting their own growth. We validate cold fibrosis by numerous approaches and also in pigs and human cardiac samples of patients after MI (detailed in point #3).

2. **Extended 28 days post-MI time point:** Using our MAMY mice, we have extended our analysis to 28 days post-MI. These data reinforce a stabilization of the cold fibrosis state beyond 14 days.
3. **Establishment of cold fibrosis in human MI:** We analyzed human Visum spatial transcriptomics data by *Kuppe, et al. 2022*. These data include uninjured and MI-injured heart samples from early (up to 15 days post-MI) and late (30 days+) time points following MI. We conclude that cold fibrosis is a conserved outcome of MI in humans.
4. **Characterization of chronic injury models:** We explored two modes of chronic cardiac injuries using our MAMY mice- left ventricular pressure overload, induced by transverse aortic constriction (TAC) and a foreign body response to the surgical suture used to ligate the left anterior descending (LAD) artery in MI. MAMY mice underwent TAC and were tracked for 28 days. TAC induced a chronic injury to the heart, predicted by our model to form a hot fibrosis stable point ¹. We quantified macrophages and myofibroblasts in three distinct histological patterns of fibrosis created by TAC: **Replacement fibrosis**- patched fibrosis lacking cardiomyocytes, **Interstitial fibrosis**- fibrosis occurring between cardiomyocytes, and **Perivascular fibrosis**². Myofibroblast numbers were elevated in all types of fibrosis. Macrophage numbers were increased in both interstitial and replacement

fibrosis, suggesting a hot fibrosis classification.

The MAMY mouse line enabled us to view the cells in their correct spatial context. At 28 days post-MI, we noticed that while macrophages were markedly reduced in the infarct zone, these cells were observed in the immediate perimeter of the suture alongside abundant myofibroblasts. This spatial coexistence of myofibroblasts and macrophages suggests a local hot fibrosis region around the suture, which resembles a foreign body response. We conclude that following permanent LAD ligation-induced MI, the injured heart acquires spatially distinct cold and hot fibrosis features.

5. ***In-vitro* stimulation of primary non-human-primate cardiac fibroblasts (NHP-cFs) with recombinant human TIMP1 proteins (rhTIMP1):** To test whether TIMP1 acts as a growth factor and whether this activity is conserved in primates, we incubated NHP-cFs with either the full wild type rhTIMP1 (rhWT-TIMP1) or with a fragment of TIMP1 lacking its C-terminus (rhN-TIMP1) together with EdU to label the proliferative capacity of cells. We find that both fragments (rhWT-TIMP1 and rhN-TIMP1) promote NHP-cFs proliferation to a similar extent, indicating that the growth factor activity of TIMP1 is conserved in primate/human.
6. ***In-vivo* neutralization of TIMP1 following MI reduces myofibroblast proliferation:** To test the effect of TIMP1 on cardiac myofibroblast proliferation following injury, we performed MI on MAMY mice and treated them with either IgG control or anti-TIMP1 neutralizing antibodies. We pulsed the mice with EdU to label proliferating cells and extracted hearts for analysis 24 hours later (4 days post-MI). Anti-TIMP1 treatment significantly reduced the amount of proliferating myofibroblasts (TdTomato+EdU+). This treatment did not attenuate macrophage abundance (GFP+ cells), proliferation

(GFP+EdU+) or phenotype- measured by Galectin-3^{3,4}. We conclude that TIMP1 promotes the proliferation of cardiac myofibroblasts *in vivo*.

Altogether our major revision efforts described above strongly enhanced the impact of this study and we are confident that it is now suitable for publication in Cell Systems.

Below is our point-by-point response, embedded within the reviewers' comments (in blue).

Reviewer #1: In this manuscript, Miyara et al. combine mathematical modeling of a macrophage-fibroblast cell circuit with in vivo experiments of myocardial infarction and single-cell RNA sequencing to find that post-MI inflammation is not sustained and that fibrosis may be targeted with TIMP1 inhibitors. The main strength of this manuscript is the range of experimental and computational methods that are brought together that are motivated from a simple mathematical model. Figure 4 is impressive on this point, as it goes from macrophage-fibroblast simulations that prioritize autocrine fibroblast signals to ligand-receptor analysis in scRNA-seq to in vitro and in vivo experiments perturbing TIMP1.

We thank the reviewer for this endorsement.

However, there are significant weaknesses that diminish enthusiasm, such as substantial overlap and lack of novelty compared to non-cited previous studies that have previously modeled these systems, established "cold fibrosis" post-MI, and previously characterized the fibrotic role of TIMP1.

Major Comments

Comment 1: Figure 1 performs time courses of scRNA-seq and also spatial RNA sequencing. While this is nice, it is not clear that, compared to previous studies, these results lead to a new conclusion about whether macrophage numbers remain elevated post-MI ("reveals cold fibrosis" in the title).

Response: We thank the reviewer for this comment. We now added MI experiments using a new

macrophage-myofibroblast double reporter mouse line (Postn^{MCM/+}; Rosa26^{TdTomto/+}; Cx3cr1^{GFP/+}) that allows accurate quantification of myofibroblast and macrophage numbers *in situ* (new Figure 1C, new Figure S1C). This mouse model overcomes technical concerns of cell extraction artifacts in scRNAseq data. The new data provides strong evidence for cold fibrosis following MI and adds deeper spatial insights (new Figure 1H-K). We also added a hot fibrosis model following chronic cardiac injury in mice by TAC (new Figure 2). The side-by-side comparison between acute and chronic cardiac injuries reinforces the distinction between the two states of fibrosis (cold/hot) predicted by the model. We further added a myofibroblast/macrophage ratio measurement in all fibrotic states tested (new Figure S2A). We find that while in cold fibrosis (MI) myofibroblasts are highly dominant in numbers over macrophages (ratio of 6:1), in hot fibrosis (TAC) there is a cell type ratio of about 1:1 between myofibroblasts to macrophages. We believe that this analysis adds a quantitative dimension to the distinction between cold and hot fibrosis.

We kindly disagree with the reviewer about the conceptual novelty of our study compared to previous publications, which we now discuss throughout the revised manuscript.

Comment 2: It is unfortunate that the manuscript neglects the contributions of previous computational models of macrophage-fibroblast interactions after myocardial infarction. Including those that have used stability analysis or focused on TIMP-1/MMP activity, the emphasis of this study. See for example, PMIDs: 20522255 21545710 23134700 35862254.

Response: We now cite these computational models of LV remodeling in the introduction, results, and discussion sections.

In the introduction, we write (page 4): “MI occurs when a blood vessel that feeds the heart is blocked, causing ischemia and cardiomyocyte cell death^{22,23}. Subsequently, immune cells rapidly invade the injured zone leading to the local expansion and activation of myofibroblasts. This results in changes in the microenvironment dictated by the release of various cytokines, growth factors and matrix proteins, as described mathematically by Lindsey and colleagues^{18,24,25}. The injured non-regenerative myocardium is replaced by a fibrotic scar over a period of several weeks, a process called cardiac remodeling, that gradually leads to reduced heart function, morbidity and mortality^{26–28}.”

In the results, we write:

On page 6: “We first explored the dynamics of macrophage and myofibroblast populations in mice after MI. To do so, we analyzed the single cell mRNA-sequencing (scRNAseq) data by *Forte, et al.*⁴⁶ and quantified the fold change (FC) ratio shift in abundance of macrophages and myofibroblasts at seven time points. Macrophage and myofibroblast abundance first increased, peaking at days 3-7. At day 14, macrophages decreased leaving persistent myofibroblasts at days 14 and 28 (Figure 1B, Figure S1A-B)⁴⁷. At 28 days post-MI the scar is mature, hence, this trajectory is consistent with cold fibrosis. Similar dynamics were observed by Jin et al.²⁵.”

On page 14: “We focused on *Timp1* since it was strongly expressed at both early (day 3) and late phases of cold fibrosis (day 28), consistent with its increased levels in serum samples following MI in mice²⁴. TIMP1 has been studied in relation to fibrosis mostly in the context of its inhibition of Matrix Metalloproteinases (MMP) activity as a pro-fibrotic and an anti-fibrotic factor^{24,69,70}, but was also reported as a growth factor in scleroderma fibroblasts⁷¹. “

In the discussion, we write (page 18):

“Previous mathematical modeling have demonstrated the important role of myofibroblast-macrophage interactions in driving and sustaining fibrosis^{18,19,24}.”

Comment 3: Do the authors believe that their findings of cold fibrosis are consistent or inconsistent with those previous computational models? For example, PMID 21545710 published 12 years ago in BMC Systems Biology included gene and protein measurements of TIMP-1, and Figure 4 shows transient post-MI kinetics of macrophages and MMP9 expression but sustained proliferation of fibroblasts (cold fibrosis).

Response: We thank the reviewer for pointing out this study which we were not aware of. It is consistent with cold fibrosis.

We now cite it in the results:

On page 6: “We first explored the dynamics of macrophage and myofibroblast populations in mice after MI. To do so, we analyzed the single cell mRNA-sequencing (scRNAseq) data by *Forte, et al.*⁴⁶ and quantified the fold change (FC) ratio shift in abundance of macrophages and myofibroblasts at seven time points. Macrophage and myofibroblast abundance first increased, peaking at days 3-7. At day 14, macrophages decreased leaving persistent myofibroblasts at days 14 and 28 (Figure 1B, Figure S1A-B)⁴⁷. At 28 days post-MI the scar is mature, hence, this trajectory is consistent with cold fibrosis. Similar dynamics were observed by Jin et al.²⁵.”

We also cite this study when we mention high TIMP1 expression after MI in the results (page 14): “We focused on *Timpl* since it was strongly expressed at both early (day 3) and late phases of cold fibrosis (day 28), consistent with its increased levels in serum samples following MI in

mice²⁴.”

Comment 4: The manuscript lacks any discussion of previous computational model of intracellular signaling networks of these cells in the heart, including those that identified autocrine factors including TIMP1 that contribute to fibroblast activation and proliferation.

Response: We have now added citations of previous modeling work as detailed above (response to comments #2-3).

Comment 5: The main target emerging from single-cell RNA sequencing studies is TIMP1, which is validated in fibroblast culture. A number of previous studies have examined TIMPs including TIMP1 in cardiac fibrosis, for example: 28373589 or for a review, <https://www.frontiersin.org/articles/10.3389/fmolb.2021.750626/full>

The authors suspect a non-MMP mechanism that would be more unique but this is not demonstrated.

Response: We thank the reviewer for this comment that helped us to clarify the discussion. We focused on the effect of TIMP1 on cardiac myofibroblast proliferation. It is plausible that this effect is due to the direct effect of TIMP1 via its cell surface receptor(s), or an indirect effect due to its anti-MMP function, or by other means. Importantly, we elaborated the TIMP1 proliferative effect *in vivo* by performing a new experiment: anti-TIMP1 antibody treatment *in vivo* following MI. We observed significantly reduced myofibroblast proliferation post-MI, using our MAMY mice following EdU pulse (new Figure 5J, new Figure S9A). At day 4 post-MI, inhibition of TIMP1 caused a decrease in the fraction of proliferative myofibroblasts (TdTomato+EdU+) but did not affect macrophage abundance (GFP+ cells), proliferation (GFP+EdU+), and phenotype

(marked by GFP+Galectin-3+), or myofibroblast activation (aSMA marker) (new Figure 5K-L, new Figure S9B-F). We conclude that TIMP1 has a mitogenic effect on cardiac myofibroblasts. We addressed this point in the revised discussion (page 20).

As written above, we also added an experiment using recombinant human TIMP1 (rhWT-TIMP1) and rhN-TIMP1 (lacking its C-terminal domain) fragments to better map its MoA. These experiments boost the impact of our study showing the effect of TIMP1 *in vitro*, in non-human-primate cardiac fibroblast cultures.

As written in the Discussion, we can not rule out that MMP-inhibition does not play a role in the mediated proliferative effect of cardiac myofibroblasts.

Comment 6: A main stated goal of this paper is to test the concept of therapeutic modulation of hot and cold fibrosis (with and without sustained inflammation) after myocardial infarction. While they find that hot fibrosis does not occur post-MI, TIMP1 neutralizing antibody is tested post-MI in Figure 4m. The fibrosis phenotype is not well described quantitatively and no gross cardiac phenotyping or *in vivo* fibroblast proliferation was measured. This should probably be compared to previous studies finding a greater phenotype.

Response: we have now added an *in vivo* myofibroblast proliferation experiment using anti-TIMP1 (aTIMP1) antibodies as detailed above (response to comment #5). We also added to the revised manuscript functional cardiac measurements using 2D echocardiography on treated (aTIMP1), and IgG control mice at days 0, 3, 7, and 35 post-MI (new Figure S9G-I). We found no significant difference in ejection fraction (EF) or fractional shortening (FS). We note a low correlation between echocardiography parameters and cardiac fibrosis phenotyping by sirius red

measurements. Thus in mice, these echocardiography-based cardiac function parameters may not be a sensitive measure of injury severity.

Taken together, in this paper we revealed a direct role of TIMP1 as a growth factor of cardiac myofibroblasts that affect cold fibrosis following MI. We further acknowledge that TIMP1 is not the only factor affecting myofibroblast proliferation following injury.

Comment 7: It may be interesting to look at fibroblast proliferation genes in the scRNA-seq.

Response: We thank the reviewer for this comment which helped us strengthen the connection between *Timpl* expression and myofibroblast proliferation. We have now looked at myofibroblast proliferation genes in a scRNAseq data set by *Forte, et al. 2020*⁵ (new Figure S7C). We find a strong correlation between *Timpl* expression in myofibroblasts and the average proliferation score of myofibroblasts (*Pearson-r = 0.94, p = 0.015*).

Reviewer #2: This manuscript describes the contribution of 2 circuits, hot and cold fibrosis, during the remodeling process post ischemic injury. By performing extensive bioinformatical and molecular analyses on both a murine and porcine model of ischemic injury the authors are able to show that the fibrotic response of the heart after damage develops into cold fibrosis, a process in which myofibroblasts activate myofibroblasts. Follow up experiments indicate TIMP1 to play a role in this process.

This is a very complete and clear study performed by key experts in the field. By using both a small and large animal model of post-ischemic injury the authors can confirm the presence and relevance of the 2 types of fibrosis-regulating circuits and are able to identify TIMP1 as

regulator of the cold fibrosis process.

We thank the reviewer for this endorsement

Some remaining questions are:

Comment 1: Related to Figure 1, why was the Visium analysis not performed beyond day 14 (day 28 or 65), when all the pro-fibrotic macrophages have disappeared?

Response: We removed the Visium analysis from the main Figures (new Figure S8). To address fibrosis at day 28 post-MI, we introduced a new macrophage-myofibroblast double reporter mouse line ($\text{Postn}^{\text{MCM}/+}; \text{Rosa26}^{\text{TdTomto}/+}; \text{Cx3cr1}^{\text{GFP}/+}$) that allows accurate quantification of the two cell types *in situ*. We analyzed both myofibroblast and macrophage numbers at day 28 and found strong evidence for cold fibrosis in the infarct zone (new Figure 1C-K). We further extended our findings to human myocardial infarction using a previously published Visium dataset by *Kuppe, et al. 2022*⁶, and showed that cold fibrosis is a conserved outcome following MI in humans (new Figure 3A-B).

Comment 2: Can the authors explain in a bit more detail how the different archetypes were determined in pigs?

Response: We have now clarified this in the methods. We used marker genes from the mouse archetypes to perform deconvolution of the bulk RNAseq pig data.

In the revised methods we now write (page 43): “Finally, deconvolution of Pig bulk-mRNA data based on the mice pareto analysis archetypes (fibroblasts/macrophages, day 3 and 28 post-MI). This was achieved similarly to the macrophage/myofibroblast cell-type deconvolution signature,

but using the top enriched gene signature lists near the vertices (Table S6). The fractional expression of each gene in the each archetype signature list was then summed to represent the fractional contribution of the archetype to the overall gene expression, and by proxy, its abundance within the bulk sample. The deconvolution algorithm estimated the proportions of each archetype within a bulk RNAseq sample. All calculations were done in Wolfram Mathematica 14.0.”

Comment 3: In Figure 3 the authors focus on day 0 and day 28 post-MI. Why not add day 3 also, where the hot fibrosis circuit is still more active?

Response: We thank the reviewer for this comment which helped us to add more data for the phase of maximum growth and infiltration at day 3^{7,8} (new Figure 4, new Figure S4, new Figure S6, new Video S1). The data is very revealing. Indeed, new archetypes are found. Macrophages on day 3 show two new archetypes associated with IL1 signaling and respiratory burst. Myofibroblasts show a new archetype associated with IL1 and IL12 signaling as well as contractile functions (*Acta2* expressing). This analysis highlights the activation and de-activation of macrophages and the temporal sequence of myofibroblast maturation. These Day 3 data thus show that the initial MI period not only creates a niche for high macrophage and myofibroblast numbers but is also enriched with immune-associated functions in both cell types.

Comment 4: Based on Figure 4e one would think there is still a lot of communication between macrophages and myofibroblasts occurring 28 days after MI. Can the authors explain how this would match with the cold fibrosis circuit being dominant at this point?

Response: We thank the reviewer for this comment which helped us to improve our analysis. In

the previous analysis, we normalized cell-cell interactions to the absolute number of total sender cells. This normalization was vulnerable to sampling biases of cells from the mature scar. We now analyze the data as a fold change of the fraction of each cell type relative to its baseline levels (day 0), thereby allowing better control for sampling biases (detailed in methods; page 46). The revised analysis shows an increase in myofibroblast autocrine signaling at day 28 and a decrease in all interactions associated with macrophages. This is consistent with cold fibrosis (new Figure 5E-F).

Comment 5: In Figure 4k the authors look at a change in Edu positive cells in response to TIMP1 regulation, but would it not be better to look at a change in myofibroblast identify or alphaSMA positive cells?

Response: We thank the reviewer for this comment which allowed us to perform the suggested experiment *in vivo* (new Figure 5K, new Figure S9E). We used our new MAMY mice (Postn^{MCM/+}; Rosa26^{TdTomto/+}; Cx3cr1^{GFP/+}) to visualize myofibroblasts and macrophages following MI and a 24hrs EdU pulse. We find that TIMP1 inhibition by neutralizing antibodies reduces the fraction of proliferating myofibroblasts (TdTomato+EdU+) but does not affect their activation state based on aSMA expression (new Figure S9E).

Comment 6: The authors show data on a porcine IR study in which the pigs are treated with Agrin. Based on the data it seems like Agrin is blocking only the cold fibrosis process. Would this fit with the proposed function of Agrin?

Response: We thank the reviewer for this comment. To answer this we re-analyzed scRNAseq data from our recently published paper *Zhang, et al. 2024⁹*. This dataset contains cardiac

interstitial cells collected 4 days post-MI and Agrin/PBS treatments. We applied NicheNet and focused on all interactions between macrophages, fibroblasts, and myofibroblasts in Agrin vs PBS treated hearts (Figure below). Our analysis suggests that Agrin acts via an alternative, TIMP-independent mechanism(s). Specifically, Agrin seems to affect primarily fibroblast but not myofibroblast signaling. This is in line with our recent publication showing that Agrin induces regenerative senescence in cardiac fibroblasts, facilitating at least one of its pro-regenerative functions. We believe this result is beyond the scope of the paper and thus we did not include it.

NicheNet analysis suggests that fibroblast signaling is reduced by Agrin treatment. **A.** Top- Experimental outline of the scRNAseq experiment performed by *Zhang, et al. 2024*. Myocardial infarction (MI) was induced by left anterior descending (LAD) in adult mouse hearts. These were subsequently treated with recombinant rat-Agrin, or PBS (control) immediately following MI by intramuscular injection and extracted by day 4. Bottom- UMAP analysis of cardiac cells corresponding to general cell type clusters. **B.** cell-type clusters were separated by treatment groups (agrin/PBS). **C.** Schematic representation of NicheNet analysis to identify ligand-receptor interactions between fibroblasts, macrophages and myofibroblasts from either agrin or PBS-treated hearts. **D.** Weighted interactions between fibroblasts, macrophage, and myofibroblasts including a fold change (FC) ratio of interactions.

Comment 7: Why do we need the inhibition of TIMP1 if Agrin has a comparable effect on fibrosis (in addition to additional protective effects)?

Response: We believe that our TIMP1 findings are important in supporting the model in which the autocrine loop of myofibroblasts is essential for cold fibrosis. The present autocrine inhibition strategy may be applied to other organs (albeit with other molecules). This was recently demonstrated in liver cirrhosis, where inhibiting a different autocrine factor of stellate cells (fibrosis-related stellate cells which are the organ-specific myofibroblast-like cells) reduced advanced fibrosis in a NASH mouse model ¹⁰.

Based on our new NicheNet analysis, in response to Comment #6, and a recently published manuscript⁹, we suggest that Agrin works through alternative mechanisms compared to TIMP1, mostly affecting cardiac fibroblasts, but not myofibroblasts. Finally, if Agrin and TIMP1 work by different mechanisms, they might synergize. This would be interesting to explore in future studies.

Comment 8: The *in vivo* study with a TIMP1 antibody needs more explanation and examination. Can the authors confirm a lowering in TIMP1 activity? How does the treatment affect cardiac function? What is the effect on the macrophage population in the heart after TIMP1 antibody treatment?

Response: We thank the reviewer for this comment. We now repeated the *in vivo* TIMP1 inhibition experiment using our new double-reporter mouse line (Postn^{MCM/+}; Rosa26^{TdTomato/+}; Cx3cr1^{GFP/+}). We found that on day 4 post-MI, TIMP1 inhibition reduces the fraction of proliferating myofibroblasts (TdTomato+EdU+), but not monocyte/macrophage infiltration

(GFP+), macrophage proliferation (GFP+EdU+) or macrophage state (based on Gal3+- a marker for pro-fibrotic macrophage^{3,4}). Moreover, this treatment did not affect myofibroblast activation (TdTomato+aSMA+) (new Figure 5J-L, new Figure S9B-F). These data provide evidence that TIMP1 has a specific growth factor activity for cardiac myofibroblasts *in vivo*. It further suggests that TIMP1 inhibition does not affect macrophage abundance, proliferative capacity, or state.

We also added to the revised manuscript data on heart function using echocardiography data on treated and control mice at days 0, 3, 7, and 35 post-MI (new Figure S9G-I). We find no significant difference in ejection fraction (EF) or fractional shortening (FS) following TIMP1 inhibition. We note that there is a low correlation between echocardiography parameters and cardiac fibrosis by sirius red measurements, and thus in mice, echocardiography-based cardiac function parameters may not be a sensitive measure of injury severity.

Comment 9: Can the authors explain what eventually halts or slows down cold fibrosis in their view?

Response: Great question! It is true that once the scar is mature, it stabilizes and transitions to a maintenance phase in which the extensive ECM growth phase is halted. We must admit that we do not fully understand the dynamics of the scar maintenance phase.

In relation to the halting or slowing down of cold fibrosis by inhibiting the myofibroblast autocrine loop: We believe that inhibiting the myofibroblast autocrine loop causes fewer divisions, and requires a higher level of myofibroblasts to sustain themselves, a level which is not attained. As a result, myofibroblasts numbers begin to decline. As a result, less ECM is formed. Other myofibroblast effects such as inhibiting ECM degradation enzymes are eliminated once myofibroblasts are eliminated, further contributing to the loss of scar tissue.

On a mathematical level, reducing the autocrine signaling strength pushes the unstable fixed point in Figure 1A towards the cold fibrosis fixed point - because more myofibroblasts are needed to sustain themselves as the autocrine loop gets weaker. At a critical inhibition level and beyond, the two points collide, and the fixed point disappears. This leaves only the healing state as a possible steady state (demonstrated in Figure 5B).

Reviewer#3 : OVERVIEW

In this article, the authors build on their previous concept of hot and cold fibrosis by studying preclinical models of cardiac fibrosis following myocardial infarction (MI) in mice and pigs, demonstrating that at later timepoints following MI these animals develop "cold" fibrosis which is characterised by myofibroblast (and not macrophage) accumulation. They proceed to suggest that this cold fibrosis is propagated by autocrine signalling by myofibroblasts and highlight TIMP-1 as a potential therapeutic target to disrupt this signalling and alleviate fibrosis. This is a very conceptually interesting manuscript and the idea of "hot" and "cold" fibrosis as pioneered by this group is potentially paradigm shifting for the fibrosis field and has important implications for chronic diseases in multiple different organs including heart, lung, liver, kidney and skin. One of the challenges is that the hot vs cold fibrosis concept is currently largely theoretical, with a lack of definitive data about the existence of these disease states in vivo, how they might differ and how they might be targeted therapeutically. The authors attempt to address some of these challenges in this manuscript. This is clearly important for the field, but in its current form there remain a number of aspects where the data presented do not adequately address these gaps in the

knowledge. Specific comments below.

We thank the reviewer for this endorsement.

DATA & CODE AVAILABILTY

The primary data (spatial transcriptomics and bulk RNA-seq) as well as code will be available from public repositories and/or the authors Git page. Much of the data used (mouse scRNAseq) was previously published and is already available online.

REQUIRED MAJOR REVISIONS

Comment 1: Model of cold fibrosis: To me it remains unclear that the models presented truly reflect "cold" fibrosis rather than just a later timepoint in the post-MI "healing" response. There appears to be a spontaneous decline in the macrophages and myofibroblast niches in the mouse model (Fig 1M) whilst in the pig model both populations in the saline group (Fig 2G) are returning towards baseline. So is this really cold fibrosis or just late healing and how can you tell the difference? The inclusion of later timepoints would assist with the interpretation of this, as would alternative injury models.

Response: As this is an important point, we have taken multiple analyses and experimental steps to fully answer it:

(1) We have improved our analyses which were vulnerable to single-cell sampling biases in scRNAseq. These may have caused the apparent myofibroblast decline as the scar matures. To do so we normalized the abundance (measured by % of total cells) to the baseline timepoint (as fold change ratio) (new Figure 1B). This better demonstrates the dramatic increase and persistence of myofibroblasts, compared to the rise and fall trajectory of macrophages.

(2) We have tested MI injury using a new double reporter mouse line ($Postn^{MCM/+}$; $Rosa26^{TdTtomato/+}$; $Cx3cr1^{GFP/+}$) and included an extended 28 days post-MI timepoint. These mice allow us to quantify in great detail the myofibroblast and macrophage populations *in situ*, bypassing the need for scRNAseq or Visium. We find that following MI, while both macrophage ($GFP+$ cells) and myofibroblasts ($TdTomato+$ cells) peak at day 7, only myofibroblasts persist, while macrophage numbers decline significantly by day 28. This provides strong evidence for cold fibrosis following MI in mice (new Figure 1C-K).

(3) We further added two new injury models to the manuscript - heart failure due to pressure overload (induced by transverse aortic constriction-TAC) and a foreign body response (FBR) to the surgical suture that remains within the myocardium for 28 days post-LAD ligation (new Figure 1H-K, new Figure 2, new Figure S1C, new Figure S2A). Both chronic injury modes revealed sustained macrophages and myofibroblasts numbers, in line with hot fibrosis. This side-by-side comparison between acute (MI) and chronic (TAC, FBR) injuries further allows us to distinguish between the two states of fibrosis (cold/hot) predicted by the model.

(4) We next added a myofibroblast/macrophage ratio measurement in all fibrotic states tested (new Figure S2A). We find that while in cold fibrosis (MI) myofibroblasts are highly dominant in numbers over macrophages (ratio of 6:1), in hot fibrosis states (TAC, FBR) there is a cell type ratio of about 1:1 between myofibroblast to macrophage. We believe that this analysis adds a quantitative dimension to the distinction between cold and hot fibrosis.

Comment 2: Cellular heterogeneity: It is well recognised now that macrophages and myofibroblasts are heterogeneous in fibrosis with different cell types and states. When describing the kinetics of hot and cold fibrosis no account is taken of which type of macrophage or

myofibroblast is present in the infarct and or border zone. This is likely to be important, as we would anticipate that scar-associated macrophages would populate the infarct/border zone and more resident macrophages would populate the remote zones. A more nuanced analysis of this (rather than simply using broad markers CD68 and Postn) is needed to fully understand the kinetics of these cell types. I would suggest that rather than simply using 2 markers to study spatial kinetics of these populations post-MI, that it would be more informative to use a deconvolution approach on the spatial transcriptomics data, applying properly annotated scRNAseq data to fully evaluate which macrophages and fibroblasts are in the infarct zone and how they change in composition over time.

Response: To address this comment, we have expanded our Pareto analysis of cellular heterogeneity from scRNAseq data of fibroblasts (including myofibroblasts) and macrophages. We find that on day 3, both populations display unique immune-related tasks (new Figure 4A-F, new Video S1). Fibroblast archetype #4, associated with immune-regulating functions (i.e. IL12 and IL1 signaling) was populated almost exclusively by day 3 cells (Day3/Day0 Fold Change; FC > 60), suggesting the acquisition of a new fibroblast function soon after injury (Figure 4C). Concomitant with fibroblasts, macrophages acquired new tasks within the first 3 days. Archetype #1 (Day3/Day0 FC > 90) showed recruited monocyte markers such as *Plac8¹¹*, and archetype #2 (Day3/Day0 FC > 20) showed resident macrophage marker genes (i.e. *Folr2¹²*). Both archetypes were enriched with GO terms associated with angiogenesis and a phagocytic response to injury (Figure 4G, Figure S4B). On day 28, while macrophage numbers significantly decline (new Figure 1H, J), they also return to their homeostatic gene expression states (new Figure 4J-L, new Video S2). Unlike macrophages, myofibroblasts further remained in high numbers and acquired a persistent ECM-producing phenotype, associated with sustained fibrosis (new Figure

4G-I, new Video S2). We conclude that the initial transient MI period with both high macrophages and myofibroblasts is enriched in immune interacting functions. In contrast, late cold fibrosis has homeostatic macrophage phenotypes and a distinct fibroblast ECM-producing state.

Due to reviewer #3, comment #6, we have moved the Visium data from the main text and figures, to serve as supportive data in Figure S8, recognizing that it had only one repeat per timepoint.

Comment 3: No mention of selection biases in scRNAseq has been included. The authors rely on previously published scRNAseq data from a mouse MI model to enumerate changes in macrophages and myofibroblasts following MI. However, scRNAseq is notoriously unreliable for cell enumeration due to selection biases in cell isolation and sequencing for different populations. This is particularly relevant for fibrotic diseases where cells embedded in dense ECM are likely to be most impacted. How can the authors be certain that all macrophage and myofibroblast populations have been captured and enumerated in the scRNAseq? It would be helpful to enumerate the myofibroblasts and macrophages at the different timepoints and cardiac regions using immunohistochemistry/immunofluorescence to confirm that the changes observed in scRNAseq are robust.

Response: We thank the reviewer for this comment. As mentioned above, in the new manuscript we present a new double reporter mouse line ($Postn^{MCM/+}; Rosa26^{TdTomo/+}; Cx3cr1^{GFP/+}$) that marks myofibroblasts with TdTomato (Tamoxifen inducible Periostin-MerCreMer cassette), and macrophages with GFP (driven by the *Cx3cr1* promoter). This mouse reporter line provides greater confidence that both myofibroblasts and macrophages can be visualized *in situ*, and it bypasses sampling bias concerns with the scRNAseq experiments (new Figure 1C-K).

Comment 4: In Fig 1D-F, the authors confirm changes in macrophage number using proportional analysis using flow cytometry. This would be much more informative if absolute numbers of the different macrophage types per gram of tissue were quantitated rather than proportions. For example the changes in proportions in macrophage subpopulations (Fig 1D-F) might reflect changes in numbers of other CD45+ cells (e.g. T cells, neutrophils) rather than a genuine change in macrophage numbers.

Response: We used the double reporter mice to obtain alternative quantitative cell counts that are more reliable, and thus we believe this comment is addressed in the new Figure 1.

Comment 5: Fig 4e-f: I'm slightly puzzled by this. The assertion from the manuscript is that D28 reflects a cold fibrosis state and the model of cold fibrosis suggests that this is dominated by autocrine myofibroblast signalling, with fewer macrophages in this niche. Would we not therefore expect there to be reduced macrophage-myofibroblast signalling in this cold fibrosis state when compared to a more hot/active fibrosis state? This is not reflected in these figures, where there appears to be no reduction in macrophage ligands in D28 when compared to D3. This calls into question whether this model truly does recapitulate cold fibrosis.

Response: We thank the reviewer for this comment which helped us to improve our analysis. Previously we normalized the number of interactions to the total number of sender cells (per timepoint). This normalization was vulnerable to single-cell sampling biases from the mature scar. We now analyze the data as a fold change of the fraction of each cell type, thereby controlling for sampling biases (see new Methods). The revised analysis shows an increase in myofibroblast autocrine signaling at D28 compared to D3, and a decrease in all interactions associated with macrophages. This is consistent with cold fibrosis (new Figure 5E-F).

Comment 6: Spatial Transcriptomics numbers: Am I correct in saying that only 1 biological replicate per timepoint has been included in the spatial transcriptomics analysis? In my opinion this is insufficient to draw robust conclusions. Either further replicates need to be added or any findings need to be validated using other approaches (e.g. immunohistochemistry) in a more robust number of samples.

Response: We have removed the mouse Visium analysis from the main figures and only used it as supportive supplementary data.

Comment 7: Pareto analysis (Fig 3): If I understand correctly, only D0 and D28 scRNAseq data is included here. However, based on the authors assertions this has entirely missed the peak in macrophage infiltration and myofibroblast activation as D28 represents cold fibrosis. Why not also include data from other timepoints here to properly dissect the kinetics of how the cellular phenotypes change over time? There may well then be more than 4 archetypes. The authors also state that macrophages have returned to a "quiescent homeostatic" state at Day 28 however no analysis has been included here for the more activated macrophage state so how can this conclusion be drawn based on the data presented?

Response: We thank the reviewer for this comment which helped us add data from day 3 to the Pareto analysis (new Figure 4A-G, new Figure S4, new Figure S6A, Video S1). Indeed, as the reviewer anticipated, we find more than 4 archetypes for both macrophages and fibroblasts (new Figure 4A, E). Hence, this analysis identified new archetypes for both cell types that were not present in our previous analysis. Macrophages at D3 show two new archetypes associated with IL1 signaling and respiratory burst. Fibroblasts show a new archetype associated with IL1 and IL12 signaling as well as contractile functions. Taken together, our updated Pareto analysis

highlights on one hand the activation and the return to quiescence of macrophages following MI and, on the other hand, the temporal sequence of fibroblast maturation, which is irreversible. The D3 time point shows immune-associated gene expression in both cell types likely suggesting a strong crosstalk between the two cell types.

Comment 8: The TIMP-1 data is interesting but the role of TIMP-1 in fibrosis is not novel. The novelty in this context would be the role of autocrine TIMP-1 signalling in propagating cold fibrosis. However, the data presented does not confirm this. Firstly, in the *in vivo* TIMP-1 blockade experiments, the TIMP-1 blocking antibody is administered early (at D0 and D3) long before cold fibrosis is established. How can the authors be sure the antifibrotic effect of this is due to modulation of autocrine signalling/cold fibrosis? Administration of the blocking antibody at much later timepoints would have been helpful. Also, it is unclear how the TIMP-1 blockade mediates the antifibrotic effect. Is this truly via reduction of autocrine myofibroblast signalling or could it reflect increased matrix degradation by increased MMP activity? I would expect more detailed characterisation of this, ideally with molecular analyses (e.g. RNA-seq, scRNAseq or spatial transcriptomics) to prove that TIMP-1 inhibition disrupts the autocrine signalling in the infarct area.

Response: We thank the reviewer for this comment that prompted us to perform a new *in vivo* experiment with the double reporter mice to assess myofibroblast proliferation following TIMP1 inhibition in the MI setting. We performed MI on adult mice and injected IgG control or anti-TIMP1 neutralizing antibodies directly to the injured area (via intramyocardial injection), and 3 days following MI using retro-orbital injection. We report that TIMP1 inhibition with an antibody following-MI reduces the fraction of proliferating myofibroblasts (TdTomato+EdU+)

(new Figure 5J-L). TIMP1 inhibition did not attenuate the total macrophage abundance/infiltration (GFP+), macrophage proliferation (GFP+EdU+), or macrophage state (based on Gal3+- a marker for pro-fibrotic macrophage^{3,4}. (new Figure S9C-F). This *in vivo* experiment is consistent with the role of TIMP1 as a myofibroblast growth factor.

While in this study we chose early intervention of TIMP1 inhibition, it would be interesting in future studies to attempt and treat the mature scar at much later time points (day 28+) although the challenge of targeting the heart at this stage is huge. Indeed a recent study was able to show a reduction in mature NASH fibrosis after late timepoint intervention (delivery was through IV injection and did not require surgery)¹⁰. Unfortunately, drug delivery to the injured myocardium after the first few days remains a challenge for the cardiology field. A late delivery approach suggested by the reviewer is very interesting, but the technical challenge of re-opening the chest cavity 28 days after the initial injury, and then assaying the mice weeks later, presents considerable obstacles. Moreover, we are not sure how this secondary systemic inflammation, achieved by re-opening the chest cavity, will affect the overall injury trajectory. We thus believe that this experiment is beyond the scope of the current manuscript. We cite the work by Wang, et al. and discuss it in the discussion.

Aiming to reinforce the autocrine function of Timp1, we have attempted to develop a conditional knockout approach to delete *Timp1* in myofibroblasts, over the last three years. Unfortunately, our findings were mostly inconclusive due to variable Cre efficiency reasons. In addition, we have collaborated with the group of Prof. Achim Kruger (TUM) who is an expert in the TIMP1 field¹³⁻¹⁶. Together with his group, we tested the proliferative capacity of 2 distinct forms of the human TIMP1 protein, on primary non-human-primate (NHP) cardiac fibroblasts (cFs).

We used the full wildtype recombinant protein (rhWT-TIMP1) as well as a recombinant fragment lacking TIMP1's C-Terminus (rhN-TIMP1). The N-terminus domain of TIMP1 is known to facilitate its anti-proteolytic activity but was also shown to perform direct receptor signaling functions¹³. Importantly, the Kruger lab was able to show in their hands that both rhWT-TIMP1 and rhN-TIMP1 can induce NHP-cFs proliferation (Figure 5J), thus providing a crucial experimental repeat, further extending our conclusions regarding the growth factor activity of TIMP1 from mice to primates/humans.

Comment 9: Mechanisms on TIMP-1 action - for the in vitro work the authors discuss the potential role of LRP1 in mediating actions of TIMP-1 on myofibroblasts. However, no proof is provided for this. It would have been helpful to block or knockdown LRP1 in these cells to determine whether TIMP-1 acts via this receptor.

Response: We have conducted new experiments and added new data and discussion (page 20) to the paper. Our data indicates that myofibroblasts but not other cells in the MI heart strongly produce *Timp1* and at least three of its receptors: *Lrp1*, *Cd63*, and *App* (new Figure S7A, G). We further show that the most (60%) of cardiac myofibroblasts co-express TIMP1 and LRP1 (new Figure 5G-H). We next tested the growth factor capacity of 2 forms of the human TIMP1 protein: wild-type recombinant TIMP1 (rhWT-TIMP1), and a recombinant fragment of TIMP1, lacking its C-Terminal domain (rhN-TIMP1). The C-terminal domain of TIMP1 was previously shown to mediate pro-inflammatory cytokine signaling through CD63 and APP^{13,14}. The N-terminal domain of TIMP1 is known to facilitate its anti-proteolytic functions but was also shown to promote signaling through CD74¹⁷. The direct binding domain of TIMP1 to LRP1 is yet to be characterized^{15,16}. We treated primary non-human-primate cardiac fibroblasts (NHP-cFs) with

both protein forms and found that rhN-TIMP1 was able to increase NHP-cFs proliferation to a similar extent as rhWT-TIMP1 (new Figure 5J). This result suggests that TIMP1 does not signal via CD63 or APP^{13,14}. Taken together, our data suggest that the growth factor activity of TIMP1 is mediated through its N-terminal domain and LRP1. We acknowledge that while these results are informative, our investigation of the possible TIMP1-LRP1 signaling was not exhaustive. Additional studies, including LRP1 knockdown experiments, would be important to fully validate the suggested mode of action. However, these experiments are beyond the scope of the current study.

MINOR REVISIONS, SUGGESTIONS & COMMENTS

Comment 10: I find fig 3d, e difficult to follow. Has the orientation of the archetype plots changed from Fig 3b,c? It would be better to keep this consistent so that the reader can understand where specific genes are expressed

Response: We have now added supplementary videos (Video S1-2) that capture the entire 3D space of any polytope presented.

Comment 11: Figure 4D: I don't see much difference in TIMP-1 interactions between D3 and D28. Why focus in this as an autocrine signalling pathway for cold fibrosis when it is equally relevant in D3 data? Have the authors considered a differential nichenet (or other differential ligand-receptor analysis algorithm) to actually determine which ligand-receptor interactions are more enriched in the "cold" fibrosis situation compared to the earlier/inflammatory state.

Response: As mentioned above, we have improved our analysis, and this bias is now corrected.

Comment 12: Typo: "pelted" should be "pelleted" in the methods

Response: Fixed.

In summary, the reviewers' comments helped us to add a large amount of new data and analysis. We present a new double reporter mouse that allows better quantification of the cell types *in situ*. We added two new injury models: TAC, and foreign body response. We strengthened the evidence for cold fibrosis in MI using the double reporter mouse and further showed that cold fibrosis is conserved in human MI. We establish *in vivo* that TIMP1 inhibition reduces myofibroblast proliferation, and further show its conserved growth factor activity in non-human-primate cardiac fibroblasts. We improved the analysis and clarity throughout. We believe the revised paper is much more rigorous, comprehensive, and clear.

References

1. Adler, M., Mayo, A., Zhou, X., Franklin, R.A., Meizlish, M.L., Medzhitov, R., Kallenberger, S.M., and Alon, U. (2020). Principles of Cell Circuits for Tissue Repair and Fibrosis. *Iscience* 23, 100841. <https://doi.org/10.1016/j.isci.2020.100841>.
2. Frangiannis, N.G. (2020). Cardiac fibrosis. *Cardiovasc. Res.* 117, 1450–1488. <https://doi.org/10.1093/cvr/cvaa324>.
3. Meijers, W.C., Velde, A.R. van der, Pascual-Figal, D.A., and Boer, R.A. de (2015). Galectin-3 and post-myocardial infarction cardiac remodeling. *Eur. J. Pharmacol.* 763, 115–121. <https://doi.org/10.1016/j.ejphar.2015.06.025>.
4. Boer, R.A. de, Yu, L., and Veldhuisen, D.J. van (2010). Galectin-3 in Cardiac Remodeling and Heart Failure. *Curr. Hear. Fail. Rep.* 7, 1–8. <https://doi.org/10.1007/s11897-010-0004-x>.
5. Forte, E., Skelly, D.A., Chen, M., Daigle, S., Morelli, K.A., Hon, O., Philip, V.M., Costa, M.W., Rosenthal, N.A., and Furtado, M.B. (2020). Dynamic Interstitial Cell Response during Myocardial Infarction Predicts Resilience to Rupture in Genetically Diverse Mice. *Cell Reports* 30, 3149-3163.e6. <https://doi.org/10.1016/j.celrep.2020.02.008>.
6. Kuppe, C., Flores, R.O.R., Li, Z., Hayat, S., Levinson, R.T., Liao, X., Hannani, M.T., Tanevski, J., Wünnemann, F., Nagai, J.S., et al. (2022). Spatial multi-omic map of human myocardial infarction. *Nature* 608, 766–777. <https://doi.org/10.1038/s41586-022-05060-x>.
7. Nahrendorf, M., Swirski, F.K., Aikawa, E., Stangenberg, L., Wurdinger, T., Figueiredo, J.-L., Libby, P., Weissleder, R., and Pittet, M.J. (2007). The healing myocardium sequentially mobilizes two monocyte subsets with divergent and complementary functions. *J. Exp. Med.* 204, 3037–3047. <https://doi.org/10.1084/jem.20070885>.
8. Fu, X., Khalil, H., Kanisicak, O., Boyer, J.G., Vagnozzi, R.J., Maliken, B.D., Sargent, M.A., Prasad, V., Valiente-Alandi, I., Blaxall, B.C., et al. (2018). Specialized fibroblast differentiated states underlie scar formation in the infarcted mouse heart. *J Clin Invest* 128, 2127–2143. <https://doi.org/10.1172/jci98215>.
9. Zhang, L., Elkahal, J., Wang, T., Rimmer, R., Genzelinakh, A., Bassat, E., Wang, J., Perez, D., Kain, D., Lendengolts, D., et al. (2024). *Egr1* regulates regenerative senescence and cardiac repair. *Nat. Cardiovasc. Res.*, 1–18. <https://doi.org/10.1038/s44161-024-00493-1>.
10. Wang, S., Li, K., Pickholz, E., Dobie, R., Matchett, K.P., Henderson, N.C., Carrico, C., Driver, I., Jensen, M.B., Chen, L., et al. (2023). An autocrine signaling circuit in hepatic stellate cells underlies advanced fibrosis in nonalcoholic steatohepatitis. *Sci Transl Med* 15, eadd3949. <https://doi.org/10.1126/scitranslmed.add3949>.
11. Dick, S.A., Macklin, J.A., Nejat, S., Momen, A., Clemente-Casares, X., Althagafi, M.G., Chen, J., Kantores, C., Hosseinzadeh, S., Aronoff, L., et al. (2019). Self-renewing resident cardiac macrophages limit adverse remodeling following myocardial infarction. *Nat Immunol* 20, 29–39. <https://doi.org/10.1038/s41590-018-0272-2>.
12. Dick, S.A., Wong, A., Hamidzada, H., Nejat, S., Nechanitzky, R., Vohra, S., Mueller, B., Zaman, R., Kantores, C., Aronoff, L., et al. (2022). Three tissue resident macrophage subsets coexist across organs with conserved origins and life cycles. *Sci. Immunol.* 7, eabf7777. <https://doi.org/10.1126/sciimmunol.abf7777>.
13. Schoeps, B., Eckfeld, C., Prokopchuk, O., Böttcher, J., Häußler, D., Steiger, K., Demir, I.E., Knolle,

P., Soehnlein, O., Jenne, D.E., et al. (2021). TIMP1 Triggers Neutrophil Extracellular Trap Formation in Pancreatic Cancer. *Cancer Res.* 81, 3568–3579. <https://doi.org/10.1158/0008-5472.can-20-4125>.

14. Eckfeld, C., Schoeps, B., Häußler, D., Frädlich, J., Bayerl, F., Böttcher, J.P., Knolle, P., Heisz, S., Prokopchuk, O., Hauner, H., et al. (2023). TIMP-1 is a novel ligand of Amyloid Precursor Protein and triggers a proinflammatory phenotype in human monocytes. *J. Cell Biol.* 222, e202206095. <https://doi.org/10.1083/jcb.202206095>.

15. Schoeps, B., Frädlich, J., and Krüger, A. (2023). Cut loose TIMP-1: an emerging cytokine in inflammation. *Trends Cell Biol.* 33, 413–426. <https://doi.org/10.1016/j.tcb.2022.08.005>.

16. Grünwald, B., Schoeps, B., and Krüger, A. (2019). Recognizing the Molecular Multifunctionality and Interactome of TIMP-1. *Trends Cell Biol* 29, 6–19. <https://doi.org/10.1016/j.tcb.2018.08.006>.

17. Schoeps, B., Eckfeld, C., Flüter, L., Keppler, S., Mishra, R., Knolle, P., Bayerl, F., Böttcher, J., Hermann, C.D., Häußler, D., et al. (2021). Identification of invariant chain CD74 as a functional receptor of tissue inhibitor of metalloproteinases-1 (TIMP-1). *J. Biol. Chem.* 297, 101072. <https://doi.org/10.1016/j.jbc.2021.101072>.

Below is our point-by-point response (in blue) to reviewers' comments.

Reviewer #1: In this revision, the authors have added substantial new experiments to further characterize macrophage-myofibroblast dynamics post-MI in mice, pigs, and human samples. They also include new in vitro and in vivo experiments showing the role of TIMP1 on fibroblast proliferation. As the authors note, the paper now contains 117 figure panels.

We thank the reviewer for this endorsement.

Comment 1: While the new experiments are very comprehensive, the response does not clarify why the authors disagree with the concern about conceptual novelty, which is that MI results in cold fibrosis.

We agree with the reviewer's suggestions - the revised text cites previous work "Similar dynamics were observed by Jin et al.²⁵.", and acknowledges that those studies were consistent with cold fibrosis.

Comments 2-4: The revised text now appropriately cites previous studies such as "Similar dynamics were observed by Jin et al. 25." and the authors acknowledged that those studies were consistent with cold fibrosis. This seems to contradict the authors' response to Comment 1.

We agree and believe that the revised manuscript provides proper credit to previous work.

Comment 5-6: The new anti-TIMP1 antibody experiment significantly bolsters the strength of the focus on TIMP1/fibroblast proliferation.

We thank the reviewer for this endorsement.

Reviewer #2: The authors were able to answer most questions raised by the reviewers and by doing so they generated a much improved version of the manuscript. There are no further questions and congratulations on such a insightful study.

We thank the reviewer for this endorsement and all meaningful comments and suggestions.

Reviewer #3: In this revised manuscript the authors have significantly improved their study by conducting additional experiments and improving their analyses. This has largely addressed my previous queries, although I still have a few outstanding points for clarification:

We thank the reviewer for this endorsement.

Comment #1: The MAMY mouse model is a nice addition to the manuscript. However basic characterisation of the model is lacking and needed for proper interpretation. What is the specificity of the labelling in myofibroblasts or macrophages? Is there any off target labelling? What is the efficiency of labelling - if only a subpopulation of either cell type is labelled then the quantitation will be misleading. This should be quantitated by flow or immunofluorescence.

Response: We thank the reviewer for endorsing the importance of adding the MAMY mouse line to the paper. The MAMY model integrates two well-established genetic systems targeting cardiac myofibroblasts and all monocyte/macrophage cell fates and has proven specificity and efficiency, as extensively validated in prior studies^{1,2,3-12,13-17}. Numerous publications have

utilized these lines individually to address their specificity, labeling efficacy, and potential off-target effects.

Our data demonstrates the specificity of the MAMY models with no TdTomato+ myofibroblasts in the uninjured heart across multiple experimental repeats (Figure 1H, K; Figure 2J, M). Using Pareto analysis, we also reveal that *Postn* is specifically expressed by archetype #4 (activated fibroblasts), a population associated with fibrosis and enriched in 28-day post-MI cells (Figure 4H-I). Additional data in our updated manuscript further extends the expression of aSMA (activated fibroblast marker) to TdTomato+ cells to the acute phase following MI ($\pm 60\%$, 4 days post-MI; Figure S9E, IgG control group). We have revised the Results section accordingly to emphasize this point.

The updated text now reads:

“To directly quantify myofibroblasts and macrophages and their spatial arrangement *in situ*, we developed a macrophage-myofibroblast double reporter mouse line, ‘MAMY’, in which cardiac myofibroblasts are labeled with tdTomato and macrophages with GFP. The MAMY mouse is based on the extensively validated global monocyte-macrophage reporter: $Cx3cr1^{GFP/GFP}$ ^{48–50}, and a tamoxifen-inducible Cre-Lox system, $Postn^{MCM/MCM}$ ^{51,52} crossed with $Rosa26^{TdTomato/TdTomato}$ ^{51,52} mice, a lineage reporter that robustly labels cardiac myofibroblasts ^{47,51} (Figure 1C, Methods).”

We have also collected and integrated data from a large number of publications that used these mouse lines and quantified their specificity, label efficacy, and off-target effects to add a new supplementary text section (Note S1).

Note S1 now reads:

“Note S1: Specificity, label efficacy, and off-target effects of the mouse lines comprising the MAMY mouse.

The myofibroblast lineage reporter (Postn^{MCM/+}Rosa26^{TdTomto/+}): The Postn^{MCM} mouse line was established in 2016 by *Kanisicak, et al.*¹, and has been used by multiple labs as a lineage reporter and as a tool to specifically knock-out genes in cardiac myofibroblasts. It has been cited >800 times.

Kanisicak, et al. showed that Periostin is exclusively expressed in injured areas and is a very specific marker of myofibroblasts in the heart and other tissues following injury. Expression of eGFP under the control of the Postn^{MCM} cassette and exposure of these mice to tamoxifen did not label almost any cells in the uninjured heart. In other tissues, minimal off-target labeling was further demonstrated in the absence of injury. This includes skeletal muscle, kidney, lung, liver, and skin following 8 weeks of tamoxifen exposure. The Postn^{MCM} cassette successfully labeled cells only at the infarcted area (7 days post-MI). Whole mount tracking of these hearts up to 7 days following MI, revealed the progressive accumulation of eGFP+ cells in the infarct zone over time. These data were reproduced by a study that tested the dynamics of *Postn* gene expression following MI in mice².

Kanisicak, et al. further characterized the label efficiency and specificity of the Postn^{MCM} lineage tracing system in the context of acute, (namely, myocardial infarction; MI) and chronic (namely, transverse aortic constriction; TAC or AngII osmotic pumps) injuries.

In the context of MI, using histological sections stained by immunofluorescence, *Kanisicak, et al.* show that 2 weeks following MI induction and tamoxifen administration, the vast majority of eGFP-labeled cells were Vimentin (fibroblast marker) positive ($\pm 98\%$), $>50\%$ were PDGFR α (fibroblast marker) positive and $\pm 80\%$ were α SMA (activated fibroblast marker) positive. But, none were CD31 (endothelial cells marker), or CD45 (immune cells marker). Consistent with these results the authors further bolster their findings with flow cytometry analysis.

To obtain higher granularity in the eGFP $^{+}$ cells characterization, *Kanisicak, et al.* sorted 3 populations of cells and analyzed them by RNA-sequencing: 1) Resident fibroblasts defined by CD31-CD45- markers. 2) Activated injured zone cells, defined by Postn-CD31-CD45- markers, and 3) Activated *Postn*-tracked eGFP $^{+}$ cells that were also CD31-CD45-. Indeed among these cells, only the postn-traced eGFP $^{+}$ cells significantly expressed myofibroblast-specific genes such as *α SMA*, *Lox*, and *Fibronectin*^{3,4}.

In the context of chronic cardiac injury (TAC) the authors were able to show that eGFP $^{+}$ cells were Vimentin and α SMA positive, But, not CD31 or CD45 positive. These data were consistent with the demonstrated specificity of the lineage reporter following MI.

In summary, the Postn^{MCM} lineage tracing system is a robust and validated tool for specifically labeling myofibroblasts in the context of cardiac injury and other tissue injuries. The work by *Kanisicak, et al.*, supported by subsequent studies⁵⁻¹⁴, demonstrates that Periostin expression is specific to myofibroblasts in injured tissues, with minimal off-target effects in uninjured regions or non-myofibroblast cells. This system has been demonstrated to reliably identify myofibroblasts across multiple experimental models, including myocardial infarction, pressure overload, and fibrotic responses in other organs⁵⁻¹⁴.

The monocyte/macrophage reporter (Cx3cr1^{GFP/+}): The Cx3cr1^{GFP} mouse model is widely used in the field of monocyte/macrophage biology. It was established by *Jung, et al.*¹⁵, a paper cited >2800 times. In the context of the heart, the Cx3cr1^{GFP} mouse line has been extensively validated to specifically and reliably label all cardiac macrophage subsets through the expression of green fluorescent protein (GFP)¹⁶⁻²⁰.

In 2012, *Pinto, et al.*¹⁹ established the specificity of Cx3cr1^{GFP+} cells in cardiac tissue, as ±95% of these cells also express myeloid cell markers, such as CD45 and CD11b. They further show that the vast majority of these cells were expressing F4/80, MHCII, CD14, and CD86- highly abundant myeloid and macrophage markers. Importantly, these cells did not express NK1.1, B220, or CD3e- NK, B, and T cell markers.

In 2014, *Malawi, et al.*²⁰ repeated these results and established that >80% of cardiac macrophages, defined by core macrophage markers (CD64, F4/80, and MerTK) are indeed GFP+ cells.

In 2017, *Hulsmans, et al.*¹⁸, repeated these results using a flow cytometry approach, establishing that CD45+CD31-CD11b+Lin-Ly6G-F4/80+ cells were indeed GFP+. importantly, CD45-CD31-MEFSK4+ cells (fibroblasts) were negative for GFP.

Hulsmans, et al., then sorted (by flow cytometry) either GFP+ cells, total macrophages, or fibroblasts and performed real-time qPCR for key macrophage, fibroblast, and myofibroblast genes. They found that GFP+ cells were expressing macrophage genes (i.e. *Csf1r*, *F4/80*, *LysM*, *MerTK*, *Cd64*, etc.) as the sorted macrophages, and did not express fibroblast (i.e. *Col1a2*, *Col3a1*, *Ddr2*) or myofibroblast genes (i.e. *Postn*).

In 2018, *Walter, et al.*¹⁶, sorted GFP+ cells from cryo-injured (a form of acute cardiac injury) Cx3cr1^{GFP} hearts and performed RNA-sequencing at different time points following injury. This experiment allowed them to observe in detail the gene expression of GFP+ cells in the steady-state and injured heart. Indeed, the cardiac GFP+ cells gene expression strongly correlated with sorted macrophage RNA-sequencing data from a variety of tissues including the lung, liver, spleen, and peritoneum.

In summary, the MAMY mouse model combines two extensively validated and widely adopted genetic tools: the Cx3cr1^{GFP} system, which has been used in thousands of studies to reliably and specifically label macrophages, including cardiac macrophages,^{16,18-20} and the Postn^{MCM} lineage reporter system, which is highly specific for identifying myofibroblasts in cardiac and other injured tissues^{1,5-14}. Demonstrated through immunophenotyping, flow cytometry, and transcriptomic analyses in multiple studies, The Cx3cr1^{GFP} reporter has consistently shown robust specificity for monocytes/macrophages, with minimal off-target labeling of non-myeloid cell types including endothelial cells, lymphocytes, fibroblasts, and other stromal cells. Similarly, the Postn^{MCM} system has been shown to exclusively label myofibroblasts in the context of cardiac injury, with negligible labeling of other cell types. Together, the established specificity of these models ensures that the data generated using this model accurately captures cardiac myofibroblasts and all monocytes/macrophages in the injured zones, and further supports the validity of our findings using the MAMY mouse line.

Note S1 References

1. Kanisicak, O., Khalil, H., Ivey, M.J., Karch, J., Maliken, B.D., Correll, R.N., Brody, M.J., Lin, S.-C.J., Aronow, B.J., Tallquist, M.D., et al. (2016). Genetic lineage tracing defines myofibroblast origin and function in the injured heart. *Nat. Commun.* 7, 12260. <https://doi.org/10.1038/ncomms12260>.
2. Gil, H., Goldshtein, M., Etzion, S., Elyagon, S., Hadad, U., Etzion, Y., and Cohen, S. (2022). Defining the timeline of periostin upregulation in cardiac fibrosis following acute myocardial infarction in mice. *Sci. Rep.* 12, 21863. <https://doi.org/10.1038/s41598-022-26035-y>.
3. Forte, E., Skelly, D.A., Chen, M., Daigle, S., Morelli, K.A., Hon, O., Philip, V.M., Costa, M.W., Rosenthal, N.A., and Furtado, M.B. (2020). Dynamic Interstitial Cell Response during Myocardial Infarction Predicts Resilience to Rupture in Genetically Diverse Mice. *Cell Reports* 30, 3149-3163.e6. <https://doi.org/10.1016/j.celrep.2020.02.008>.
4. Farbehi, N., Patrick, R., Dorison, A., Xaymardan, M., Janbandhu, V., Wystub-Lis, K., Ho, J.W., Nordon, R.E., and Harvey, R.P. (2019). Single-cell expression profiling reveals dynamic flux of cardiac stromal, vascular and immune cells in health and injury. *eLife* 8, e43882. <https://doi.org/10.7554/elife.43882>.
5. Khalil, H., Kanisicak, O., Prasad, V., Correll, R.N., Fu, X., Schips, T., Vagnozzi, R.J., Liu, R., Huynh, T., Lee, S.-J., et al. (2017). Fibroblast-specific TGF- β -Smad2/3 signaling underlies cardiac fibrosis. *J. Clin. Investig.* 127, 3770–3783. <https://doi.org/10.1172/jci94753>.
6. Fu, X., Khalil, H., Kanisicak, O., Boyer, J.G., Vagnozzi, R.J., Maliken, B.D., Sargent, M.A., Prasad, V., Valiente-Alandi, I., Blaxall, B.C., et al. (2018). Specialized fibroblast differentiated states underlie scar formation in the infarcted mouse heart. *J Clin Invest* 128, 2127–2143. <https://doi.org/10.1172/jci98215>.
7. Aghajanian, H., Kimura, T., Rurik, J.G., Hancock, A.S., Leibowitz, M.S., Li, L., Scholler, J., Monslow, J., Lo, A., Han, W., et al. (2019). Targeting Cardiac Fibrosis with Engineered T cells. *Nature* 573, 430–433. <https://doi.org/10.1038/s41586-019-1546-z>.
8. Meng, Q., Bhandary, B., Bhuiyan, M.S., James, J., Osinska, H., Valiente-Alandi, I., Shay-Winkler, K., Gulick, J., Molkentin, J.D., Blaxall, B.C., et al. (2018). Myofibroblast-Specific TGF β Receptor II Signaling in the Fibrotic Response to Cardiac Myosin Binding Protein C-Induced Cardiomyopathy. *Circ. Res.* 123, 1285–1297. <https://doi.org/10.1161/circresaha.118.313089>.
9. Huo, J.-L., Jiao, L., An, Q., Chen, X., Qi, Y., Wei, B., Zheng, Y., Shi, X., Gao, E., Liu, H.-M., et al. (2021). Myofibroblast Deficiency of LSD1 Alleviates TAC-Induced Heart Failure. *Circ. Res.* 129, 400–413. <https://doi.org/10.1161/circresaha.120.318149>.
10. Xiang, F.-L., Fang, M., and Yutzey, K.E. (2017). Loss of β -catenin in resident cardiac fibroblasts attenuates fibrosis induced by pressure overload in mice. *Nat. Commun.* 8, 712. <https://doi.org/10.1038/s41467-017-00840-w>.
11. Meng, Q., Yang, B., Qiao, Y., Wu, Y., Chen, J., Lin, X., and Molkentin, J.D. (2024). Genetic and Pharmacologic Inhibition of JAK1/2 Antagonizes Cardiac Fibrosis. *Circulation* 150, 899–901. <https://doi.org/10.1161/circulationaha.124.070340>.

12. Hortells, L., Valiente-Alandi, I., Thomas, Z.M., Agnew, E.J., Schnell, D.J., York, A.J., Vagnozzi, R.J., Meyer, E.C., Molkentin, J.D., and Yutzey, K.E. (2020). A specialized population of Periostin-expressing cardiac fibroblasts contributes to postnatal cardiomyocyte maturation and innervation. *Proc. Natl. Acad. Sci.* 117, 21469–21479. <https://doi.org/10.1073/pnas.2009119117>.
13. Ock, S., Ham, W., Kang, C.W., Kang, H., Lee, W.S., and Kim, J. (2021). IGF-1 protects against angiotensin II-induced cardiac fibrosis by targeting α SMA. *Cell Death Dis.* 12, 688. <https://doi.org/10.1038/s41419-021-03965-5>.
14. Liu, H., Zhang, S., Xu, S., Koroleva, M., Small, E.M., and Jin, Z.G. (2019). Myofibroblast-specific YY1 promotes liver fibrosis. *Biochem. Biophys. Res. Commun.* 514, 913–918. <https://doi.org/10.1016/j.bbrc.2019.05.004>.
15. Jung, S., Aliberti, J., Graemmel, P., Sunshine, M.J., Kreutzberg, G.W., Sher, A., and Littman, D.R. (2000). Analysis of Fractalkine Receptor CX3CR1 Function by Targeted Deletion and Green Fluorescent Protein Reporter Gene Insertion. *Mol. Cell. Biol.* 20, 4106–4114. <https://doi.org/10.1128/mcb.20.11.4106-4114.2000>.
16. Walter, W., Alonso-Herranz, L., Trappetti, V., Crespo, I., Ibberson, M., Cedenilla, M., Karaszewska, A., Núñez, V., Xenarios, I., Arroyo, A.G., et al. (2018). Deciphering the Dynamic Transcriptional and Post-transcriptional Networks of Macrophages in the Healthy Heart and after Myocardial Injury. *Cell Rep.* 23, 622–636. <https://doi.org/10.1016/j.celrep.2018.03.029>.
17. Heidt, T., Courties, G., Dutta, P., Sager, H.B., Sebas, M., Iwamoto, Y., Sun, Y., Silva, N.D., Panizzi, P., Laan, A.M. van der, et al. (2014). Differential Contribution of Monocytes to Heart Macrophages in Steady-State and After Myocardial Infarction. *Circ. Res.* 115, 284–295. <https://doi.org/10.1161/circresaha.115.303567>.
18. Hulsmans, M., Clauss, S., Xiao, L., Aguirre, A.D., King, K.R., Hanley, A., Hucker, W.J., Wülfers, E.M., Seemann, G., Courties, G., et al. (2017). Macrophages Facilitate Electrical Conduction in the Heart. *Cell* 169, 510-522.e20. <https://doi.org/10.1016/j.cell.2017.03.050>.
19. Pinto, A.R., Paolicelli, R., Salimova, E., Gospocic, J., Slonimsky, E., Bilbao-Cortes, D., Godwin, J.W., and Rosenthal, N.A. (2012). An Abundant Tissue Macrophage Population in the Adult Murine Heart with a Distinct Alternatively-Activated Macrophage Profile. *PLoS ONE* 7, e36814. <https://doi.org/10.1371/journal.pone.0036814>.
20. Molawi, K., Wolf, Y., Kandalla, P.K., Favret, J., Hagemeyer, N., Frenzel, K., Pinto, A.R., Klapproth, K., Henri, S., Malissen, B., et al. (2014). Progressive replacement of embryo-derived cardiac macrophages with age. *J. Exp. Med.* 211, 2151–2158. <https://doi.org/10.1084/jem.20140639>.

”

Comment #2: How does the MAMY model add anything over a simpler approach such as immunostaining of myofibroblasts and macrophages in the models? Could immunostaining be added for the ICR mice to confirm the spatial location of the macrophages and the fact that day 65 represents cold fibrosis?

Response: The dynamic nature of gene and protein expression in cardiac myofibroblasts and monocyte/macrophage populations has been characterized in the revised manuscript (Figure 4) and previously by numerous studies^{1,4,13,17-20}. To our knowledge, no single protein marker labels all monocyte/macrophage or myofibroblast populations as efficiently and comprehensively as the CX3CR1^{GFP/+} knock-in mouse and the Postn^{MCM/+};Rosa26^{TdTomato/+} mouse, respectively. This is particularly valuable for longitudinal studies following injury. The Postn marker serves as a lineage reporter. This is an advantage over immunostaining which captures Postn expression as a snapshot in time. Since Postn expression is dynamic^{1,21}, lineage tracing offers a valuable advantage. As for macrophages, GFP half-life is on the order of a day² whereas CX3CR1 is a dynamic protein with a shorter half-life²², therefore the MAMY system offers a longer view than immunostaining.

This is the first time these two validated mouse models have been combined to study macrophages and myofibroblasts in the same tissue, enabling the assessment of their spatial relationships during cardiac fibrosis development. Given their prior reliability in labeling macrophages and myofibroblasts in other tissues^{1,2,23-25}, the MAMY mouse could also be used to study different forms of fibrosis.

Although adding immunostaining at day 65 would support the conclusion of cold fibrosis, we note that the paper has already multiple independent lines of evidence for cold fibrosis in MI.

This includes single-cell RNA sequencing (scRNA-seq), flow cytometry, immunofluorescence, Visium spatial omics, and detailed cross-species analyses in the clinically relevant porcine model and human MI samples. We further complement these with *in-silico* modeling, and cell-cell interactions that further validate cold fibrosis as a state controlled by an autocrine growth-factor loop, and identify a factor among this loop for further *in-vitro* and *in-vivo* perturbations. These allowed us to demonstrate and characterize the macrophage and myofibroblast population dynamic following MI, their spatial localization, their division of labor in the development of cold fibrosis, and their cellular communication in that context. Given this comprehensive and multifaceted validation of cold fibrosis, we feel that additional immunostaining in ICR mice is not essential to further substantiate our conclusions.

Comment #3: The regenerative (P1 mice) and non-regenerative model (P8 mice) is not well described (Fig s7D). Please explain this better.

Response: We have now clarified the text and references regarding the regenerative and non-regenerative models of cardiac regeneration.

In the Results, we write: “The P1 (postnatal day 1) mouse heart is known for its robust regenerative potential²⁸. Timp1 shows a difference in expression in the regenerative (P1) versus non-regenerative hearts (P8-8 day old mice) in response to MI - it peaked and dropped quickly in P1 hearts following MI and remained high in post-MI P8 hearts (Figure S7D).”

Comment #4: LRP1 signalling: I still think it would have been helpful to either block or knockdown LRP1 in fibroblasts to show this abrogates TIMP1 induced proliferation. As a minimum, the fact that the mechanism regulating this process is not defined should be included in the discussion.

Response: We have revised the Discussion to address this point. The updated text now reads:

“We show that TIMP1 acts as a growth factor for cardiac myofibroblast and suggest that this activity is due to interaction with LRP1, but not with CD63 or APP. Future studies employing direct blocking or knockdown of LRP1 are necessary to fully clarify its role in TIMP1-induced proliferation.”

Comment #5: Typo Fig S3J - should be "none" in the table

Response: Fixed. Thanks!

Comment #6: Typo - Toe "pinche" in the methods should be pinch

Response: Fixed. Thanks!

Response to Reviewers References

1. Kanisicak, O., Khalil, H., Ivey, M.J., Karch, J., Maliken, B.D., Correll, R.N., Brody, M.J., Lin, S.-C.J., Aronow, B.J., Tallquist, M.D., et al. (2016). Genetic lineage tracing defines myofibroblast origin and function in the injured heart. *Nat. Commun.* 7, 12260. <https://doi.org/10.1038/ncomms12260>.
2. Jung, S., Aliberti, J., Graemmel, P., Sunshine, M.J., Kreutzberg, G.W., Sher, A., and Littman, D.R. (2000). Analysis of Fractalkine Receptor CX3CR1 Function by Targeted Deletion and Green Fluorescent Protein Reporter Gene Insertion. *Mol. Cell. Biol.* 20, 4106–4114. <https://doi.org/10.1128/mcb.20.11.4106-4114.2000>.
3. Khalil, H., Kanisicak, O., Prasad, V., Correll, R.N., Fu, X., Schips, T., Vagnozzi, R.J., Liu, R., Huynh, T., Lee, S.-J., et al. (2017). Fibroblast-specific TGF- β -Smad2/3 signaling underlies cardiac fibrosis. *J. Clin. Investig.* 127, 3770–3783. <https://doi.org/10.1172/jci94753>.
4. Fu, X., Khalil, H., Kanisicak, O., Boyer, J.G., Vagnozzi, R.J., Maliken, B.D., Sargent, M.A., Prasad, V., Valiente-Alandi, I., Blaxall, B.C., et al. (2018). Specialized fibroblast differentiated states underlie scar formation in the infarcted mouse heart. *J Clin Invest* 128, 2127–2143. <https://doi.org/10.1172/jci98215>.
5. Aghajanian, H., Kimura, T., Rurik, J.G., Hancock, A.S., Leibowitz, M.S., Li, L., Scholler, J., Monslow, J., Lo, A., Han, W., et al. (2019). Targeting Cardiac Fibrosis with Engineered T cells. *Nature* 573, 430–433. <https://doi.org/10.1038/s41586-019-1546-z>.
6. Meng, Q., Bhandary, B., Bhuiyan, M.S., James, J., Osinska, H., Valiente-Alandi, I., Shay-Winkler, K., Gulick, J., Molkentin, J.D., Blaxall, B.C., et al. (2018). Myofibroblast-Specific TGF β Receptor II Signaling in the Fibrotic Response to Cardiac Myosin Binding Protein C-Induced Cardiomyopathy. *Circ. Res.* 123, 1285–1297. <https://doi.org/10.1161/circresaha.118.313089>.
7. Huo, J.-L., Jiao, L., An, Q., Chen, X., Qi, Y., Wei, B., Zheng, Y., Shi, X., Gao, E., Liu, H.-M., et al. (2021). Myofibroblast Deficiency of LSD1 Alleviates TAC-Induced Heart Failure. *Circ. Res.* 129, 400–413. <https://doi.org/10.1161/circresaha.120.318149>.
8. Xiang, F.-L., Fang, M., and Yutzey, K.E. (2017). Loss of β -catenin in resident cardiac fibroblasts attenuates fibrosis induced by pressure overload in mice. *Nat. Commun.* 8, 712. <https://doi.org/10.1038/s41467-017-00840-w>.
9. Meng, Q., Yang, B., Qiao, Y., Wu, Y., Chen, J., Lin, X., and Molkentin, J.D. (2024). Genetic and Pharmacologic Inhibition of JAK1/2 Antagonizes Cardiac Fibrosis. *Circulation* 150, 899–901. <https://doi.org/10.1161/circulationaha.124.070340>.
10. Hortells, L., Valiente-Alandi, I., Thomas, Z.M., Agnew, E.J., Schnell, D.J., York, A.J., Vagnozzi, R.J., Meyer, E.C., Molkentin, J.D., and Yutzey, K.E. (2020). A specialized population of Periostin-expressing cardiac fibroblasts contributes to postnatal cardiomyocyte maturation and innervation. *Proc. Natl. Acad. Sci.* 117, 21469–21479. <https://doi.org/10.1073/pnas.2009119117>.
11. Ock, S., Ham, W., Kang, C.W., Kang, H., Lee, W.S., and Kim, J. (2021). IGF-1 protects against angiotensin II-induced cardiac fibrosis by targeting α SMA. *Cell Death Dis.* 12, 688.

<https://doi.org/10.1038/s41419-021-03965-5>.

12. Liu, H., Zhang, S., Xu, S., Koroleva, M., Small, E.M., and Jin, Z.G. (2019). Myofibroblast-specific YY1 promotes liver fibrosis. *Biochem. Biophys. Res. Commun.* *514*, 913–918. <https://doi.org/10.1016/j.bbrc.2019.05.004>.
13. Walter, W., Alonso-Herranz, L., Trappetti, V., Crespo, I., Ibberson, M., Cedenilla, M., Karaszewska, A., Núñez, V., Xenarios, I., Arroyo, A.G., et al. (2018). Deciphering the Dynamic Transcriptional and Post-transcriptional Networks of Macrophages in the Healthy Heart and after Myocardial Injury. *Cell Rep.* *23*, 622–636. <https://doi.org/10.1016/j.celrep.2018.03.029>.
14. Heidt, T., Courties, G., Dutta, P., Sager, H.B., Sebas, M., Iwamoto, Y., Sun, Y., Silva, N.D., Panizzi, P., Laan, A.M. van der, et al. (2014). Differential Contribution of Monocytes to Heart Macrophages in Steady-State and After Myocardial Infarction. *Circ. Res.* *115*, 284–295. <https://doi.org/10.1161/circresaha.115.303567>.
15. Hulsmans, M., Clauss, S., Xiao, L., Aguirre, A.D., King, K.R., Hanley, A., Hucker, W.J., Wülfers, E.M., Seemann, G., Courties, G., et al. (2017). Macrophages Facilitate Electrical Conduction in the Heart. *Cell* *169*, 510–522.e20. <https://doi.org/10.1016/j.cell.2017.03.050>.
16. Pinto, A.R., Paolicelli, R., Salimova, E., Gospocic, J., Slonimsky, E., Bilbao-Cortes, D., Godwin, J.W., and Rosenthal, N.A. (2012). An Abundant Tissue Macrophage Population in the Adult Murine Heart with a Distinct Alternatively-Activated Macrophage Profile. *PLoS ONE* *7*, e36814. <https://doi.org/10.1371/journal.pone.0036814>.
17. Molawi, K., Wolf, Y., Kandalla, P.K., Favret, J., Hagemeyer, N., Frenzel, K., Pinto, A.R., Klapproth, K., Henri, S., Malissen, B., et al. (2014). Progressive replacement of embryo-derived cardiac macrophages with age. *J. Exp. Med.* *211*, 2151–2158. <https://doi.org/10.1084/jem.20140639>.
18. Dick, S.A., Macklin, J.A., Nejat, S., Momen, A., Clemente-Casares, X., Althagafi, M.G., Chen, J., Kantores, C., Hosseinzadeh, S., Aronoff, L., et al. (2019). Self-renewing resident cardiac macrophages limit adverse remodeling following myocardial infarction. *Nat Immunol* *20*, 29–39. <https://doi.org/10.1038/s41590-018-0272-2>.
19. Dick, S.A., Wong, A., Hamidzada, H., Nejat, S., Nechanitzky, R., Vohra, S., Mueller, B., Zaman, R., Kantores, C., Aronoff, L., et al. (2022). Three tissue resident macrophage subsets coexist across organs with conserved origins and life cycles. *Sci. Immunol.* *7*, eabf7777. <https://doi.org/10.1126/sciimmunol.abf7777>.
20. Zaman, R., Hamidzada, H., Kantores, C., Wong, A., Dick, S.A., Wang, Y., Momen, A., Aronoff, L., Lin, J., Razani, B., et al. (2021). Selective loss of resident macrophage-derived insulin-like growth factor-1 abolishes adaptive cardiac growth to stress. *Immunity* *54*, 2057–2071.e6. <https://doi.org/10.1016/j.immuni.2021.07.006>.
21. Gil, H., Goldshtein, M., Etzion, S., Elyagon, S., Hadad, U., Etzion, Y., and Cohen, S. (2022). Defining the timeline of periostin upregulation in cardiac fibrosis following acute myocardial infarction in mice. *Sci. Rep.* *12*, 21863. <https://doi.org/10.1038/s41598-022-26035-y>.
22. Hamon, P., Loyher, P.-L., Chanville, C.B. de, Licata, F., Combadière, C., and Boissonnas, A. (2017). CX3CR1-dependent endothelial margination modulates Ly6Chigh monocyte systemic

deployment upon inflammation in mice. *Blood* 129, 1296–1307.

<https://doi.org/10.1182/blood-2016-08-732164>.

23. Vietinghoff, S. von, and Kurts, C. (2021). Regulation and function of CX3CR1 and its ligand CX3CL1 in kidney disease. *Cell Tissue Res.* 385, 335–344.

<https://doi.org/10.1007/s00441-021-03473-0>.

24. David, B.A., Rezende, R.M., Antunes, M.M., Santos, M.M., Lopes, M.A.F., Diniz, A.B., Pereira, R.V.S., Marchesi, S.C., Alvarenga, D.M., Nakagaki, B.N., et al. (2016). Combination of Mass Cytometry and Imaging Analysis Reveals Origin, Location, and Functional Repopulation of Liver Myeloid Cells in Mice. *Gastroenterology* 151, 1176–1191.

<https://doi.org/10.1053/j.gastro.2016.08.024>.

25. Paschalis, E.I., Lei, F., Zhou, C., Kapoulea, V., Thanos, A., Dana, R., Vavvas, D.G., Chodosh, J., and Dohlman, C.H. (2018). The Role of Microglia and Peripheral Monocytes in Retinal Damage after Corneal Chemical Injury. *Am. J. Pathol.* 188, 1580–1596.

<https://doi.org/10.1016/j.ajpath.2018.03.005>.

Below is our point-by-point response (in blue) to reviewers' comments.

Reviewer comments:

Reviewer #3: The authors have addressed all of my previous comments with this revised manuscript. I congratulate them on this really important body of work.

We thank the reviewer for this endorsement.

I have a couple of remaining minor points:

1) Please consider the color schemes when comparing macrophages and myfibroblasts (e.g Fig 1B, C, H or Fig 2J). These are currently red and green and it would be challenge for readers with red green color blindness to interpret these figures.

Response: We thank the reviewer for this comment which helped us to improve the readability and accessibility of our manuscript.

We maintained the standard colors for Green Fluorescence Protein (GFP; green) and TdTomato (red) to align with widely recognized fluorescent protein nomenclature, ensuring clarity for readers familiar with these markers. Additionally, we confirmed through direct feedback from color-blind colleagues that the adaptations made in the manuscript provide sufficient clarity for all readers.

- In Figure 1B, we changed the red myofibroblast line to a striated pattern for improved clarity.
- In Figure 1C, we validated that the distinct shapes associated with the specific mouse genotype names (Postn^{MCM/+}Rosa26^{TdTomto/TdTomato} or Cx3cr1^{GFP/GFP}) are sufficient for interpretation without relying on color.
- In Figures 1H and 2J, we ensured that the signals in the separated GFP and TdTomato channels are distinguishable, allowing readers with red-green color blindness to interpret the data independently.

2) I'd suggest referring to note S1 in the main text/results.

Response: We have changed the main text in the Results section to include a direct reference to Note S1.

The text now reads: “The MAMY mouse is based on the extensively validated global monocyte-macrophage reporter: $Cx3cr1^{GFP/GFP}$ ^{48–50}, and a tamoxifen-inducible Cre-Lox system, $Postn^{MCM/MCM}$ ^{51,52} crossed with $Rosa26^{TdTomato/TdTomato}$ ^{51,52} mice, a lineage reporter that robustly labels cardiac myofibroblasts^{47,51} (Figure 1C, Methods, Note S1). ”

3) Please review the interpretation of Fig S9E - they describe that TIMP1 blockade did not reduce a-SMA myofibroblasts, but there appears to be a trend to reduction in these data. Would you not expect the a-SMA population to be reduced with TIMP1 inhibition?

Response: We agree with the reviewer and changed the text in the Results section accordingly.

The text now reads: “Moreover, while α TIMP1 treatment did not significantly attenuate the fraction of aSMA+ myofibroblasts, a slight trend was observed (Figure S9E).“